# MEC: Machine-Learning-Assisted Generalized Entropy Calibration for Semi-Supervised Mean Estimation

Se Yoon Lee [1]  Jae Kwang Kim [2]

## Abstract

Obtaining high-quality labels is costly, whereas unlabeled covariates are often abundant, motivating semi-supervised inference methods with reliable uncertainty quantification. Prediction-powered inference (PPI) leverages a machine-learning predictor trained on a small labeled sample to improve efficiency, but it can lose efficiency under model misspecification and suffer from coverage distortions due to label reuse. We introduce Machine-Learning-Assisted Generalized Entropy Calibration (MEC), a cross-fitted, calibration-weighted variant of PPI. MEC improves efficiency by reweighting labeled samples to better align with the target population, using a principled calibration framework based on Bregman projections. This yields robustness to affine transformations of the predictor and relaxes requirements for validity by replacing conditions on raw prediction error with weaker projection-error conditions. As a result, MEC attains the semiparametric efficiency bound under weaker assumptions than existing PPI variants. Across simulations and a real-data application, MEC achieves near-nominal coverage and tighter confidence intervals than CF-PPI and vanilla PPI.

## 1. Introduction

In many inference problems, high-quality labels are scarce while unlabeled covariates are abundant. Recent advancements in machine learning (ML) offer substantial potential to mitigate the reliance on gold-standard data while ensuring valid scientific discovery. Semi-supervised inference capitalizes on this setting by combining a small labeled sample with a large unlabeled sample to estimate population quantities with valid uncertainty quantification (Zhang et al.,

2019; Motwani & Witten, 2023; Wang et al., 2020).

A powerful recent approach to this problem is prediction-powered inference (PPI) (Angelopoulos et al., 2023). The core idea is elegant: use an ML predictor to impute outcomes on all unlabeled samples, then correct for bias using labeled residuals. PPI is broadly applicable to many scientific problems, as it can leverage flexible ML models, including deep neural networks (LeCun et al., 2015), gradient-boosted trees (Chen & Guestrin, 2016), random forests (Breiman, 2001), and BART (Chipman et al., 2010).

However, in practice, the efficiency of PPI can deteriorate when the predictive model is imperfect, the labeled fraction is small, or the PPI is applied incautiously. In response, multiple variants have been proposed. Angelopoulos et al. (2024) develop PPI++, an efficiency-enhanced version of PPI that optimizes how predictions and labels are combined, potentially yielding tighter confidence intervals than the original PPI. Zrnic & Candès (2024) introduce a cross-prediction technique that trains the predictor out of fold via sample splitting to avoid overfitting from label reuse; in this paper, we refer to their method as cross-fitted PPI (CF–PPI). Other variants of PPI include Fisch et al. (2024); Gu & Xia (2024); Einbinder et al. (2025); Luo et al. (2024), reflecting the growing popularity and adaptability of the PPI framework in modern semi-supervised inference.

In this paper, we present a new PPI variant inspired by weight calibration (Deville & Särndal, 1992): **Machine-Learning-Assisted Generalized Entropy Calibration (ML-GEC; hereafter, MEC)**. MEC adapts generalized entropy calibration (GEC) (Kwon et al., 2025) to the PPI framework, utilizing Bregman projections to calibrate weights for enhanced efficiency. This approach parallels long-standing practices in survey sampling—where carefully chosen weights mitigate bias and improve efficiency (Fuller, 2002; Hainmueller, 2012)—but with a crucial distinction. While classical GEC is typically applied to finite-population mean estimation with prediction rules restricted to the linear span of a preset basis, MEC targets superpopulation mean estimation for semi-supervised inference. Crucially, MEC accommodates essentially arbitrary ML predictors to construct the calibration basis, thus fully leveraging the expressive power of modern ML.

---

[1]Texas A&M University [2]Iowa State University. Correspondence to: Se Yoon Lee <seyoonlee.stat.math@gmail.com>.

*Proceedings of the 43rd International Conference on Machine Learning*, Seoul, Korea. PMLR 306, 2026. Copyright 2026 by the author(s).

We prove that the MEC estimator enjoys desirable asymptotic properties under standard regularity conditions. Theoretically, MEC relaxes the requirements for consistency and asymptotic normality by replacing assumptions on the raw prediction error with weaker assumptions on its projection error. This is enabled by using an out-of-fold predictor for both cross-prediction and calibration-basis construction. We further show that MEC is robust to affine transformations of the predictor, thereby reducing misspecification-driven variance inflation and attaining the semiparametric efficiency bound under weaker conditions than CF-PPI/PPI. We also show that MEC generalizes PPI++ in a dual sense (calibration form vs. regression form), and they are asymptotically equivalent when the generator is quadratic. MEC is computationally stable and fast because the calibration basis is always two-dimensional, regardless of the covariate dimension. Simulations corroborate these guarantees, demonstrating improved coverage and robustness to moderate misspecification across diverse scenarios and varying labeled data fractions.

The article is organized as follows. Section 2 introduces notation and basic setup. Section 3 reviews the related works. Section 4 develops the MEC formulation and algorithm. Section 5 presents theoretical guarantees. Section 6 reports simulation experiments. Section 7 discusses practical considerations and extensions of MEC, and Section 8 concludes with broader implications and future directions.

## 2. Basic setup

We study statistical inference for semi-supervised mean estimation, where collecting high-quality labels is challenging but feature observations are abundant. We posit the outcome-regression model $Y = m_0(X) + \varepsilon$ for a generic draw $(X, Y) \sim P$, with $\mathbb{E}[\varepsilon \mid X] = 0$ and $\text{Var}[\varepsilon \mid X] < \infty$, where $m_0(x) := \mathbb{E}[Y \mid X = x]$ denotes the true regression function. The inferential target is the superpopulation mean outcome $\theta_0 = \mathbb{E}[Y]$.

We work in an assumption-lean framework, following recent developments in semiparametric methods and the general PPI framework (Wang et al., 2020; Zhang et al., 2019; Chernozhukov et al., 2018; Van der Laan & Rose, 2011; Kennedy, 2024; Miao et al., 2025): (i) no parametric model is imposed on $m_0$; (ii) no specific distributional assumptions are imposed on the joint law $P$—the only assumptions are standard regularity conditions (e.g., finite second moments) for asymptotic results; and (iii) no parametric or explicit convergence rate conditions on the ML predictor used downstream.

We introduce the semi-supervised data structure. We observe two independent samples: an unlabeled i.i.d. covariate sample of size $N$, $\{X_i\}_{i=1}^N \overset{\text{i.i.d.}}{\sim} P_X$, and an independent

labeled i.i.d. sample of size $n$, $\{(X_j, Y_j)\}_{j \in S} \overset{\text{i.i.d.}}{\sim} P$. Here $S$ is an index set with $|S| = n$, $P_X$ is the $X$-marginal of $P$, and $X \in \mathcal{X} \subset \mathbb{R}^d$. We focus on settings where the unlabeled covariate sample is typically much larger than the labeled sample, as in applications where collecting features is far cheaper than acquiring labels. Let $f_N := n/N$ denote the label fraction. Throughout, $N \to \infty$, $n = n(N) \to \infty$, and $f_N \to f \in (0, 1)$; for brevity we write $f$ for $f_N$.

## 3. Related works

### 3.1. Prediction-powered inference

Angelopoulos et al. (2023) proposed the PPI estimator of the superpopulation mean,

$$\widehat{\theta}_{\text{PPI},m} = \frac{1}{N} \sum_{i=1}^N m(X_i) + \frac{1}{n} \sum_{j \in S} \{Y_j - m(X_j)\}, \quad (1)$$

where $m : \mathcal{X} \to \mathbb{R}$ is a (possibly misspecified) prediction rule. For the moment, we treat $m$ as fixed—that is, not trained on the labeled data.

In (1), the first term is a plug-in prediction component computed over the unlabeled covariates using $m$, while the second term serves as a residual-correction based on the labeled data. By the linearity of expectation, we have $\mathbb{E}_P[\widehat{\theta}_{\text{PPI},m}] = \mathbb{E}_P[m(X)] + \mathbb{E}_P[Y] - \mathbb{E}_P[m(X)] = \mathbb{E}_P[Y] = \theta_0$ for any fixed predictor $m$. Consequently, $\widehat{\theta}_{\text{PPI},m}$ is an unbiased estimator of the population mean, irrespective of whether $m$ is misspecified. The choice of $m$ dictates the estimator's efficiency; specifically, the estimator is semiparametrically efficient if and only if the predictor coincides with the oracle regression function $m(x) = m_0(x) = \mathbb{E}[Y \mid X = x]$. (See Section C in the Appendix for theoretical details.)

Angelopoulos et al. (2023) show that, under mild regularity conditions, the $100(1 - \alpha)\%$ Wald-type confidence interval attains asymptotically nominal coverage (see Theorem S1 in (Angelopoulos et al., 2023) for details):

$$\liminf_{N, n(N) \to \infty} \mathbb{P}\big(\theta_0 \in C_\alpha^{\text{PPI},m}\big) \geq 1 - \alpha, \quad (2)$$

where

$$C_\alpha^{\text{PPI},m} := \left[\widehat{\theta}_{\text{PPI},m} \pm z_{1-\alpha/2}\, \widehat{\text{se}}_m\right], \quad (3)$$

$$\widehat{\text{se}}_m^2 = \frac{\widehat{\text{Var}}_N\{m(X)\}}{N} + \frac{\widehat{\text{Var}}_n\{Y - m(X)\}}{n},$$

where $z_{1-\alpha} = \Phi^{-1}(1 - \alpha)$ denotes the critical value of the standard normal distribution corresponding to cumulative probability $1 - \alpha$. Here $\widehat{\text{Var}}_N$ and $\widehat{\text{Var}}_n$ denote empirical variances over all $N$ covariates and over the labeled set $S$, respectively.

In this paper, we consider the setting where no pre-trained predictive model is available; therefore, the predictor $m$

must be trained using the labeled data $\{(X_j, Y_j)\}_{j \in S}$. This framework aligns with the outcome-regression perspective in causal inference under a known, constant propensity score (Kennedy, 2024; Chernozhukov et al., 2018; van der Laan & Rubin, 2006).

Let $\widehat{m}$ denote a model trained on the labeled sample $\{(X_j, Y_j)\}_{j \in S}$. In practice, $\widehat{m}$ is typically obtained using flexible ML algorithms such as random forests, gradient boosting, or deep neural networks. Replacing $m$ with $\widehat{m}$ yields the implemented PPI estimator

$$\widehat{\theta}_{\text{PPI}} = \frac{1}{N} \sum_{i=1}^{N} \widehat{m}(X_i) + \frac{1}{n} \sum_{j \in S} \{Y_j - \widehat{m}(X_j)\}. \quad (4)$$

An associated confidence interval is obtained by substituting $\widehat{m}$ for $m$ in (3).

We note that the vanilla PPI method (4)—which uses the single-fitted predictor $\widehat{m}$ learned from the labeled data without any additional safeguards or efficiency enhancements—has two major caveats:

**I. Label reuse.** The rectifier term in (4) reuses the labeled outcomes *both* to train $\widehat{m}$ and to evaluate residuals. When $\widehat{m}$ is highly flexible, this reuse can induce overfitting bias and distort variance estimation. Consequently, the $100(1 - \alpha)\%$ Wald-type confidence interval may fail to achieve nominal coverage.

**II. Efficiency shortfall.** PPI is not, in general, semiparametrically efficient. Its asymptotic variance is minimized only when the predictor coincides with the true regression function $m_0(x) = \mathbb{E}[Y \mid X = x]$; otherwise, misspecification inflates variance and degrades efficiency.

The primary objective of this paper is to address these challenges. Briefly, we resolve the former issue (**I**) via cross-fitting (sample splitting) and address the latter issue (**II**) through weight calibration, with a careful selection of the basis function. Notably, this calibration approach constitutes our main methodological contribution to the PPI framework.

### 3.2. Cross-fitted prediction-powered inference

Zrnic & Candès (2024) proposed CF–PPI, which uses cross-prediction to address the label-reuse issue in the vanilla PPI estimator (4). The central idea of cross-prediction is sample splitting, which is widely used in modern causal-inference workflows—including targeted learning and double ML—to mitigate overfitting of nuisance parameters and to ensure valid asymptotic inference (Newey & Robins, 2018; Chernozhukov et al., 2017; Zheng & van der Laan, 2010).

We illustrate the CF–PPI implementation. For a chosen integer $K$ (typically $K = 5$ or 10), partition the labeled set $S$ into $K$ folds $S^{(1)}, \ldots, S^{(K)}$. For each $k \in \{1, \ldots, K\}$, fit $\widehat{m}^{(k)}$ using the labels in $S \setminus S^{(k)}$. Let $\kappa(i) \in \{1, \ldots, K\}$

denote the fold index of unit $i \in S$. Define the out-of-fold predictor

$$\widehat{m}^{(-)}(X_i) := \begin{cases} \widehat{m}^{(\kappa(i))}(X_i), & i \in S, \\ \widehat{m}^{\star}(X_i), & i \notin S, \end{cases} \quad (5)$$

where $\widehat{m}^{\star}$ is an aggregate model used for the plug-in term (e.g., the full-sample fit or the average $K^{-1} \sum_{k=1}^{K} \widehat{m}^{(k)}$).

The CF–PPI estimator for the mean $\theta_0 = \mathbb{E}[Y]$ is then

$$\widehat{\theta}_{\text{PPI}}^{\text{cf}} = \frac{1}{N} \sum_{i=1}^{N} \widehat{m}^{(-)}(X_i) + \frac{1}{n} \sum_{j \in S} \{Y_j - \widehat{m}^{(-)}(X_j)\}, \quad (6)$$

where the single-fit predictor $\widehat{m}$ in (4) is replaced by the out-of-fold predictor $\widehat{m}^{(-)}$. An associated $100(1 - \alpha)\%$ Wald-type confidence interval is obtained by replacing $m$ with $\widehat{m}^{(-)}$ in the PPI variance formula, i.e.,

$$\widehat{\text{se}}_{\text{cf}}^2 = \frac{\widehat{\text{Var}}_N\{\widehat{m}^{(-)}(X)\}}{N} + \frac{\widehat{\text{Var}}_n\{Y - \widehat{m}^{(-)}(X)\}}{n},$$

and using the usual Wald interval $\widehat{\theta}_{\text{PPI}}^{\text{cf}} \pm z_{1-\alpha/2} \widehat{\text{se}}_{\text{cf}}$.

To understand how cross-prediction avoids label reuse, we rewrite the rectifier term in (6) as

$$\Delta_{\theta}^{\text{cf}} = \frac{1}{N} \sum_{i=1}^{N} \frac{\delta_i}{f} \left\{ Y_i - \widehat{m}^{(\kappa(i))}(X_i) \right\}, \quad (7)$$

where $\delta_i = I(i \in S)$ indicates whether unit $i$ is labeled.

In (7), each $Y_i$ is paired with an out-of-fold prediction $\widehat{m}^{(\kappa(i))}(X_i)$ from a model that did not use $(X_i, Y_i)$ in training. Conditional on the fold assignment $\kappa$ and the fitted models $\{\widehat{m}^{(k)}\}_{k=1}^{K}$, the summands $Y_i - \widehat{m}^{(\kappa(i))}(X_i)$ are independent. This conditional independence eliminates "*double-dipping*"—that is, using the same data for training and evaluation—yields an empirical mean with the usual influence-function behavior, and obviates Donsker-type entropy restrictions for asymptotic normality (Kennedy, 2024).

In Subsection 5.1, we distinguish our analysis by establishing the asymptotic properties of CF-PPI through empirical process theory—a framework not utilized in the original study by Zrnic & Candès (2024). While their work is primarily limited to establishing a CLT, it does not address the convergence rates of the out-of-fold predictor $\widehat{m}^{(-)}$ in (5), nor does it identify the precise conditions required to attain semiparametric efficiency. Our analysis fills these theoretical gaps, which are central to semi-supervised inference.

# 4. Machine-learning-assisted generalized entropy calibration

## 4.1. Bregman divergence

We introduce a GEC framework (Gneiting & Raftery, 2007; Kwon et al., 2025) that uses the Bregman divergence (Bregman, 1967) as the core distance, aligning with the Deville–Särndal calibration paradigm (Deville & Särndal, 1992; Deville et al., 1993). Unlike Deville & Särndal (1992), which measures discrepancy in a multiplicative ratio space, the Bregman approach operates in the additive weight space, yielding an exact projection interpretation, a Pythagorean identity, and a clean primal–dual symmetry via convex conjugacy (Amari & Nagaoka, 2000).

Let $G : \mathcal{V} \to \mathbb{R}$ be a prespecified generator that is strictly convex and twice continuously differentiable, with domain $\mathcal{V} \subset \mathbb{R}$ an open interval. Table 1 in the Appendix lists representative choices of the generator $G$. Write $g(u) = G'(u)$. For $u, v \in \mathcal{V}$, the scalar-valued Bregman divergence generated by $G$ is $D_G(u\|v) := G(u) - G(v) - g(v)(u - v)$. The quantity $D_G(u\|v)$ is the gap between $G(u)$ and the first–order Taylor approximation of $G$ at $v$; equivalently, it measures how far $G(u)$ lies above the tangent line to $G$ at $v$. By strict convexity, $D_G(u\|v) \geq 0$ with equality if and only if $u = v$.

We describe the use of Bregman divergence for weight calibration. Let $\omega = \{\omega_j : j \in S\}$ denote the calibration weights and $\omega^{(0)} = \{\omega_j^{(0)} : j \in S\}$ the baseline (design) weights. The former are the calibration targets, whereas the latter are typically known and fixed. Given a generator $G$ with derivative $g = G'$, we use the separable Bregman divergence anchored at $\omega^{(0)}$, defined by $D_G(\omega\|\omega^{(0)}) = \sum_{j \in S}\{G(\omega_j) - G(\omega_j^{(0)}) - g(\omega_j^{(0)})(\omega_j - \omega_j^{(0)})\}$. Recall that $S$ denotes the index set for the labeled sample, so the pair $(\omega_j, \omega_j^{(0)})$ is attached to each labeled unit. In particular, in our setting, the baseline weights are constant: $\omega_j^{(0)} = d_j = N/n$ for all $j \in S$, which is the inverse of the labeling fraction $f = n/N$.

## 4.2. Weight calibration for prediction-powered inference

We introduce a weight calibration framework for PPI, on which the MEC estimator is built. The central idea is to calibrate the rectifier in (6) using weights, and then construct an intermediate estimator with respect to a basis $h$:

$$\widehat{\theta}_{\mathrm{PPI}}^{\mathrm{wc},h} = \frac{1}{N}\sum_{i=1}^{N}\widehat{m}^{(-)}(X_i) + \frac{1}{N}\sum_{j \in S}\widehat{\omega}_j\left\{Y_j - \widehat{m}^{(-)}(X_j)\right\},$$

where $\widehat{m}^{(-)}$ is the out-of-fold predictor (5), and $\widehat{\omega} = \{\widehat{\omega}_j\}_{j \in S}$ are calibrated weights for the labeled units $S$, ob-

tained by solving the Bregman projection

$$\widehat{\omega} = \arg\min_{\omega \in (0,\infty)^n} D_G(\omega\|d) \qquad (8)$$

subject to the calibration constraint

$$\sum_{j \in S}\omega_j\, h(X_j) = \sum_{i=1}^{N}h(X_i), \qquad (9)$$

where $h : \mathcal{X} \subset \mathbb{R}^d \to \mathbb{R}^p$ denotes a user-chosen $p$-dimensional calibration basis, and $z_j := h(X_j) = (h_1(X_j), \ldots, h_p(X_j))^\top$. The calibrated weights $\widehat{\omega}$ are computed using the dual Newton solver, as described in Subsection C.1 in the Appendix.

We note that the rectifier term of $\widehat{\theta}_{\mathrm{PPI}}^{\mathrm{wc},h}$, denoted

$$\Delta_\theta^{\mathrm{wc},h} := \frac{1}{N}\sum_{j \in S}\widehat{\omega}_j\{Y_j - \widehat{m}^{(-)}(X_j)\} \qquad (10)$$

inherits the avoidance of label reuse via sample-splitting as in the rectifier term (7) of CF–PPI, provided that the calibrated weight $\widehat{\omega}_j$ is independent of $Y_j$ given $X_j$.

The proposed framework is a weight-calibrated generalization of CF-PPI. To see this, observe that the estimator $\widehat{\theta}_{\mathrm{PPI}}^{\mathrm{wc},h}$ with respect to basis $h$ reduces to the CF-PPI estimator $\widehat{\theta}_{\mathrm{PPI}}^{\mathrm{cf}}$ in (6) when $h$ is chosen as the constant function $h(x) = 1$. In this case, the calibration constraint (9) simplifies to $\sum_{j \in S}\omega_j\, h(X_j) = \sum_{j \in S}\omega_j = \sum_{i=1}^{N}h(X_i) = N$. Given baseline weights $d_j = N/n$ and any strictly convex generator $G$, the Bregman projection is equivalent to maximizing the Lagrangian $\mathcal{L}(\omega, \lambda) = -\sum_{j \in S}\{G(\omega_j) - G(d_j) - g(d_j)(\omega_j - d_j)\} + \lambda(\sum_{j \in S}\omega_j - N)$. The first-order condition $-g(\omega_j) + g(d_j) + \lambda = 0$ implies that $g(\omega_j)$ is constant across $j$. Since $g$ is strictly increasing, all $\omega_j$ must be equal; combining this with $\sum_{j \in S}\omega_j = N$ yields $\omega_j = N/n = d_j$ for all $j \in S$. Plugging these weights into $\widehat{\theta}_{\mathrm{PPI}}^{\mathrm{wc},h}$ exactly recovers the CF-PPI estimator.

Consequently, we recommend including the intercept 1 as a component of the basis $h$ *by default* under the proposed weight calibration framework for PPI. This inclusion ensures that the proposed framework inherits the desirable property of avoiding label reuse from CF-PPI, automatically resolving the primary limitation of vanilla PPI (see **I. Label reuse** in Subsection 3.1). Including an intercept is also standard practice in survey sampling for Horvitz–Thompson-type rectification (Horvitz & Thompson, 1952; Deville & Särndal, 1992), as adopted in $\widehat{\theta}_{\mathrm{PPI}}^{\mathrm{wc},h}$.

One commonly used calibration basis $h$ is a moment-feature basis $h(X_j) = (1, X_{j1}, \ldots, X_{jd}, X_{j1}^2, \ldots, X_{j\ell}X_{jk}) \in \mathbb{R}^p$ (for selected $\ell < k$), which enforces the equality of selected moments-intercept, first- and second-order moments, and

some interactions-between the labeled set and the population, thus shrinking the discrepancy $N^{-1} \sum_{i=1}^{N} h(X_i) - n^{-1} \sum_{j \in S} h(X_j)$, as typically used in survey sampling (Deville & Särndal, 1992; Breidt & Opsomer, 2017; Haziza & Beaumont, 2017). However, in semi-supervised inference, the analyst typically does not know *a priori* which moments are most relevant for bias reduction; specifying a large, high-dimensional basis can be impractical or unstable (e.g., inducing high-variance weights), which motivates ML predictor-based basis, as discussed in the next subsection.

### 4.3. Machine-learning-assisted generalized entropy calibration estimator

We construct the MEC estimator using the weight calibration framework for PPI proposed in the previous subsection. We adopt an out-of-fold predictor basis with an intercept (hereafter, simply called the "predictor basis"):

$$h(X_j) = \left(1, \widehat{m}^{(-)}(X_j)\right) \in \mathbb{R}^2. \qquad (11)$$

The calibration constraints (9) thus take the form $\sum_{j \in S} \omega_j = N$ and $\sum_{j \in S} \omega_j \widehat{m}^{(-)}(X_j) = \sum_{i=1}^{N} \widehat{m}^{(-)}(X_i)$. The first constraint ensures the avoidance of label reuse, as discussed in the previous subsection. The second constraint aligns the weighted predictor total in the labeled sample with the full population total; this explicitly addresses **II. Efficiency shortfall** in Subsection 3.1, which we theoretically establish in Section 5.

Recall that $\kappa(j)$ denotes the fold of unit $j$ and $\widehat{m}^{(\kappa(j))}$ the predictor trained without fold $\kappa(j)$, thus, the second constraint after calibration can be written as

$$\sum_{j \in S} \widehat{\omega}_j \, \widehat{m}^{(\kappa(j))}(X_j) = \sum_{i=1}^{N} \widehat{m}^{\star}(X_i),$$

where the summand on the left-hand side satisfies

$$\begin{aligned}
\widehat{\omega}_j \, \widehat{m}^{(\kappa(j))}(X_j) &= \widehat{\omega}_j(\widehat{\lambda}) \, \widehat{m}^{(\kappa(j))}(X_j) \\
&= g^{-1}\!\left(g(d_j) + z_j^\top \widehat{\lambda}\right) \widehat{m}^{(\kappa(j))}(X_j) \\
&= g^{-1}\!\left(g(d_j) + \widehat{\lambda}_1 + \widehat{\lambda}_2 \, \widehat{m}^{(\kappa(j))}(X_j)\right) \widehat{m}^{(\kappa(j))}(X_j),
\end{aligned}$$

which depends on the covariate $X_j$ but does not involve $Y_j$ directly through $\widehat{m}^{(\kappa(j))}(\cdot)$ for each $j \in S$. The last equality follows from the calibration map $\omega_j(\lambda) = g^{-1}\!\left(g(d_j) + z_j^\top \lambda\right)$ and the predictor basis $z_j = h(X_j) = (1, \widehat{m}^{(\kappa(j))}(X_j))$ (11). $\lambda$ denotes the dual variable (i.e., Lagrange multipliers) arising in the optimization; see Section C.1 of the Appendix for details.

In summary, the out-of-fold predictor $\widehat{m}^{(-)}$ in (5) is used twice: (i) to form the difference $Y_j - \widehat{m}^{(-)}(X_j)$ in the rectifier term $\Delta_\theta^{\mathrm{wc},h}$ (10), and (ii) to define the predictor basis $h = (1, \widehat{m}^{(-)})$ (11) for computing the calibrated weights $\widehat{\omega}_j$.

This safeguard–*double label-reuse avoidance*–is essential to MEC's asymptotic validity and stable variance estimation developed in Subsection 5.2.

The MEC estimator is then expressed as

$$\widehat{\theta}_{\mathrm{MEC}} := \widehat{\theta}_{\mathrm{PPI}}^{\mathrm{wc},h=(1,\widehat{m}^{(-)})} = \frac{1}{N} \sum_{j \in S} \widehat{\omega}_j Y_j. \qquad (12)$$

The immediate benefit of the MEC estimator is computational stability. It intrinsically promotes weight stability because the basis is only two–dimensional ($p = 2$), regardless of the covariate dimension $d$, the labeled sample size $n$, or the population size $N$. Consequently, the dual Newton solver (Section C.1 in the Appendix) operates in a very low–dimensional space and is fast and numerically stable. Each iteration forms the $p \times p$ Hessian $H(\lambda) = \nabla^2 \ell(\lambda) = Z^\top \mathrm{diag}\!\left(1/g'(\omega)\right) Z$, which costs $O(np^2)$ to assemble and $O(p^3)$ to solve for the Newton step; with $p = 2$, this cost is negligible even for large $n$.

### 4.4. Dual expression of the MEC estimator

We show that the MEC estimator $\widehat{\theta}_{\mathrm{MEC}}$ (12) admits a *dual generalized regression estimator (GREG) representation* (Särndal, 1980; Deville & Särndal, 1992; Wu & Sitter, 2001); that is, it can be written as a prediction term plus a design-weighted residual correction.

**Theorem 4.1.** *Assume the conditions of the basic setup in Section 2. Consider the MEC estimator* (12)

$$\widehat{\theta}_{\mathrm{MEC}} = \frac{1}{N} \sum_{j \in S} \widehat{\omega}_j \, Y_j.$$

*Consider the GREG estimator*

$$\widehat{\theta}_{\mathrm{GREG}} = \frac{1}{N} \sum_{i=1}^{N} \widehat{y}_i + \frac{1}{N} \sum_{j \in S} d_j \, (Y_j - \widehat{y}_j) \qquad (13)$$

*where* $\widehat{y}_i = \widehat{\beta}^\top z_i = \widehat{\beta}_0 + \widehat{\beta}_1 \, \widehat{m}^{(-)}(X_i)$, *obtained by a weighted least squares regression of* $Y_j$ *on* $z_j$ *with weights* $q_j$, *so that* $\widehat{\beta}$ *satisfies the weighted normal equations*

$$\left(\sum_{j \in S} q_j \, z_j z_j^\top\right) \widehat{\beta} = \sum_{j \in S} q_j \, z_j Y_j, \qquad q_j := \frac{1}{g'(d_j)}. \qquad (14)$$

*Then* $\widehat{\theta}_{\mathrm{MEC}} = \widehat{\theta}_{\mathrm{GREG}} + R_N$, *with* $R_N = \mathcal{O}_{\mathbb{P}}\!\left(\|\widehat{\lambda}\|^2\right)$.

*In particular, if* $\|\widehat{\lambda}\| = \mathcal{O}_{\mathbb{P}}(n^{-1/2})$ *and standard moment conditions hold, then* $R_N = o_{\mathbb{P}}(N^{-1/2})$, *so the representation is asymptotically exact. Moreover, when $G$ is quadratic, we have the exact identity* $\widehat{\theta}_{\mathrm{MEC}} = \widehat{\theta}_{\mathrm{GREG}}$ *(i.e., $R_N = 0$).*

Theorem 4.1 reveals that the MEC estimator $\widehat{\theta}_{\mathrm{MEC}}$ in (12) is not merely a GEC-type estimator, but rather a

geometric projection estimator that admits an efficient regression representation. In particular, under our setting, the GREG estimator $\widehat{\theta}_{\text{GREG}}$ in (13) simplifies to $\widehat{\theta}_{\text{GREG}} = (1/N) \sum_{i=1}^{N} \widehat{\beta}_1 \widehat{m}^{(-)}(X_i) + (1/n) \sum_{j \in S} (Y_j - \widehat{\beta}_1 \widehat{m}^{(-)}(X_j))$, where the intercept $\widehat{\beta}_0$ cancels out. Furthermore, solving the normal equations (14) yields the familiar covariance–variance form for the slope $\widehat{\beta}_1 = \sum_{j \in S} q_j (m_j - \bar{m}_q)(Y_j - \bar{Y}_q) / \sum_{j \in S} q_j (m_j - \bar{m}_q)^2 = \text{Cov}(\widehat{m}^{(-)}(X), Y) / \text{Var}(\widehat{m}^{(-)}(X))$, where $m_j = \widehat{m}^{(-)}(X_j)$, $\bar{m}_q = \sum_{j \in S} q_j m_j / \sum_{j \in S} q_j$, and $\bar{Y}_q = \sum_{j \in S} q_j Y_j / \sum_{j \in S} q_j$. (The second equality holds because $q_j = 1/g'(d_j)$ is constant in $j$, so the weights cancel from both the numerator and the denominator.) This result is asymptotically equivalent to the optimal-tuning result for PPI++ (see Example 6.1 in (Angelopoulos et al., 2024)). MEC generalizes PPI++ in a dual sense and the two are asymptotically equivalent when the generator is quadratic; thus, MEC includes PPI++ as a special case.

Finally, we construct the $100(1 - \alpha)\%$ Wald-type confidence interval for the population mean $\theta_0$ as $\widehat{\theta}_{\text{MEC}} \pm z_{1-\alpha/2} \widehat{\text{se}}_{\text{MEC}}$, where the standard error uses the GREG-style decomposition implied by the MEC–GREG duality

$$\widehat{\text{se}}_{\text{MEC}}^2 = \frac{1}{N} \widehat{\text{Var}}_U(\widehat{\beta}_1 \widehat{m}_U) + \frac{1}{n} \widehat{\text{Var}}_S(Y - \widehat{\beta}_1 \widehat{m}_S).$$

Here, $\widehat{m}_S = \{\widehat{m}^{(-)}(X_j) : j \in S\}$ and $\widehat{m}_U = \{\widehat{m}^{(-)}(X_i) : i \in U\}$ denote the out-of-fold predictions for the labeled and unlabeled units, respectively.

Additionally, Theorem 4.1 implies that MEC generalizes the classical estimator by adaptively leveraging labeled data; if the cross-prediction carries no linear signal for $Y$ on $S$ (i.e., $\text{Cov}(\widehat{m}^{(-)}(X), Y) = 0$), then $\widehat{\beta}_1 = 0$, so $\widehat{\theta}_{\text{MEC}} = (1/n) \sum_{j \in S} Y_j + R_N$ (with $R_N = o_p(n^{-1/2})$ under standard conditions and $\widehat{\text{se}}_{\text{MEC}}^2 = (1/n)\widehat{\text{Var}}_S(Y)$. This is intuitively desirable: if the predictor carries little information for the labeled outcomes, there is little to borrow from the unlabeled data, so a desirable estimator should shrink to the classical estimator. Thus, MEC induces data-adaptive post-prediction inference (Miao et al., 2025).

## 5. Statistical properties

### 5.1. Asymptotic theory of cross-fitted PPI estimator

We first derive sufficient conditions for the asymptotic properties of the CF–PPI estimator $\widehat{\theta}_{\text{PPI}}^{\text{cf}}$ given in (6).

**Theorem 5.1.** *Assume the conditions of the basic setup in Section 2. Let $\widehat{m}^{(-)}$ be the out-of-fold predictor (5), and consider CF–PPI estimation for the mean*

$$\widehat{\theta}_{\text{PPI}}^{\text{cf}} = \frac{1}{N} \sum_{i=1}^{N} \widehat{m}^{(-)}(X_i) + \frac{1}{n} \sum_{j \in S} \{Y_j - \widehat{m}^{(-)}(X_j)\}.$$

*Then:*

**(i) Consistency.** *If the out-of-fold error is stochastically bounded in $L_2(P_X)$,*

$$\|\widehat{m}^{(-)} - m_0\|_{L_2(P_X)} = O_p(1),$$

*then $\widehat{\theta}_{\text{PPI}}^{\text{cf}} \xrightarrow{p} \theta_0$.*

**(ii) Asymptotic normality.** *If, in addition, the cross-fit predictor is $L_2(P_X)$-consistent,*

$$\|\widehat{m}^{(-)} - m_0\|_{L_2(P_X)} = o_p(1),$$

*then $\sqrt{N}(\widehat{\theta}_{\text{PPI}}^{\text{cf}} - \theta_0) \xrightarrow{d} \mathcal{N}(0, \sigma_f^2)$ with*

$$\sigma_f^2 = \text{Var}(m_0(X)) + \frac{1}{f} \text{Var}(Y - m_0(X)). \quad (15)$$

We briefly compare Theorem 5.1 with the prior analysis of Zrnic & Candès (2024). Their work establishes a CLT for CF–PPI for the mean under stability conditions on the fold-specific learners (see their Assumptions 1 and 2). While these conditions capture algorithmic stability, they are not standard within empirical process theory and do *not* directly characterize the convergence behavior of the out-of-fold predictor or its implications for semiparametric efficiency.

By contrast, our analysis of the semi-supervised mean proceeds under milder, classical assumptions. In particular, consistency of CF–PPI follows from the minimal requirement of $L_2$ stochastic boundedness, $\|\widehat{m}^{(-)} - m_0\|_{L_2(P_X)} = O_p(1)$, while asymptotic normality is obtained under mere $L_2$ consistency, $\|\widehat{m}^{(-)} - m_0\|_{L_2(P_X)} = o_p(1)$. These conditions are natural in modern ML theory and allow us to explicitly characterize the efficiency properties of CF–PPI through the convergence rate of the out-of-fold predictor.

In Theorem 5.1, we state the asymptotic distribution on the $\sqrt{N}$ scale. Equivalently, since $n/N \to f \in (0, 1)$, the same result can be written on the $\sqrt{n}$ scale as $\sqrt{n}(\widehat{\theta}_{\text{PPI}}^{\text{cf}} - \theta_0) \xrightarrow{d} \mathcal{N}(0, f \text{Var}\{m_0(X)\} + \text{Var}\{Y - m_0(X)\})$. This expression separates the two sources of uncertainty. The first term, $f \text{Var}\{m_0(X)\}$, comes from averaging the prediction component over the unlabeled covariates, whereas the second term, $\text{Var}\{Y - m_0(X)\}$, comes from the residual variation in the labeled sample. When $f$ is small, meaning that the unlabeled sample is much larger than the labeled sample, the first component is small, so most remaining uncertainty comes from the labeled residuals. Thus, unlabeled covariates improve efficiency by reducing the limiting variance, but they do not remove the $\sqrt{n}$-rate dependence on the number of labeled observations.

### 5.2. Asymptotic theory of the MEC estimator

We now present the main theoretical result of this paper.

**Theorem 5.2.** *Assume the conditions of the basic setup in Section 2, and consider the MEC estimator* (12)

$$\widehat{\theta}_{\mathrm{MEC}} = \frac{1}{N} \sum_{j \in S} \widehat{\omega}_j \, Y_j.$$

*Define the calibration subspace*

$$W := \mathrm{span}\{h\} = \mathrm{span}\{1, \widehat{m}^{(-)}(\cdot)\}$$
$$= \{\, a + b\, \widehat{m}^{(-)}(\cdot) : a, b \in \mathbb{R} \,\} \subset L_2(P_X),$$

*and let* $\Pi_W$ *denote the* $L_2(P_X)$ *projection onto* $W$, $\Pi_W m_0 = \arg\min_{v \in W} \mathbb{E}\big[(m_0(X) - v(X))^2\big]$. *Define the projection error* $m_{0,\perp} := m_0 - \Pi_W m_0$, *representing the component of the true regression function orthogonal to the calibration subspace* $W$.

**Assumptions**

*A1.* $\mathbb{E}\|h(X)\|^2 < \infty$.

*A2. Calibrated weights satisfy the symmetric Bregman bound* $D_G(\widehat{\omega}\|d) + D_G(d\|\widehat{\omega}) = O_p(1)$.

*A3. With* $A_n := (1/n) \sum_{j \in S} q_j z_j z_j^\top$, $q_j := 1/g'(d_j)$, *one has* $A_n \xrightarrow{P} \Sigma_q \succ 0$, *where "* $\succ 0$ *" denotes positive definiteness.*

*A4. Let* $\widehat{\lambda}$ *be the dual optimizer of the calibration program with* $\|\widehat{\lambda}\| = O_p(n^{-1/2})$.

*Then:*

**(i) Consistency.** *If A1–A2 hold and the projection error is stochastically bounded in* $L_2(P_X)$,

$$\|m_{0,\perp}\|_{L_2(P_X)} = \|m_0 - \Pi_W m_0\|_{L_2(P_X)} = O_p(1),$$

*then* $\widehat{\theta}_{\mathrm{MEC}} \xrightarrow{P} \theta_0$.

**(ii) Asymptotic normality.** *If A1–A4 hold and the projection error is* $L_2(P_X)$-*consistent,*

$$\|m_{0,\perp}\|_{L_2(P_X)} = \|m_0 - \Pi_W m_0\|_{L_2(P_X)} = o_P(1),$$

*then* $\sqrt{N}(\widehat{\theta}_{\mathrm{MEC}} - \theta_0) \xrightarrow{d} \mathcal{N}(0, \sigma_f^2)$ *with*

$$\sigma_f^2 = \mathrm{Var}\big(m_0(X)\big) + \frac{1}{f}\, \mathrm{Var}\big(Y - m_0(X)\big).$$

We state regularity conditions A1–A4 for establishing consistency and asymptotic normality of the MEC estimator. Assumption A1 guarantees square-integrability of the calibration moments and the plug-in term. Assumption A2

prevents explosive or overly concentrated weights by keeping the calibrated weights $\widehat{\omega}$ in an $O_p(1)$ Bregman neighborhood of the baseline $d$, which justifies first-order linearization of the calibration map. Assumption A3 is an identifiability/curvature condition for the two-dimensional predictor basis: the $q_j$-weighted Gram matrix converges to a positive-definite limit, ensuring a well-conditioned Newton step for the dual and a valid quadratic expansion. Assumption A4 localizes the dual solution at the root-$n$ scale so that the perturbation $\widehat{\omega} - d$ is of order $n^{-1/2}$ along the feasible manifold, yielding an influence-function expansion and the stated CLT. Together, A1–A4 are standard regularity conditions for Bregman-projection calibration (Gneiting & Raftery, 2007; Kwon et al., 2025).

An important advantage of MEC over CF-PPI is that it relaxes the predictor–related conditions required for consistency and asymptotic normality. To see this, note that the following inequality holds by orthogonal projection:

$$\|m_{0,\perp}\|_{L_2(P_X)} \leq \|\widehat{m}^{(-)} - m_0\|_{L_2(P_X)}, \qquad (16)$$

since $W = \mathrm{span}\{h\} = \mathrm{span}\{1, \widehat{m}^{(-)}\}$. Under A1–A2 (moment conditions and weight stability), the MEC estimator is consistent whenever the projection error is stochastically bounded in $L_2(P_X)$, that is, $\|m_{0,\perp}\|_{L_2(P_X)} = O_p(1)$. Under A1–A4 (adding a weighted–Gram limit and a root-$n$ dual solution), asymptotic normality holds if the projection error vanishes, $\|m_{0,\perp}\|_{L_2(P_X)} = o_p(1)$, yielding the same limit law as CF–PPI with the variance of efficient influence function (EIF) (see Equation (25) in the Appendix). By (16), these projection-based requirements are weaker than conditions stated directly on the prediction error, showing how MEC relaxes the assumptions needed for large-sample validity relative to CF–PPI.

We interpret the benefit of MEC through the lens of orthogonality learning (Foster & Syrgkanis, 2023; Chernozhukov et al., 2018; Mackey et al., 2018). In Section C in the Appendix, for any fixed predictor $m$, we derived the misspecification-driven inflation $(N/n - 1)\mathbb{E}[(m_0(X) - m(X))^2]$. Under MEC, the variance inflation contracts to

$$\text{Variance Inflation (MEC)} = \left(\frac{N}{n} - 1\right) \mathbb{E}\big[m_{0,\perp}(X)^2\big].$$

Thus, MEC removes all components of $m_0$ captured by the ML predictor $\widehat{m}^{(-)}$; only the orthogonal remainder can inflate the variance. In particular, if $m_0 \in W$ (i.e., $m_0 = a + b\, \widehat{m}^{(-)}$), the variance inflation under MEC is exactly zero—demonstrating robustness of weight calibration to misspecification up to an *affine rescaling of* $\widehat{m}^{(-)}$.

# 6. Simulation experiments

We numerically illustrate the improved performance of our proposed MEC (12) against vanilla PPI (4) and CF–PPI (6).

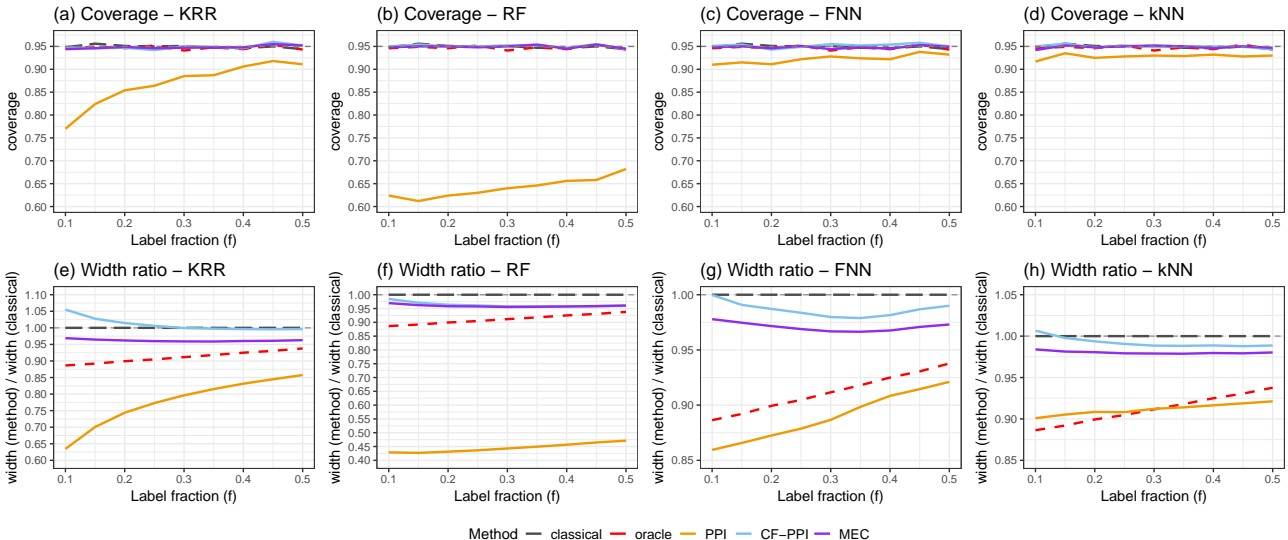

*Figure 1.* Coverage and width ratios of 95% confidence intervals across label fractions $f$ for four ML predictors. Each column corresponds to one predictor—KRR, RF, FNN, and kNN. MEC (quadratic generator) attains near-nominal coverage and the narrowest valid intervals, consistently improving efficiency over CF–PPI. Vanilla PPI undercovers, especially at small $f$. Classical and oracle baselines are shown for reference. (See Figure 5 for the results with MEC using other generators; MEC performance is robust to the choice of generator.)

Implementations follow Sections 3 and 4. Following (Angelopoulos et al., 2023; Zrnic & Candès, 2024; Miao et al., 2025), we report (i) empirical coverage of nominal 95% Wald confidence intervals (CIs) and (ii) a width ratio,

$$\mathrm{WR}_{\mathrm{method}}(f) = \frac{\mathrm{CI\ width}_{\mathrm{method}}(f)}{\mathrm{CI\ width}_{\mathrm{classical}}(f)},$$

where the classical estimator is the label-only sample mean with its usual standard error, and $f = n/N$ denotes the labeled fraction. We run $R = 2000$ Monte Carlo replications.

A desirable method should (i) achieve coverage close to 95% uniformly over $f$ and (ii) satisfy $\mathrm{WR}_{\mathrm{oracle}}(f) \leq \mathrm{WR}_{\mathrm{method}}(f) < 1$, indicating efficiency gains from using unlabeled covariates relative to the classical estimator. Here, $\mathrm{WR}_{\mathrm{oracle}}(f)$ is the width ratio of the oracle PPI estimator relative to the classical estimator; the oracle PPI plugs in the true regression function $m_0$ and attains the semiparametric efficiency bound (see Section C in the Appendix). The best-performing method is the one that satisfies these two conditions and has $\mathrm{WR}_{\mathrm{method}}(f)$ as close as possible to $\mathrm{WR}_{\mathrm{oracle}}(f)$, indicating near-attainment of the semiparametric efficiency bound. Methods with coverage far from 95% are invalid, regardless of the width ratio.

For MEC and CF–PPI, we use four learners with 5-fold cross-prediction ($K = 5$): kernel ridge regression (KRR; Gaussian kernel; (Schölkopf & Smola, 2002)), random forest (RF; (Breiman, 2001)), a feedforward neural network (FNN; (Goodfellow et al., 2016)), and $k$-nearest neighbors

(kNN; (Cover & Hart, 1967)). For vanilla PPI, we use the same learners with single-fit training (no sample splitting). All predictors across MEC, CF–PPI, and vanilla PPI follow the same hyperparameter-tuning protocol to ensure a fair comparison. See Subsection G.1 in the Appendix for details.

**Synthetic data.** We draw covariates for an unlabeled set of size $N = 1000$ and a labeled set $S$ of size $n = fN$ with $f \in [0.1, 0.5]$ as i.i.d. samples with $d = 10$ features; $x_1, \ldots, x_d \overset{\text{i.i.d.}}{\sim} \mathcal{N}(0, 1)$. Outcomes are observed only for the labeled sample: for $j \in S$, $Y_j = m_0(X_j) + \varepsilon_j$ with $\varepsilon_j \sim \mathcal{N}(0, \sigma_y^2)$ and $\sigma_y = 5$. Following Li & Fu (2017); Ray & Szabó (2019), we use the true regression function $m_0(x) = \sum_{k=1}^{d} g_k(x_k)$ with $g_1(x) = e^{-x}$, $g_2(x) = x^2$, $g_3(x) = x$, $g_4(x) = \mathbb{I}\{x > 0\}$, $g_5(x) = \cos x$, and $g_k(x) \equiv 0$ for $k = 6, \ldots, d$. Thus, only the first five coordinates of $X$ influence $Y$; the remaining features are noise.

**Results.** Figure 1 reports coverage (panels (a)–(d)) and width ratios (panels (e)–(h)). We plot MEC with the quadratic generator only (see Subsection G.2 in the Appendix for other generators, which exhibit similar qualitative behavior). MEC attains near-nominal 95% coverage across all label fractions $f$ while delivering tighter intervals than CF–PPI. Vanilla PPI often undercovers, especially when $f$ is small, reflecting overfitting due to label reuse. CF–PPI generally restores validity but can yield width ratios above 1 at $f = 0.1$ with KRR and kNN, indicating intervals wider than the classical estimator (i.e., no efficiency gain). Across

predictors, MEC is most efficient with KRR or RF; this is expected in our setting, where FNN and kNN are configured as relatively weak, fast learners. Overall, MEC exhibits the most stable and efficient performance across $f$ and learners compared with PPI and CF–PPI. Additional simulation experiments in the Appendix show similar results, further demonstrating the superior performance of MEC.

# 7. Extensions and practical considerations

**Choice of calibration basis.** The choice of calibration basis $h$ determines both the resulting estimator and its robustness properties. In this paper, we mainly focus on the two-dimensional out-of-fold predictor basis $h(x) = (1, \widehat{m}^{(-)}(x))$ in (11). One may also consider richer bases, such as $h(x) = (1, \widehat{m}^{(-)}(x), \widehat{m}^{(-)}(x)^2, \widehat{m}^{(-)}(x)^3)$, or bases formed from multiple out-of-fold ML predictors, such as $h(x) = (1, \widehat{m}_1^{(-)}(x), \widehat{m}_2^{(-)}(x), \widehat{m}_3^{(-)}(x))$. Technically, these choices enlarge the calibration subspace from $W = \mathrm{span}\{1, \widehat{m}^{(-)}\}$ to richer spaces, such as $\mathrm{span}\{1, \widehat{m}^{(-)}, (\widehat{m}^{(-)})^2, (\widehat{m}^{(-)})^3\}$ for a single predictor, or $\mathrm{span}\{1, \widehat{m}_1^{(-)}, \widehat{m}_2^{(-)}, \widehat{m}_3^{(-)}\}$ for multiple predictors. Thus, the affine-robustness argument for the two-dimensional basis extends, in the first case, to robustness against low-order nonlinear transformations of a single predictor, and in the second case, to calibration over several candidate prediction rules. Additionally, the latter choice is naturally connected to ensemble learning (Dietterich, 2000) and Super Learner (Van der Laan et al., 2007), since the calibration constraints can balance several candidate prediction rules simultaneously, although this comes at the cost of increased computation and potentially less stable calibrated weights.

**MEC under weighted estimating equations.** The MEC framework can be formulated beyond semi-supervised mean estimation as a general calibration device for weighted estimating equations; see Section B of the Appendix. Roughly speaking, when an estimator is defined through baseline weights in a weighted estimating equation, MEC updates these weights through a Bregman projection so that the weighted sample satisfies calibration constraints based on outcome-relevant ML predictions, while preserving the original estimating-equation structure. This perspective highlights the broad applicability of MEC to settings such as weighted regression, missing-data estimation (Robins et al., 1994; Lipsitz et al., 1999; Qi et al., 2005), causal effect estimation (Hernán et al., 2000; Robins et al., 2000), and weighted Cox regression (Lin & Wei, 1989; Binder, 1992).

This contrasts with EIF-based or orthogonal-score methods, such as targeted maximum likelihood estimation (TMLE) (van der Laan & Rubin, 2006; Van der Laan & Rose, 2011) and double/debiased machine learning (DML) (Chernozhukov et al., 2018), which typically require deriving

an EIF or constructing an orthogonal score for the specific problem at the outset. Although MEC can be analyzed from an EIF-based perspective, as shown in Subsection 5.2, its implementation does not require deriving the EIF in advance.

**Settings where MEC may not yield efficiency gains.** The efficiency gains from MEC are not automatic for every target parameter. MEC is most useful when the auxiliary covariates contain information about the outcome- or score-relevant variation in the estimating equation. In semi-supervised mean estimation, this condition is natural because $X$ helps predict $Y$ through the regression function $m_0(x) = \mathbb{E}[Y \mid X = x]$, so the unlabeled covariates can be used to reduce variability. In contrast, for a target functional whose estimating equation has little or no component predictable from $X$, the unlabeled covariate distribution carries limited information for improving efficiency. In such settings, calibration has little room to reduce variance.

# 8. Conclusion

We develop MEC, a novel variant of PPI for semi-supervised mean estimation that extends PPI through principled weight calibration using a predictor-based basis. MEC reweights labeled residuals via a Bregman projection, yielding robustness to predictor misspecification up to affine transformations while maintaining numerical stability through a low-dimensional dual Newton solver. Under mild regularity conditions, MEC attains the semiparametric efficiency bound when the projection error vanishes. Compared with CF–PPI, MEC achieves efficiency under weaker assumptions by replacing conditions on raw prediction error with weaker projection-error requirements. Empirically, MEC maintains near-nominal coverage and produces tighter confidence intervals than both the vanilla PPI and CF–PPI across a range of data-generating scenarios, with a real-data application further supporting its practical utility.

Promising directions for future work include extensions to broader causal and semiparametric parameter-estimation problems, as well as deeper connections to orthogonal statistical learning and generalized moment-equation frameworks. Although we outline MEC in a general weighted estimating-equation setting in the Appendix, together with two additional examples illustrating its applicability, rigorous theoretical guarantees in such broader settings require further investigation. Among several important directions, a key open problem is to clarify the theoretical advantages of MEC across different estimands and outcome types. In particular, it remains to determine when MEC can attain semiparametric efficiency under weaker assumptions, when it provides relative efficiency gains over benchmark methods, and whether such guarantees must be established in a problem-specific manner.

## Impact Statement

This work advances reliable statistical inference under scarce labeling—a pervasive challenge in modern ML systems. By integrating machine-learning predictors with principled calibration, MEC enables valid confidence intervals even when models are misspecified. We foresee broader impact in enhancing reproducibility and trustworthiness of machine-learning-based decision systems.

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

# A. Setup and notation

## A.1. Asymptotic notation

Unless stated otherwise, limits are taken as the sample size $N \to \infty$ (and $n = n(N) \to \infty$ when present). For deterministic sequences $a_N, b_N > 0$, we write $a_N = O(b_N)$ if $\sup_N a_N/b_N < \infty$, and $a_N = o(b_N)$ if $a_N/b_N \to 0$. For random quantities $X_N$ and positive scales $a_N$, $X_N = O_p(a_N)$ means $X_N/a_N$ is tight (bounded in probability), and $X_N = o_p(a_N)$ means $X_N/a_N \xrightarrow{p} 0$. We use $\xrightarrow{p}$ and $\xrightarrow{d}$ to denote convergence in probability and in distribution, respectively.

## A.2. Empirical-process notation

We adopt standard empirical-process notation (Geer, 2000; Pollard, 1989; Kennedy, 2016). Let $P$ denote the super–population distribution of $X$. Define the empirical measures

$$P_N := \frac{1}{N} \sum_{i=1}^{N} \delta_{X_i},$$

$$P_n := \frac{1}{n} \sum_{j \in S} \delta_{X_j},$$

where $\delta_z$ is the Dirac measure at $z$ and $S$ is an index set of labeled data with $|S| = n$. For any measurable $h$,

$$P_N h := \int h \, dP_N = \frac{1}{N} \sum_{i=1}^{N} h(X_i),$$

$$P_n h := \int h \, dP_n = \frac{1}{n} \sum_{j \in S} h(X_j),$$

$$P h := \int h \, dP = \mathbb{E}_P[h(X)].$$

When a function depends on both covariates and outcomes, we use the same notation with the obvious modification; e.g.,

$$P_n h = \frac{1}{n} \sum_{j \in S} h(X_j, Y_j),$$

$$P h = \mathbb{E}_P[h(X, Y)].$$

For example, if $h(x) = \mathbf{1}\{x \in A\}$, then $P_n h$ is the empirical probability of the event $A$ within the labeled subset $S$. By the law of large numbers, $P_N h \to P h$ and (in our setting, $n/N \to f \in (0,1)$) $P_n h \to P h$ in probability. This notation streamlines decompositions of estimators and estimands and the statement of convergence results.

We define the weighted empirical measure

$$P_n^{\omega} h \; := \; \frac{1}{N} \sum_{j \in S} \omega_j \, h(X_j).$$

By definition, if $\omega_j = d_j \equiv N/n$ (the base weights), then

$$P_n^d h \; = \; \frac{1}{N} \sum_{j \in S} d_j \, h(X_j) \; = \; \frac{1}{n} \sum_{j \in S} h(X_j) \; = \; P_n h.$$

Thus, in our setting, $P_n^d = P_n$.

# B. Machine-learning-assisted generalized entropy calibration

## B.1. General framework

We describe MEC as a calibration device for weighted estimating equations (Robins et al., 1994; Lipsitz et al., 1999; Qi et al., 2005). The semi-supervised mean-estimation problem studied in the main text is one special case of this framework. This broader formulation clarifies how MEC differs from other approaches that use flexible ML predictions as intermediate inputs for statistically principled estimation, including TMLE (Van der Laan & Rose, 2011; 2018), DML (Chernozhukov et al., 2018), and prediction-assisted adjustment methods (Guo & Basse, 2023; Lin, 2013; Cohen & Fogarty, 2024).

**Conceptual workflow.** Figure 2 describes the conceptual workflow for constructing the MEC estimator; the notation in the figure will be introduced shortly. Starting from a well-defined weighted estimating equation for the target parameter, MEC first specifies baseline source-set weights $\{\widehat{d}_i : i \in \mathcal{I}_S\}$. These weights may be design weights (Horvitz & Thompson, 1952), inverse-probability weights (Seaman & White, 2013; Li et al., 2018), generalizability or transportability weights (Stuart et al., 2011; Pressler & Kaizar, 2013), or normalized constant weights corresponding to unweighted estimating equations. In causal settings, this step often corresponds to propensity-score modeling (Rosenbaum & Rubin, 1983). Separately, MEC constructs a cross-fitted calibration basis $\widehat{h}_i$ using ML predictions. In causal settings, it is conceptually related to prognostic-score modeling (Hansen, 2008). Given a convex generator, MEC then computes a minimal Bregman perturbation of the baseline weights, following generalized entropy calibration (Kwon et al., 2025), to obtain MEC-updated weights $\{\widehat{w}_i : i \in \mathcal{I}_S\}$ that balance the calibration basis between the weighted source sample and the target sample. The MEC estimator $\widehat{\theta}_{\mathrm{MEC}}$ is obtained by plugging these updated weights into the original weighted estimating equation. Under suitable calibration regularity, this update preserves the target population estimating equation while modifying its finite-sample weighted approximation in outcome-relevant directions, which may improve efficiency.

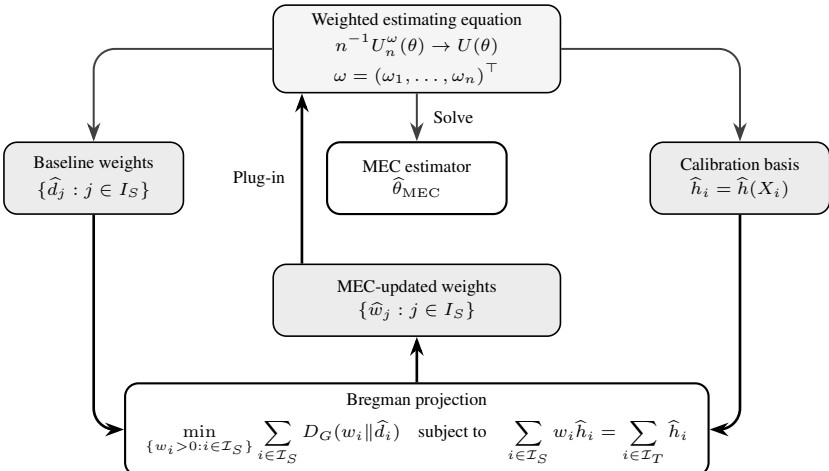

Figure 2. Conceptual diagram for constructing the MEC estimator. The notation $\omega$ denotes a generic set of weights in the weighted estimating function $U_n^\omega(\theta)$. In the MEC construction, $\widehat{d}_i$ denotes the baseline weight and $\widehat{w}_i$ denotes the MEC-updated weight. When all weights in $\omega$ are updated, we continue to use $\omega$, rather than introducing separate $w$-notation, to avoid notational clutter. This is the case in the semi-supervised mean-estimation setting in the main paper, where all weights in the rectifier term are updated.

## B.2. MEC implementation under weighted estimating-equations

**Weighted estimating-equation setup.** Let $O_i$, $i = 1, \ldots, n$, denote independent observed data units in the estimating-equation problem under consideration. The components of $O_i$ are application-specific and may include covariates, outcomes, treatment or source indicators, sampling indicators, and variables needed to define the estimating equation and baseline weights. Let $\theta_0$ denote the target parameter, characterized as the unique root of the population estimating equation

$$U(\theta_0) = 0.$$

We suppose that $\theta_0$ is a low-dimensional Euclidean parameter, such as a mean, regression coefficient, marginal log-hazard ratio, or average treatment effect.

Suppose that $U(\theta)$ is the population limit of a weighted sample estimating function $U_n^\omega(\theta)$. Here, $\omega$ denotes the collection of weights attached to the weighted contributions in $U_n^\omega(\theta)$; these need not correspond one-to-one to all observed units when the estimating equation contains both weighted and unweighted components. For example, in semi-supervised mean estimation (Example 1 in Subsection B.3), the weights are attached only to the labeled residual-correction terms, whereas the target covariate plug-in terms are unweighted.

For an admissible choice of estimating-equation weights, assume that

$$n^{-1}U_n^\omega(\theta) \xrightarrow{p} U(\theta)$$

uniformly over $\theta \in \mathcal{N}(\theta_0)$, where $\mathcal{N}(\theta_0)$ is a neighborhood of $\theta_0$. The limit may be a positive scalar multiple of $U(\theta)$, which has the same unique root and is therefore immaterial for consistency. Then, under standard $Z$-estimation regularity conditions (Van der Vaart, 2000) and, in causal-inference settings, the relevant identification assumptions (Imbens & Rubin, 2015), any solution $\widehat{\theta}$ to

$$U_n^\omega(\theta) = 0 \tag{17}$$

is consistent for $\theta_0$. We note that the notation $U_n^\omega(\theta)$ is somewhat abstract; therefore, in Subsection B.3, we provide concrete examples of MEC applications for clarity.

**Step 1: Specify the initial weights and the source and target sets.** MEC starts from baseline weights that define a valid weighted estimating equation for the target parameter. Let $\mathcal{I}_S$ denote the source set whose weights may be updated, and let $\mathcal{I}_T$ denote the target set to which the source set is calibrated. For $i \in \mathcal{I}_S$, let $\widehat{d}_i$ denote the baseline weight. When these calibrated source weights form only part of a larger estimating-equation weight vector $\omega$, we denote their MEC-updated versions by $\widehat{w}_i$. Thus, MEC updates the relevant entries of $\omega$ by replacing $\widehat{d}_i$ with $\widehat{w}_i$; see Example 3 in Subsection B.3. If MEC updates all weights appearing in $\omega$, then no separate $w$-notation is needed; we simply write $\widehat{\omega}_i$ for the MEC-updated weights; see Examples 1 and 2 in Subsection B.3.

The baseline weights $\{\widehat{d}_i : i \in \mathcal{I}_S\}$ encode the original sampling, treatment-assignment, or transport structure of the problem. When $O_i$ includes a treatment, sampling, or source indicator, constructing $\widehat{d}_i$ may require estimating a propensity score or source-selection probability using either a parametric logistic regression model or a flexible ML method, as is standard in causal inference (Rosenbaum & Rubin, 1983; Imbens & Rubin, 2015). What is essential is that the solution to (17) based on the baseline weights be a valid estimator of the target parameter; at a minimum, consistency should hold under the assumed identification conditions.

Because MEC is fundamentally a weight-updating technique, the scaling of the baseline weights $\{\widehat{d}_i : i \in \mathcal{I}_S\}$ is important. In the total-scale calibration formulation, which we use by default, we recommend normalizing the baseline weights as

$$\sum_{i \in \mathcal{I}_S} \widehat{d}_i = |\mathcal{I}_T|, \tag{18}$$

so that the baseline source weights are on the same total scale as the target sample, where $|\cdot|$ denotes set cardinality. This normalization can often be achieved by rescaling the initial weights as $\widehat{d}_i \leftarrow |\mathcal{I}_T|\widehat{d}_i / \sum_{j \in \mathcal{I}_S} \widehat{d}_j$, for $i \in \mathcal{I}_S$.

**Step 2: Construct the calibration basis.** The distinctive feature of MEC is that the baseline weights $\{\widehat{d}_i : i \in \mathcal{I}_S\}$ are updated using an outcome-predictive calibration basis constructed from cross-fitted ML predictions. We use the superscript $(-)$ when we wish to emphasize the out-of-fold construction. Specifically, define the $p$-dimensional basis

$$\widehat{h}_i = \widehat{h}(X_i) = \left(1, \widehat{b}_1^{(-)}(X_i), \widehat{b}_2^{(-)}(X_i), \ldots, \widehat{b}_{p-1}^{(-)}(X_i)\right) \in \mathbb{R}^p, \tag{19}$$

where $\widehat{b}_\ell^{(-)}(\cdot)$, $\ell = 1, \ldots, p-1$, denotes an out-of-fold ML prediction used to form the calibration basis. In the main paper, we write $h_i = h(X_i)$, without a hat, to avoid notational clutter, as in (11); here, however, we keep the hat explicit.

The choice of basis in (19) is problem-specific and is left to the analyst; see Section 7 for a related discussion. The term *"machine-learning-assisted"* reflects the use of the predictive power of ML to guide the weight update along directions that may be informative for efficiency. Cross-fitting is central to this construction: each basis value is evaluated out of

sample, so that, conditional on the training folds, the ML-based basis can be treated as a covariate-dependent feature basis, analogous to fixed moment features in survey calibration. This separation between nuisance training and validation observations helps avoid empirical-process restrictions, such as Donsker-type conditions, as in double machine learning and cross-validated targeted maximum likelihood estimation (Chernozhukov et al., 2018; Van der Laan & Rose, 2011). This distinguishes MEC from classical weight calibration, where auxiliary variables are typically prespecified and the target is often a finite-population total or mean (Deville & Särndal, 1992; Haziza & Beaumont, 2017). Model-assisted calibration is closer in spirit because it may use prediction models to form calibration features, but cross-fitting is not central in the classical design-based finite-population setting (Breidt & Opsomer, 2017; Särndal et al., 2003).

**Step 3: Solve the Bregman calibration problem.** MEC obtains calibrated weights $\{\widehat{w}_i : i \in \mathcal{I}_S\}$, which we refer to as MEC-updated weights, by solving

$$\min_{\{w_i > 0 : i \in \mathcal{I}_S\}} \sum_{i \in \mathcal{I}_S} D_G(w_i \| \widehat{d}_i) \quad \text{subject to} \quad \sum_{i \in \mathcal{I}_S} w_i \widehat{h}_i = T_h, \tag{20}$$

where $D_G(\cdot\|\cdot)$ is the Bregman divergence generated by a strictly convex function $G$. Specifically, letting $g = G'$, $D_G(u\|v) = G(u) - G(v) - g(v)(u - v)$. Since we assumed a separable Bregman divergence in Subsection 4.1 of the main paper, the formulation in (20) is equivalent to the Bregman projection in (8).

Table 1 lists representative Bregman entropies for the MEC framework, reporting the generator $G(u)$, its derivative $g(u) = G'(u)$, and the closed-form divergence $D_G(u\|v)$ for several classical choices, including quadratic, Kullback–Leibler, empirical likelihood, Hellinger, log–log, inverse, and Rényi.

*Table 1.* Representative Bregman generators.

| ENTROPY | $G(u)$ | $g(u) = G'(u)$ | $D_G(u\|v)$ |
|---|---|---|---|
| Quadratic | $\frac{1}{2}u^2$ | $u$ | $\frac{1}{2}(u - v)^2$ |
| Kullback–Leibler | $u \log u$ | $\log u + 1$ | $u \log(u/v) - u + v$ |
| Empirical likelihood | $-\log u$ | $-u^{-1}$ | $-\log(u/v) + u/v - 1$ |
| Squared Hellinger | $(\sqrt{u} - 1)^2$ | $1 - u^{-1/2}$ | $\sqrt{v}\left(1 - \sqrt{u/v}\right)^2$ |
| Inverse | $\frac{1}{2u}$ | $-\frac{1}{2}u^{-2}$ | $\frac{1}{2}\left(\frac{1}{u} - \frac{1}{v}\right) + \frac{u-v}{2v^2}$ |
| Rényi ($\alpha > 0$) | $\frac{u^{\alpha+1}}{\alpha+1}$ | $u^\alpha$ | $\frac{u^{\alpha+1} - v^{\alpha+1}}{\alpha+1} - v^\alpha(u - v)$ |

Figure 3 shows the shape of the Bregman divergences $D_G(u\|v)$ for six representative entropy generators in Table 1. The figure illustrates how the choice of generator $G$ determines the local geometry of the divergence: while the quadratic entropy produces a symmetric parabola, the others can exhibit asymmetric or heavy-tailed growth, reflecting distinct penalty structures for deviations between $u$ and $v$. These geometric differences underlie the varying weighting behavior in entropy-based calibration methods.

In the total-scale formulation, which we use by default, one natural choice is $T_h = \sum_{j \in \mathcal{I}_T} \widehat{h}_j$. Here, because the calibration basis in (19) includes an intercept as its first component, the total-scale constraint in (20) implies

$$\sum_{i \in \mathcal{I}_S} w_i = |\mathcal{I}_T|.$$

Therefore, both the baseline weights $\{\widehat{d}_i : i \in \mathcal{I}_S\}$ and the MEC-updated weights $\{\widehat{w}_i : i \in \mathcal{I}_S\}$ are on the same total scale. It is important to note that, with $\alpha_i = w_i/|\mathcal{I}_T|$, the same constraint can be equivalently written as $\sum_{i \in \mathcal{I}_S} \alpha_i \widehat{h}_i = (1/|\mathcal{I}_T|) \sum_{j \in \mathcal{I}_T} \widehat{h}_j$ and $\sum_{i \in \mathcal{I}_S} \alpha_i = 1$. Thus, the calibration step matches the empirical target mean of the basis.

**Step 4: Form the MEC estimator.** Finally, the MEC-updated weights $\{\widehat{w}_i : i \in \mathcal{I}_S\}$ are inserted into the weighted estimating equation (17), yielding the MEC estimator $\widehat{\theta}_{\mathrm{MEC}}$.

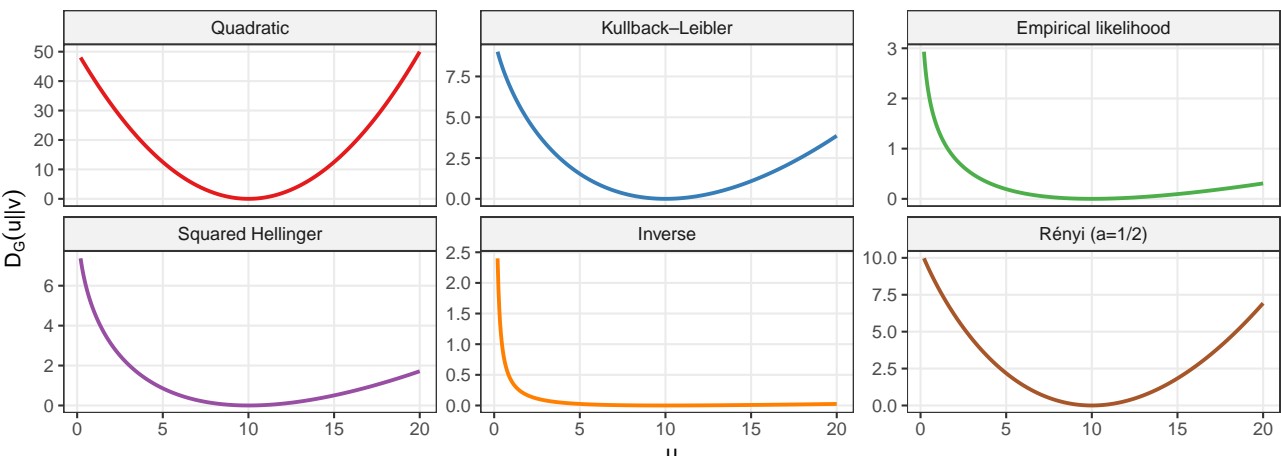

*Figure 3.* Bregman divergences $D_G(u\|v)$, $v = 10$ for six representative entropy generators (Quadratic, Kullback–Leibler, Empirical likelihood, Squared Hellinger, Inverse, and Rényi with $\alpha = 1/2$); see Table 1.

### B.3. Examples of MEC applications

In this paper, our main application of MEC is semi-supervised mean estimation. Here, we present two additional examples illustrating how MEC can be applied to different weighted estimating equations.

**Example 1. Semi-supervised mean estimation (this paper).** Let $O_i = (X_i, \delta_i, \delta_i Y_i)$, $i \in \mathcal{I}_T = \{1, \ldots, N\}$, denote the observed data unit, where $X_i \in \mathcal{X}$ is the covariate vector, $Y_i \in \mathbb{R}$ is the response, and $\delta_i = I(i \in \mathcal{I}_S)$ is the label indicator. Thus, covariates are observed for all units in the target sample $\mathcal{I}_T$, whereas outcome information is available only for the labeled subset $\mathcal{I}_S = \{i : \delta_i = 1\}$, with $|\mathcal{I}_S| = n$. Equivalently, the observed data consist of $\{X_i : i \in \mathcal{I}_T\}$ and $\{(X_j, Y_j) : j \in \mathcal{I}_S\}$. In typical semi-supervised settings, $N$ is much larger than $n$.

The target parameter is the superpopulation mean

$$\theta_0 = \mathbb{E}(Y),$$

which is characterized as the unique solution to $U(\theta_0) = \mathbb{E}(Y - \theta_0) = 0$.

Given a prediction rule $m : \mathcal{X} \to \mathbb{R}$ for the response $Y$ given covariates $X$, the underlying weighted estimating equation can be written as

$$U_{n,N}^\omega(\theta) = U_{n,N}^\omega(\theta; m) = \sum_{i \in \mathcal{I}_T} \{m(X_i) - \theta\} + \sum_{j \in \mathcal{I}_S} \omega_j \{Y_j - m(X_j)\} = 0,$$

where $\omega_j$ denotes the weight assigned to the labeled-sample residual-correction, or rectifier, term in PPI (Angelopoulos et al., 2023; 2024).

In this example, the target-sample plug-in terms have fixed unit weights, while the labeled residual-correction terms carry weights $\omega_j$. MEC updates only these labeled-sample weights, leaving the target-sample unit weights fixed. Thus, no separate $w$-notation is needed; we continue to write $\widehat{\omega}_j$ for the MEC-updated labeled-sample weights. This is consistent with the notation used in the main paper when illustrating MEC for PPI.

In the baseline semi-supervised estimator, $\omega_j = \widehat{d}_j = d_j = N/n$, so $\sum_{j \in \mathcal{I}_S} \widehat{d}_j = N = |\mathcal{I}_T|$. Thus, the baseline weights are already on the target-sample scale.

Let $n, N \to \infty$ with $n/N \to f \in (0, 1)$. Then

$$\frac{1}{N} U_{n,N}^\omega(\theta; m) = \frac{1}{N} \sum_{i \in \mathcal{I}_T} \{m(X_i) - \theta\} + \frac{1}{n} \sum_{j \in \mathcal{I}_S} \{Y_j - m(X_j)\} \xrightarrow{p} \mathbb{E}(Y - \theta) = U(\theta).$$

Equivalently, $n^{-1} U_{n,N}^\omega(\theta; m) \xrightarrow{p} f^{-1} U(\theta)$, which has the same unique root as $U(\theta)$. Thus, $U_{n,N}^\omega(\theta; m) = 0$ is a valid sample analogue of the population equation $U(\theta) = 0$.

For MEC, we first construct a cross-fitted ML predictor $\widehat{m}^{(-)}$, where $\widehat{m}^{(-)}(X_j)$ denotes the out-of-fold prediction for $j \in \mathcal{I}_S$. We then use the predictor basis $\widehat{h}_j = (1, \widehat{m}^{(-)}(X_j))$ to update the baseline weights $\widehat{d}_j = N/n$ through the MEC calibration step. The resulting MEC-updated weights $\widehat{\omega}_j$ satisfy

$$\sum_{j \in \mathcal{I}_S} \widehat{\omega}_j = N, \qquad \sum_{j \in \mathcal{I}_S} \widehat{\omega}_j \widehat{m}^{(-)}(X_j) = \sum_{i \in \mathcal{I}_T} \widehat{m}^{(-)}(X_i).$$

Solving $U_{n,N}^{\widehat{\omega}}(\theta; \widehat{m}^{(-)}) = 0$ gives $\widehat{\theta}_{\mathrm{MEC}} = N^{-1} \sum_{i \in \mathcal{I}_T} \widehat{m}^{(-)}(X_i) + N^{-1} \sum_{j \in \mathcal{I}_S} \widehat{\omega}_j \{Y_j - \widehat{m}^{(-)}(X_j)\}$. By the second calibration constraint, this reduces to $\widehat{\theta}_{\mathrm{MEC}} = N^{-1} \sum_{j \in \mathcal{I}_S} \widehat{\omega}_j Y_j$, as derived in (12).

**Example 2. Average treatment-effect (ATE) estimation and a doubly robust (DR) representation.** Let $O_i = (X_i, A_i, Y_i)$, $i = 1, \ldots, n$, denote the observed data unit, where $X_i \in \mathcal{X}$ is the covariate vector, $A_i \in \{0, 1\}$ is the treatment indicator, and $Y_i$ is the observed outcome. Let $Y_i^1$ and $Y_i^0$ denote the potential outcomes under treatment and control, respectively, so that, under consistency, $Y_i = A_i Y_i^1 + (1 - A_i) Y_i^0$. The average treatment effect is

$$\theta_0 = \mathbb{E}(Y^1 - Y^0).$$

Under consistency, positivity, and no unmeasured confounding (Imbens & Rubin, 2015), $\theta_0$ can be characterized as the unique solution to the inverse-probability-weighted (IPW) population estimating equation

$$U_{\mathrm{ipw}}(\theta_0) = \mathbb{E}\left\{ \frac{AY}{\pi(X)} - \frac{(1-A)Y}{1 - \pi(X)} - \theta_0 \right\} = 0,$$

where $\pi(X) = \mathbb{P}(A = 1 \mid X)$ denotes the true propensity score. Let $\mathcal{I}_{S,1} = \{i : A_i = 1\}$ and $\mathcal{I}_{S,0} = \{i : A_i = 0\}$ denote the treated and control source sets, respectively, and let $\mathcal{I}_T$ denote the target sample representing the covariate distribution of interest. For the ATE in the observed population, a natural default choice is $\mathcal{I}_T = \mathcal{I}_{S,1} \cup \mathcal{I}_{S,0} = \{1, \ldots, n\}$.

The weighted estimating equation can be written as

$$U_n^{\omega_1, \omega_0}(\theta) = \sum_{i \in \mathcal{I}_{S,1}} \omega_{1i} Y_i - \sum_{i \in \mathcal{I}_{S,0}} \omega_{0i} Y_i - |\mathcal{I}_T|\theta = 0,$$

where $\omega_{1i}$ and $\omega_{0i}$ are treatment-specific estimating-equation weights. At the population level, the corresponding inverse-probability weights are $\omega_{1i} = 1/\pi(X_i)$ for $i \in \mathcal{I}_{S,1}$ and $\omega_{0i} = 1/\{1 - \pi(X_i)\}$ for $i \in \mathcal{I}_{S,0}$. Since $|\mathcal{I}_T| = n$ under the default ATE target, under the identification conditions above,

$$\frac{1}{n} U_n^{\omega_1, \omega_0}(\theta) \xrightarrow{p} \mathbb{E}\left\{ \frac{AY}{\pi(X)} - \frac{(1-A)Y}{1 - \pi(X)} - \theta \right\} = \mathbb{E}(Y^1 - Y^0 - \theta) = U(\theta).$$

Thus, $U_n^{\omega_1, \omega_0}(\theta) = 0$ is a valid sample analogue of the population estimating equation $U(\theta) = 0$.

We first define the unnormalized treatment-specific baseline weights as $\widehat{d}_{1i} = 1/\widehat{\pi}^{(-)}(X_i)$ for $i \in \mathcal{I}_{S,1}$ and $\widehat{d}_{0i} = 1/\{1 - \widehat{\pi}^{(-)}(X_i)\}$ for $i \in \mathcal{I}_{S,0}$, where $\widehat{\pi}^{(-)}(\cdot)$ denotes the cross-fitted propensity-score estimator. For exact finite-sample total-scale normalization, we use

$$\widehat{d}_{1i} \leftarrow \frac{|\mathcal{I}_T|\widehat{d}_{1i}}{\sum_{j \in \mathcal{I}_{S,1}} \widehat{d}_{1j}}, \quad i \in \mathcal{I}_{S,1}, \qquad \widehat{d}_{0i} \leftarrow \frac{|\mathcal{I}_T|\widehat{d}_{0i}}{\sum_{j \in \mathcal{I}_{S,0}} \widehat{d}_{0j}}, \quad i \in \mathcal{I}_{S,0}.$$

Then $\sum_{i \in \mathcal{I}_{S,1}} \widehat{d}_{1i} = \sum_{i \in \mathcal{I}_{S,0}} \widehat{d}_{0i} = |\mathcal{I}_T|$, placing each treatment-specific source sample on the same total scale as the target sample.

For MEC, the baseline treatment-specific weights $\widehat{d}_{1i}$ and $\widehat{d}_{0i}$ are replaced by MEC-updated weights $\widehat{\omega}_{1i}$, $i \in \mathcal{I}_{S,1}$, and $\widehat{\omega}_{0i}$, $i \in \mathcal{I}_{S,0}$, respectively. These updated weights remain close to the baseline IPW weights, in the Bregman sense, while balancing an outcome-predictive calibration basis. For example, one may use

$$\widehat{h}_i = (1, \widehat{m}_0^{(-)}(X_i), \widehat{m}_1^{(-)}(X_i)),$$

where $\widehat{m}_a^{(-)}(\cdot)$, $a = 0, 1$, denotes an out-of-fold estimator of the treatment-specific outcome regression

$$m_a(x) = \mathbb{E}(Y \mid A = a, X = x), \qquad a = 0, 1.$$

The estimators $\widehat{m}_1^{(-)}(\cdot)$ and $\widehat{m}_0^{(-)}(\cdot)$ are obtained by cross-fitting outcome-regression learners within the treated and control samples, respectively. This is analogous to cross-fitted outcome nuisance estimation in DML (Chernozhukov et al., 2018) and CV-TMLE (Van der Laan & Rose, 2011; 2018).

The MEC-updated treated and control weights are constructed to satisfy

$$\sum_{i \in \mathcal{I}_{S,1}} \widehat{\omega}_{1i} \widehat{h}_i = \sum_{i \in \mathcal{I}_T} \widehat{h}_i, \qquad \sum_{i \in \mathcal{I}_{S,0}} \widehat{\omega}_{0i} \widehat{h}_i = \sum_{i \in \mathcal{I}_T} \widehat{h}_i. \tag{21}$$

The resulting MEC-type ATE estimator is the solution to $U_n^{\widehat{\omega}_1, \widehat{\omega}_0}(\theta) = 0$, namely

$$\widehat{\theta}_{\text{MEC}} = \frac{1}{|\mathcal{I}_T|} \left[ \sum_{i \in \mathcal{I}_{S,1}} \widehat{\omega}_{1i} Y_i - \sum_{i \in \mathcal{I}_{S,0}} \widehat{\omega}_{0i} Y_i \right]. \tag{22}$$

Thus, in contrast to Example 1, where there is a single set of estimating-equation weights, MEC here updates two treatment-specific sets of weights corresponding to the treated and control mean components of the ATE.

The same estimator can also be written through a doubly robust orthogonal-score representation (Bang & Robins, 2005). The Neyman-orthogonal ATE score is

$$U_{\text{orth}}(\theta_0; \eta_0) = \mathbb{E} \left[ m_1(X) - m_0(X) + \frac{A}{\pi(X)} \{Y - m_1(X)\} - \frac{1 - A}{1 - \pi(X)} \{Y - m_0(X)\} - \theta_0 \right] = 0,$$

where $\eta_0 = \{\pi, m_0, m_1\}$. Using the same MEC-updated treatment-specific weights $\widehat{\omega}_{1i}$ and $\widehat{\omega}_{0i}$, the orthogonal-score estimator is obtained by solving $U_{n,\text{orth}}^{\widehat{\omega}_1, \widehat{\omega}_0}(\theta; \widehat{\eta}) = 0$, which yields

$$\widehat{\theta}_{\text{MEC-DR}} = \frac{1}{|\mathcal{I}_T|} \left[ \sum_{i \in \mathcal{I}_T} \{\widehat{m}_1^{(-)}(X_i) - \widehat{m}_0^{(-)}(X_i)\} + \sum_{i \in \mathcal{I}_{S,1}} \widehat{\omega}_{1i} \{Y_i - \widehat{m}_1^{(-)}(X_i)\} - \sum_{i \in \mathcal{I}_{S,0}} \widehat{\omega}_{0i} \{Y_i - \widehat{m}_0^{(-)}(X_i)\} \right]$$

$$= \frac{1}{|\mathcal{I}_T|} \left[ \sum_{i \in \mathcal{I}_{S,1}} \widehat{\omega}_{1i} Y_i - \sum_{i \in \mathcal{I}_{S,0}} \widehat{\omega}_{0i} Y_i \right] = \widehat{\theta}_{\text{MEC}}.$$

The second equality follows from the calibration constraints on $\widehat{m}_0^{(-)}$ and $\widehat{m}_1^{(-)}$ in (21). Therefore, when the treatment-specific outcome regressions are included in the calibration basis, the MEC-type IPW estimator $\widehat{\theta}_{\text{MEC}}$ in (22) is algebraically equivalent to the corresponding MEC-type DR estimator $\widehat{\theta}_{\text{MEC-DR}}$. This equivalence is a special feature of the linear mean-estimation problem, where calibration on the treatment-specific outcome regressions makes the IPW representation coincide with the Neyman-orthogonal representation.

**Example 3. Marginal log-hazard ratio estimation via weighted Cox regression.** In the externally controlled single-arm trial setting, let $\mathcal{I}_T = \{i : A_i = 1\}$ denote the treated trial target set and $\mathcal{I}_S = \{i : A_i = 0\}$ denote the external-control source set, with $n_1 = |\mathcal{I}_T|$. For each subject, the observed data are $O_i = (Y_i, \delta_i, A_i, X_i)$, where $Y_i = \min(T_i, C_i)$ is the observed follow-up time, $\delta_i = I(T_i \leq C_i)$ is the event indicator, $A_i \in \{0, 1\}$ is the treatment/source indicator, and $X_i$ is the baseline covariate vector. Define the counting process and at-risk process by $\mathcal{N}_i(t) = I(Y_i \leq t, \delta_i = 1)$ and $\mathcal{Y}_i(t) = I(Y_i \geq t)$. The target parameter is the ATT marginal log-hazard ratio $\theta_{ATT}$. Under the marginal proportional hazards model (Hernán et al., 2000), $\theta_{ATT}$ is characterized as the root of the population limit of the ATT-weighted Cox score (Lin & Wei, 1989; Binder, 1992),

$$U_n^\omega(\theta) = \sum_{i \in \mathcal{I}_T \cup \mathcal{I}_S} \int \omega_i \{A_i - \bar{A}_\omega(t; \theta)\} \, d\mathcal{N}_i(t) = 0,$$

where

$$\bar{A}_\omega(t;\theta) = \frac{\sum_{i\in\mathcal{I}_T\cup\mathcal{I}_S}\omega_i\mathcal{Y}_i(t)\exp(\theta A_i)A_i}{\sum_{i\in\mathcal{I}_T\cup\mathcal{I}_S}\omega_i\mathcal{Y}_i(t)\exp(\theta A_i)}, \quad \omega_i = A_i + (1-A_i)q(X_i), \quad q(X_i) = \frac{\pi(X_i)}{1-\pi(X_i)} = \frac{\mathbb{P}(A=1\mid X)}{\mathbb{P}(A=0\mid X)}.$$

For $i\in\mathcal{I}_S$, the baseline ATT transport weight is normalized as $\widehat{d}_i = n_1\widehat{q}_i/\sum_{j\in\mathcal{I}_S}\widehat{q}_j$, where $\widehat{q}_i = \widehat{\pi}_i/(1-\widehat{\pi}_i)$, with $\widehat{\pi}_i = \widehat{\pi}^{(-)}(X_i)$, and $n_1 = |\mathcal{I}_T|$. Thus, $\sum_{i\in\mathcal{I}_S}\widehat{d}_i = n_1 = |\mathcal{I}_T|$, so the baseline external-control weights are on the same total scale as the treated trial sample. MEC replaces the normalized baseline external-control weights $\widehat{d}_i$, $i\in\mathcal{I}_S$, by MEC-updated weights $\widehat{w}_i$ that remain close to $\widehat{d}_i$ and satisfy

$$\sum_{i\in\mathcal{I}_S}\widehat{w}_i\widehat{h}_i = \sum_{i\in\mathcal{I}_T}\widehat{h}_i, \qquad \widehat{h}_i = \{1, \widehat{S}_0^{(-)}(t_1\mid X_i),\ldots,\widehat{S}_0^{(-)}(t_{p-1}\mid X_i)\}^\top.$$

Here, $\widehat{S}_0^{(-)}(t\mid X_i)$ denotes a cross-fitted estimate of the external-control survival function. Because $\widehat{h}_i$ includes an intercept, the calibration constraint implies $\sum_{i\in\mathcal{I}_S}\widehat{w}_i = n_1$, so the calibrated external-control weights remain on the same total scale as the treated trial sample. The final Cox estimating-equation weight is $\widehat{\omega}_i = A_i + (1-A_i)\widehat{w}_i$. The MEC estimator $\widehat{\theta}_{\mathrm{MEC}}$ is then obtained by solving the weighted Cox estimating equation $U_n^{\widehat{\omega}}(\theta) = 0$.

### B.4. Relationship with TMLE, DML, and prediction-assisted adjustment

MEC is related to several modern frameworks that use flexible ML predictions as intermediate inputs for statistically principled estimation. In TMLE, ML estimates of nuisance functions are first obtained and then updated through a fluctuation step so that the resulting estimator solves an EIF estimating equation (Van der Laan & Rose, 2011; 2018). In DML, ML estimates of nuisance functions are inserted into a Neyman-orthogonal score (Chernozhukov et al., 2018; Mackey et al., 2018; Foster & Syrgkanis, 2023). Thus, in both TMLE and DML, ML predictions enter *directly* through nuisance-function components of an EIF-based or orthogonal-score representation.

MEC uses ML predictions differently: *implicitly*, through the construction of the calibration basis in (19). Starting from a weighted estimating equation, which need not be EIF-based or Neyman-orthogonal, MEC uses cross-fitted outcome-predictive features to define calibration constraints and obtains the updated weights as a minimal Bregman perturbation of the baseline weights. Thus, the prediction rule determines outcome- or score-relevant directions along which the weighted source sample is balanced to the target sample, rather than being used to form a plug-in estimator, specify a fluctuation submodel, or construct an orthogonal score.

It is important to emphasize that not starting from an EIF or an orthogonal score does not mean that MEC is disconnected from orthogonality or efficiency considerations. In fact, a carefully chosen calibration basis may recover an orthogonal or doubly robust representation; see Example 2 of Subsection B.3. Moreover, as discussed in this paper, semiparametric efficiency can still be studied theoretically through the asymptotic properties of the resulting MEC estimator, even though the implementation of MEC does not require deriving an EIF or orthogonal score in advance; see Theorem 5.2.

MEC is also related in spirit to second-stage adjustment strategies for ML predictions, such as generalized Oaxaca–Blinder (GOB) estimators (Guo & Basse, 2023) and no-harm calibration in randomized experiments (Lin, 2013; Cohen & Fogarty, 2024). GOB estimators use fitted outcome models to construct covariate-adjusted treatment-effect estimators, while no-harm calibration modifies such nonlinear adjustment procedures to avoid efficiency loss relative to simpler benchmarks. MEC follows a similar broad principle of using ML predictions as intermediate adjustment information, but it implements the adjustment through Bregman calibration of weights.

In summary, the broad applicability of MEC comes from its formulation at the level of weighted estimating equations. MEC is not tied to a specific EIF-targeting step, as in TMLE, a Neyman-orthogonal score construction, as in DML, or a randomized-experiment mean-estimation framework, as in GOB and no-harm calibration. Instead, MEC may be applied whenever the problem provides three ingredients: a valid weighted estimating equation, baseline weights to be perturbed, and a source–target calibration structure based on outcome-predictive features. This includes semi-supervised inference (Zhang et al., 2019; Angelopoulos et al., 2023; Zrnic & Candès, 2024), missing-data problems (Robins et al., 1995; Little & Rubin, 2019), causal-inference settings (Imbens & Rubin, 2015; Rosenbaum & Rubin, 1983), and weighted survival estimating equations (Lin & Wei, 1989; Binder, 1992; Robins et al., 2000), among many other examples.

## C. Semiparametric efficiency lower bound

We study the semiparametric efficiency lower bound for estimating the superpopulation mean $\theta_0 = \mathbb{E}[Y]$ under our basic setup. For any fixed (possibly misspecified) predictor $m : \mathcal{X} \to \mathbb{R}$, the PPI estimator solves the estimating equation

$$\widehat{U}_{\mathrm{PPI}}(\theta) = \frac{1}{N} \sum_{i=1}^{N} \{m(X_i) - \theta\} + \frac{1}{n} \sum_{j \in S} \{Y_j - m(X_j)\} = 0,$$

whose solution is $\widehat{\theta}_{\mathrm{PPI},m}$ in (1). A standard linearization (Yuan & Jennrich, 1998; Van der Vaart, 2000), with inclusion indicators $\delta_i = \mathbb{I}\{i \in S\}$, yields the following asymptotically linear representation:

$$\widehat{\theta}_{\mathrm{PPI},m} - \theta_0 = \frac{1}{N} \sum_{i=1}^{N} \phi(O_i) + o_p(N^{-1/2}),$$

where $O_i = (X_i, Y_i, \delta_i)$ with $\delta \perp (X, Y)$ and the influence function is $\phi(O_i) = m(X_i) - \theta_0 + (\delta_i/f)\{Y_i - m(X_i)\}$. Consequently, asymptotic normality holds:

$$\sqrt{N}\, (\widehat{\theta}_{\mathrm{PPI},m} - \theta_0) \xrightarrow{d} \mathcal{N}(0, \sigma_f^2), \tag{23}$$

where $\sigma_f^2 = \mathrm{Var}(\phi(O_i)) = \mathrm{Var}(Y) + (f^{-1} - 1)\mathbb{E}[(Y - m(X))^2]$. By the law of total variance and $Y = m_0(X) + \varepsilon$ with $\mathbb{E}[\varepsilon \mid X] = 0$, we can decompose $\sigma_f^2$ as

$$\sigma_f^2 = \underbrace{\mathrm{Var}(m_0(X)) + f^{-1}\, \mathbb{E}[\mathrm{Var}(Y \mid X)]}_{\text{semiparametric efficiency lower bound}} + \underbrace{(f^{-1} - 1)\, \mathbb{E}[(m_0(X) - m(X))^2]}_{\text{inflation due to misspecification } \geq 0}. \tag{24}$$

In particular, the variance is minimized at the oracle predictor $m(x) = m_0(x) = \mathbb{E}[Y \mid X = x]$, which yields the oracle PPI estimator

$$\widehat{\theta}_{\mathrm{PPI},m_0} = \frac{1}{N} \sum_{i=1}^{N} m_0(X_i) + \frac{1}{n} \sum_{j \in S} \{Y_j - m_0(X_j)\},$$

which minimizes the asymptotic variance of PPI, which is

$$\sigma_{f,\mathrm{eff}}^2 = \mathrm{Var}\left(\phi^{\mathrm{eff}}(O_i)\right) = \mathrm{Var}(m_0(X)) + \frac{1}{f}\mathbb{E}[\mathrm{Var}(Y \mid X)], \tag{25}$$

where $\phi^{\mathrm{eff}}$ denotes the efficient influence function (EIF)

$$\phi^{\mathrm{eff}}(O_i) = m_0(X_i) - \theta_0 + \frac{\delta_i}{f}\{Y_i - m_0(X_i)\}. \tag{26}$$

This result coincides with the semiparametric efficiency lower bound for regular, asymptotically linear (RAL) estimators of $\theta_0 = \mathbb{E}[Y]$ under missing-at-random sampling (Robins et al., 1994): for any RAL estimator $\widehat{\theta}$, it holds

$$\mathrm{Var}(\widehat{\theta}) \geq \frac{1}{N} \left\{ \mathrm{Var}(\mathbb{E}[Y \mid X]) + \mathbb{E}\left[\frac{1}{\pi(X)} \mathrm{Var}(Y \mid X)\right] \right\},$$

and in our setting $\pi(X) = f = n/N$, which reduces exactly to (25). Hence the oracle PPI estimator $\widehat{\theta}_{\mathrm{PPI},m_0}$ attains the semiparametric efficiency lower bound.

We briefly discuss the implications of this derivation. Throughout, the predictor $m$ is treated as a nuisance parameter, while the superpopulation mean $\theta_0 = \mathbb{E}[Y]$ is the target parameter. In practice, however, $m$ must be estimated. To approach the semiparametric efficiency bound, the fitted predictor should be close to the oracle predictor, so that the inflation term in (24) is small; in this regime, the resulting PPI procedures are nearly semiparametrically efficient. Conversely, substantial misspecification—i.e., when the learned predictor is far from the oracle predictor—leads to an increase in variance of order $(f^{-1} - 1)\, \mathbb{E}[(m_0(X) - m(X))^2]$, an effect that becomes more pronounced as the labeling fraction $f$ decreases.

### C.1. Dual Newton solver

We describe the numerical optimization used to obtain the calibrated weights. The central idea is to convert the constrained Bregman projection into an unconstrained root–finding problem in the dual variables and to solve it by Newton–Raphson with backtracking.

First, we state the primal problem. Define the labeled design matrix and the population totals as

$$Z := \left[ z_j^\top \right]_{j \in S} = \left[ h(X_j)^\top \right]_{j \in S} \in \mathbb{R}^{n \times p}, \qquad \mu := \sum_{i=1}^{N} z_i = \sum_{i=1}^{N} h(X_i) \in \mathbb{R}^p, \tag{27}$$

where $h : \mathcal{X} \subset \mathbb{R}^d \to \mathbb{R}^p$ denotes a user-chosen $p$-dimensional calibration basis, and $z_j := h(X_j) = (h_1(X_j), \dots, h_p(X_j))^\top \in \mathbb{R}^p$. The calibration constraint (9), $\sum_{j \in S} \omega_j h(X_j) = \sum_{i=1}^{N} h(X_i)$, can then be expressed in vector form as $Z^\top \omega = \mu$.

Given baseline weights $d = (d_j)_{j \in S} \in \mathbb{R}^n$ (i.e., $d_j \equiv N/n = 1/f$ in our setting) and a strictly convex generator $G$ with derivative $g = G'$, the calibrated weights are the Bregman projection of $d$ onto the affine subspace defined by the calibration constraint (9):

$$\widehat{\omega} = \arg \min_{\omega \in \mathcal{M}} D_G(\omega \| d) \quad \text{subject to} \quad \mathcal{M} := \{\omega \in (0, \infty)^n : Z^\top \omega = \mu\}. \tag{28}$$

Next, we derive the dual formulation of (28). Introduce Lagrange multipliers $\lambda \in \mathbb{R}^p$ for the calibration constraints and consider the Lagrangian

$$\mathcal{L}(\omega, \lambda) = -\sum_{j \in S} \Big\{ G(\omega_j) - G(d_j) - g(d_j)(\omega_j - d_j) \Big\} + \lambda^\top (Z^\top \omega - \mu). \tag{29}$$

Because $G$ is strictly convex, $D_G(\cdot \| d)$ is strictly convex in $\omega$, and the constraints $Z^\top \omega = \mu$ are affine; hence (28) is a strictly convex program. Whenever the feasible set $\{\omega : Z^\top \omega = \mu\}$ is nonempty, the Karush–Kuhn–Tucker (KKT) conditions are necessary and sufficient, and the optimizer is unique (Kuhn & Tucker, 1951). In particular, at the optimum the stationarity condition holds.

Solving the KKT stationarity condition $\partial \mathcal{L} / \partial \omega_j = 0$ for each $j \in S$ yields $g(\omega_j) = g(d_j) + z_j^\top \lambda$, and hence

$$\omega_j(\lambda) = g^{-1}\big(g(d_j) + z_j^\top \lambda\big). \tag{30}$$

Thus, the optimal weights can be expressed as functions of the Lagrange multiplier $\lambda$.

Substituting (30) into (29) gives

$$\begin{aligned}
\ell(\lambda) &= -\sum_{j \in S} \Big\{ G\big(g^{-1}(\nu_j)\big) - G(d_j) - g(d_j)\big(g^{-1}(\nu_j) - d_j\big) \Big\} + \lambda^\top \Big( \sum_{j \in S} \omega_j(\lambda) z_j - \mu \Big) \\
&= -\sum_{j \in S} G\big(g^{-1}(\nu_j)\big) + \sum_{j \in S} g(d_j)\, g^{-1}(\nu_j) + \sum_{j \in S} (z_j^\top \lambda)\, g^{-1}(\nu_j) - \lambda^\top \mu + \text{const} \\
&= -\sum_{j \in S} G\big(g^{-1}(\nu_j)\big) + \sum_{j \in S} \big(g(d_j) + z_j^\top \lambda\big) g^{-1}(\nu_j) - \lambda^\top \mu + \text{const} \\
&= \sum_{j \in S} \Big[ \nu_j\, g^{-1}(\nu_j) - G\big(g^{-1}(\nu_j)\big) \Big] - \lambda^\top \mu + \text{const},
\end{aligned} \tag{31}$$

where $\nu_j := g(d_j) + z_j^\top \lambda$ and $\omega_j(\lambda) = g^{-1}(\nu_j)$, and const denotes a term that is constant with respect to $\lambda$.

Let $F$ be the convex conjugate of $G$,

$$F(\nu) = \sup_{\omega} \{\nu \omega - G(\omega)\} = \nu\, g^{-1}(\nu) - G\big(g^{-1}(\nu)\big).$$

By elementary calculus, $F'(\nu) = g^{-1}(\nu)$. Then, from (31), up to an additive constant (independent of $\lambda$), the profiled dual objective is

$$\ell(\lambda) = \sum_{j \in S} F\big(g(d_j) + z_j^\top \lambda\big) - \lambda^\top \mu. \tag{32}$$

Since $F$ is convex, $\ell(\lambda)$ is convex, and

$$\nabla \ell(\lambda) = \sum_{j \in S} F'(\nu_j)\, z_j - \mu = \sum_{j \in S} g^{-1}(\nu_j)\, z_j - \mu = \sum_{j \in S} \omega_j(\lambda)\, z_j - \mu. \tag{33}$$

The first–order condition $\nabla \ell(\widehat{\lambda}) = 0$ recovers the calibration constraint $Z^\top \omega(\widehat{\lambda}) = \mu$. Because the primal objective is strictly convex and the constraints are affine, the dual is strictly convex and the minimizer is unique: $\widehat{\lambda} = \arg\min_\lambda \ell(\lambda)$. Plugging $\widehat{\lambda}$ into (30) yields the calibrated weights $\widehat{\omega}_j = \omega_j(\widehat{\lambda}) = g^{-1}\big(g(d_j) + z_j^\top \widehat{\lambda}\big)$ for each $j \in S$.

To minimize the convex dual objective $\ell$ in (32), we use (damped) Newton–Raphson (Boyd & Vandenberghe, 2004). For illustration, we assume that $g$ and $g^{-1}$ act on vectors componentwise, and that the index set of labeled units is $S = \{1, \ldots, n\}$. Let $u := g(d) \in \mathbb{R}^n$ with $u_j = g(d_j)$ and $\nu := u + Z\lambda \in \mathbb{R}^n$ with $\nu_j = g(d_j) + z_j^\top \lambda$. The $n$-dimensional weight vector is $\omega(\lambda) := \big(g^{-1}(\nu_1), \ldots, g^{-1}(\nu_n)\big)^\top$, and the Hessian is

$$\nabla^2 \ell(\lambda) = Z^\top \operatorname{diag}\big(1/g'(\omega_1(\lambda)), \ldots, 1/g'(\omega_n(\lambda))\big) Z \in \mathbb{R}^{p \times p}.$$

Since the generator $G$ is strictly convex, $g' = G'' > 0$; with $Z$ of full column rank, the Hessian is positive definite. A Newton step solves $\nabla^2 \ell(\lambda^{(t)})\, \Delta^{(t)} = -\nabla \ell(\lambda^{(t)})$ (e.g., via Cholesky) and updates

$$\lambda^{(t+1)} = \lambda^{(t)} + \eta_t\, \Delta^{(t)} = \lambda^{(t)} - \eta_t\, [\nabla^2 \ell(\lambda^{(t)})]^{-1} \nabla \ell(\lambda^{(t)}), \quad \eta_t \in (0, 1]. \tag{34}$$

Pure Newton uses $\eta_t = 1$; damping can be used to enforce monotone decrease of $\ell$. Convergence to the optimal dual variable $\widehat{\lambda}$ can be monitored with a stopping rule, $\|\nabla \ell(\lambda^{(t)})\|_2 = \|Z^\top \omega(\lambda^{(t)}) - \mu\|_2 \leq \varepsilon$, for a small tolerance such as $\varepsilon = 10^{-10}$. The corresponding optimal weights are backtracked by $\widehat{\omega} = \omega(\widehat{\lambda}) = g^{-1}\big(g(d) + Z\widehat{\lambda}\big) \in \mathbb{R}^n$. Under these conditions, Newton's method converges rapidly when $n > p$.

Figure 4 illustrates the dual Newton solver's iterations for a single realization of the synthetic data with $f = 0.2$ from Section 6 of the main document. Panel (a) displays the calibration residual $\|Z^\top \omega - \mu\|_2$ with tolerance $\varepsilon = 10^{-10}$, and panel (b) displays the Bregman objective $D_G(\omega \,\|\, d)$ across iterations. For the quadratic divergence, a single Newton step attains the exact solution; for the other divergences, convergence typically occurs within 10 iterations. Panel (b) further shows that the optimized objective $D_G(\widehat{\omega} \,\|\, d)$ remains bounded, supporting Assumption A2 that $D_G(\widehat{\omega} \,\|\, d) + D_G(d \,\|\, \widehat{\omega}) = O_p(1)$, which is used in Theorem 5.2.

To summarize, the calibration–weighting optimization can be approached from a dual perspective, which often yields a more computationally efficient solution. The primal problem seeks the optimal weights by minimizing a Bregman divergence—an $n$-dimensional optimization. The dual formulation converts this into an unconstrained optimization over the Lagrange multiplier vector $\lambda$, which is $p$-dimensional (with $p$ equal to the dimension of the codomain of the basis function $h$). When $n \gg p$, this reduction in dimensionality provides a substantial computational advantage.

## D. Proof of Theorem 4.1 (Dual GREG representation of the MEC estimator)

Recall the definition of the Bregman divergence

$$D_G(\omega \| d) = \sum_{j \in S} \Big\{ G(\omega_j) - G(d_j) - g(d_j)(\omega_j - d_j) \Big\},$$

and consider the Lagrangian

$$\mathcal{L}(\omega, \lambda) = -\sum_{j \in S} \Big\{ G(\omega_j) - G(d_j) - g(d_j)(\omega_j - d_j) \Big\} + \lambda^\top \Big( \sum_{j \in S} \omega_j z_j - \mu \Big),$$

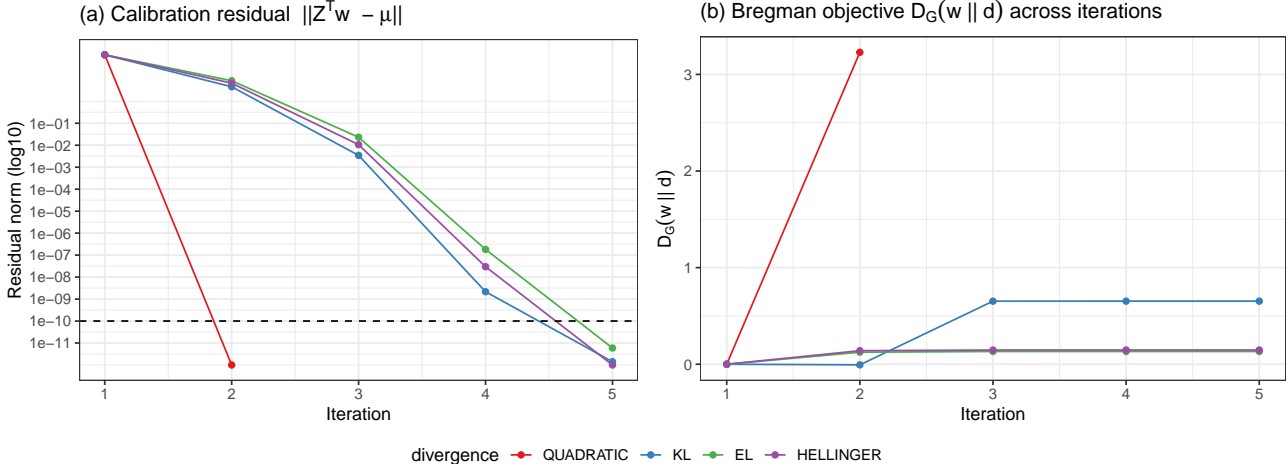

*Figure 4.* Iteration procedure of dual Newton solver on a single realization of the synthetic data with $f = 0.2$ in Section 6 of the main document. Panel (a) shows the calibration residual $\|Z^\top \omega - \mu\|_2$ (tolerance $\varepsilon = 10^{-10}$); panel (b) shows the Bregman objective $D_G(\omega\|d)$ by iteration. The quadratic case converges in one step; others converge within $\leq 10$ steps, with objectives remaining bounded.

where $\mu$ is defined in (27), $\mu = \sum_{i=1}^{N} z_i$, with $z_j := h(X_j) = (h_1(X_j), \ldots, h_p(X_j))^\top \in \mathbb{R}^p$.

KKT stationarity gives, for each $j \in S$,

$$\partial_{\omega_j} \mathcal{L} = -g(\widehat{\omega}_j) + g(d_j) + z_j^\top \lambda = 0 \quad \implies \quad g(\widehat{\omega}_j) = g(d_j) + z_j^\top \lambda.$$

By strict convexity and $C^2$–smoothness of $G$ on $(0, \infty)$, $g$ is strictly increasing and $g^{-1}$ is $C^2$. A second–order Taylor expansion of $g^{-1}$ around $g(d_j)$ yields, for some remainder $r_j(\lambda)$,

$$\widehat{\omega}_j = g^{-1}\big(g(d_j) + z_j^\top \widehat{\lambda}\big) = d_j + \frac{1}{g'(d_j)} z_j^\top \widehat{\lambda} + r_j(\widehat{\lambda}), \qquad |r_j(\widehat{\lambda})| \leq K \|\widehat{\lambda}\|^2, \tag{35}$$

where $K < \infty$ depends on local bounds on $(g^{-1})''$ and $\{z_j\}$. Write $q_j := 1/g'(d_j)$.

Summing (35) after multiplying by $z_j$ gives

$$\sum_{j \in S} \widehat{\omega}_j z_j = \sum_{j \in S} d_j z_j + \Big(\sum_{j \in S} q_j z_j z_j^\top\Big)\widehat{\lambda} + \sum_{j \in S} r_j(\widehat{\lambda})\, z_j = \mu,$$

so

$$\Big(\sum_{j \in S} q_j z_j z_j^\top\Big)\widehat{\lambda} = \mu - \sum_{j \in S} d_j z_j - \sum_{j \in S} r_j(\widehat{\lambda})\, z_j. \tag{36}$$

Let $\widehat{\beta}$ be the WLS solution

$$\Big(\sum_{j \in S} q_j z_j z_j^\top\Big)\widehat{\beta} = \sum_{j \in S} q_j z_j Y_j, \qquad \widehat{y}_i = \widehat{\beta}^\top z_i,$$

equivalently

$$\sum_{j \in S} q_j z_j \big(Y_j - \widehat{\beta}^\top z_j\big) = 0. \tag{37}$$

Now, we show the main decomposition. Start from

$$\widehat{\theta}_{\text{MEC}} - \widehat{\theta}_{\text{GREG}} = \frac{1}{N} \left\{ \sum_{j \in S} \widehat{\omega}_j Y_j - \sum_{i \in U} \widehat{y}_i - \sum_{j \in S} d_j (Y_j - \widehat{y}_j) \right\}$$

$$= \frac{1}{N} \left\{ \sum_{j \in S} \widehat{\omega}_j Y_j - \sum_{j \in S} \widehat{\omega}_j \widehat{\beta}^\top z_j - \sum_{j \in S} d_j (Y_j - \widehat{\beta}^\top z_j) \right\}$$

$$= \frac{1}{N} \sum_{j \in S} (\widehat{\omega}_j - d_j) \left( Y_j - \widehat{\beta}^\top z_j \right),$$

where $U$ denotes the index set for unlabeled data. The second equality holds since

$$\sum_{i \in U} \widehat{y}_i = \widehat{\beta}^\top \mu = \widehat{\beta}^\top Z^\top \omega = \sum_{j \in S} \widehat{\omega}_j \widehat{\beta}^\top z_j.$$

Insert (35) gives

$$\widehat{\theta}_{\text{MEC}} - \widehat{\theta}_{\text{GREG}} = \frac{1}{N} \sum_{j \in S} \left( q_j z_j^\top \widehat{\lambda} + r_j(\widehat{\lambda}) \right) \left( Y_j - \widehat{\beta}^\top z_j \right)$$

$$= \underbrace{\frac{1}{N} \sum_{j \in S} q_j z_j^\top \widehat{\lambda} \left( Y_j - \widehat{\beta}^\top z_j \right)}_{T_1} + \underbrace{\frac{1}{N} \sum_{j \in S} r_j(\widehat{\lambda}) \left( Y_j - \widehat{\beta}^\top z_j \right)}_{T_2}.$$

By (37), the term $(T_1)$ vanishes exactly:

$$T_1 = \frac{1}{N} \widehat{\lambda}^\top \sum_{j \in S} q_j z_j \left( Y_j - \widehat{\beta}^\top z_j \right) = 0.$$

Hence

$$\widehat{\theta}_{\text{MEC}} - \widehat{\theta}_{\text{GREG}} = T_2 = \frac{1}{N} \sum_{j \in S} r_j(\widehat{\lambda}) \left( Y_j - \widehat{\beta}^\top z_j \right).$$

Under standard moment conditions ensuring $\frac{1}{n} \sum_{j \in S} |Y_j - \widehat{\beta}^\top z_j| = \mathcal{O}_{\mathbb{P}}(1)$ and using $|r_j(\widehat{\lambda})| \leq K \|\widehat{\lambda}\|^2$ uniformly in $j$, we obtain

$$\left| \widehat{\theta}_{\text{MEC}} - \widehat{\theta}_{\text{GREG}} \right| \leq \frac{K}{N} \sum_{j \in S} \|\widehat{\lambda}\|^2 |Y_j - \widehat{\beta}^\top z_j| = \mathcal{O}_{\mathbb{P}}(\|\widehat{\lambda}\|^2),$$

i.e.

$$\widehat{\theta}_{\text{MEC}} = \widehat{\theta}_{\text{GREG}} + R_N, \qquad R_N = \mathcal{O}_{\mathbb{P}}\left( \|\widehat{\lambda}\|^2 \right).$$

Furthermore, if $\|\widehat{\lambda}\| = \mathcal{O}_{\mathbb{P}}(n^{-1/2})$ and the above moment bounds hold, then $R_N = o_{\mathbb{P}}(N^{-1/2})$, so the representation is asymptotically exact.

Particularly, for the quadratic generator, $G(w) = \frac{1}{2} w^2$, we have $g(w) = w$ and $g'(w) \equiv 1$, so from the KKT condition

$$\widehat{\omega}_j = d_j + z_j^\top \widehat{\lambda} \quad (j \in S),$$

i.e., $r_j(\lambda) \equiv 0$ and $q_j \equiv 1$. Repeating the above steps without Taylor error gives

$$\widehat{\theta}_{\text{MEC}} = \widehat{\theta}_{\text{GREG}}.$$

This completes the proof.

## E. Proof of Theorem 5.1 (Asymptotic theory of cross-fitted PPI estimator)

**Lemma E.1** (Consistency of the CF-PPI estimator for the mean). *Let $(X, Y)$ have joint law $P$ with $X \sim P_X$ and $Y = m_0(X) + \varepsilon$, where $\mathbb{E}[\varepsilon \mid X] = 0$ and $\mathbb{E}[Y^2] < \infty$. The target parameter is the population mean, $\theta_0 := \mathbb{E}[Y]$. Let $\{X_i\}_{i=1}^N \overset{\text{i.i.d.}}{\sim} P_X$ be an unlabeled sample and $\{(X_j, Y_j)\}_{j \in S}$, $|S| = n$, be an independent labeled sample from $P$, with $N, n \to \infty$ and $n/N \to f \in (0, 1)$. For a chosen integer $K$ (typically $K = 5$ or 10), partition the labeled set $S$ into $K$ folds $S^{(1)}, \ldots, S^{(K)}$. For each $k \in \{1, \ldots, K\}$, fit $\widehat{m}^{(k)}$ using the labels in $S \setminus S^{(k)}$. Let $\kappa(i) \in \{1, \ldots, K\}$ denote the fold index of unit $i \in S$. Define the piecewise out-of-fold predictor*

$$\widehat{m}^{(-)}(X_i) := \begin{cases} \widehat{m}^{(\kappa(i))}(X_i), & i \in S, \\ \widehat{m}^\star(X_i), & i \notin S, \end{cases}$$

*where $\widehat{m}^\star$ is an aggregate model used for the plug-in term (e.g., the full-sample fit or the average $K^{-1} \sum_{k=1}^K \widehat{m}^{(k)}$). Consider the CF–PPI estimator*

$$\widehat{\theta}_{\text{PPI}}^{\text{cf}} = \frac{1}{N} \sum_{i=1}^N \widehat{m}^{(-)}(X_i) + \frac{1}{n} \sum_{j \in S} \{Y_j - \widehat{m}^{(-)}(X_j)\}.$$

*If the out–of–fold error is stochastically bounded in $L_2(P_X)$,*

$$\|\widehat{m}^{(-)} - m_0\|_{L_2(P_X)} = \left( \mathbb{E}\big[ (\widehat{m}^{(-)}(X) - m_0(X))^2 \big] \right)^{1/2} = O_p(1), \tag{38}$$

*then $\widehat{\theta}_{\text{PPI}}^{\text{cf}} \overset{p}{\to} \theta_0$.*

*Proof.* Using the notations of empirical process theory, the difference between the CF-PPI estimator and the target mean parameter can be written as follows:

$$\begin{aligned}
\widehat{\theta}_{\text{PPI}}^{\text{cf}} - \theta_0 &= \left\{ P_N \widehat{m}^{(-)} \right\} + \left\{ P_n \big( Y - \widehat{m}^{(-)} \big) \right\} - P\, m_0 \\
&\qquad \text{(since } \theta_0 = \mathbb{E}[Y] = PY = P\, m_0 \text{ as } Y = m_0 + \varepsilon \text{ and } \mathbb{E}[\varepsilon \mid X] = 0) \\
&= \underbrace{\big( P_N \widehat{m}^{(-)} - P_N m_0 \big)}_{A_1} + \underbrace{\big( P_n (Y - \widehat{m}^{(-)}) - P_n(Y - m_0) \big)}_{A_2} \\
&\quad + \underbrace{\big( P_N m_0 + P_n(Y - m_0) - P m_0 \big)}_{A_3} \\
&\qquad \text{(add and subtract } P_N m_0 \text{ and } P_n(Y - m_0)) \\
&= \underbrace{\big( P_N - P_n \big)\big( \widehat{m}^{(-)} - m_0 \big)}_{A_1 + A_2} + \underbrace{\big( P_N - P \big) m_0}_{\text{from the } P_N m_0 - P m_0 \text{ part}} + \underbrace{P_n(Y - m_0)}_{\text{remaining part of } A_3} \\
&= \big( P_N - P \big) m_0 + \underbrace{\big( P_n - P \big)(Y - m_0)}_{\text{since } P(Y - m_0) = 0} + \big( P_N - P_n \big)\big( \widehat{m}^{(-)} - m_0 \big) \\
&= \underbrace{\big( P_N - P \big) m_0}_{\substack{\text{unlabeled fluctuation} \\ (A)}} + \underbrace{\big( P_n - P \big)(Y - m_0)}_{\substack{\text{labeled residual fluctuation} \\ (B)}} + \underbrace{\big( P_N - P_n \big)\big( \widehat{m}^{(-)} - m_0 \big)}_{\substack{\text{nuisance remainder} \\ (C)}}. \tag{$\star$}
\end{aligned}$$

We briefly sketch the proof. The CF–PPI decomposition yields three pieces: the first two, $(A) = (P_N - P)\, m_0$ and $(B) = (P_n - P)\{Y - m_0\}$, are empirical–process terms with *fixed* indices because $m_0(x) = \mathbb{E}[Y \mid X = x]$ is nonrandom (oracle). Hence they fluctuate at the $O_p(N^{-1/2})$ and $O_p(n^{-1/2})$ scales (with $n/N \to f \in (0, 1)$, so $O_p(n^{-1/2}) = O_p(N^{-1/2})$), constitute the *first–order* part of the expansion, and require no Donsker/entropy control. The third piece, $(C) = (P_N - P_n)(\widehat{m} - m_0)$, would *without* cross–fitting typically require a Donsker condition for the nuisance class (see Section 4.2 of (Kennedy, 2016)); however, under cross–fitting the nuisance $\widehat{m} = \widehat{m}^{(-)}$ is trained on folds disjoint from the

evaluation sample, rendering it conditionally fixed and thus $o_p(N^{-1/2})$, i.e., only a *second–order* remainder. Intuitively, cross–fitting preserves design–unbiasedness of the score and pushes the learning error from $\widehat{m}$ out of the first–order limit.

Now, we are ready to analyze each of the three terms in $(\star)$ separately; this is central to the proof of the theorem and will also be used later in establishing asymptotic normality.

**Unlabeled fluctuation.** Note that the first term $(A) = (P_N - P)\, m_0$ is the empirical average of the fixed function $m_0$, and hence, by the CLT, it behaves asymptotically as a centered normal random variable with variance $\mathrm{Var}\{m_0(X)\}/N$, up to $o_p(N^{-1/2})$ error. More precisely, by the i.i.d. CLT, the unlabeled part follows:

$$\sqrt{N}\left\{(P_N - P)\, m_0\right\} \xrightarrow{d} \mathcal{N}\big(0, \mathrm{Var}\{m_0(X)\}\big), \qquad (P_N - P)\, m_0 = \frac{1}{\sqrt{N}}\, Z_1 + o_p(N^{-1/2}), \tag{39}$$

with $Z_1 \sim \mathcal{N}(0, \mathrm{Var}\{m_0(X)\})$. Because $\sqrt{N}\left\{(P_N - P)\, m_0\right\} = O_p(1)$, it holds $(A) = (P_N - P)m_0 = o_p(1)$. (One can use the weak law of large numbers directly for the same conclusion of $(P_N - P)m_0 = o_p(1)$.)

**Labeled residual fluctuation.** Next, the second term $(B) = (P_n - P)\,(Y - m_0)$ is the empirical fluctuation of the residuals $Y - m_0(X)$. Since the residuals have mean zero under the true distribution, this term also satisfies a CLT with variance $\mathrm{Var}\{Y - m_0(X)\}/n$. Like the first term, by CLT, the labeled residual part follows (note $P(Y - m_0) = 0$):

$$\sqrt{n}\left\{(P_n - P)\,(Y - m_0)\right\} \xrightarrow{d} \mathcal{N}\big(0, \mathrm{Var}\{Y - m_0(X)\}\big), \tag{40}$$

$$(P_n - P)\,(Y - m_0) = \frac{1}{\sqrt{n}}\, Z_2 + o_p(n^{-1/2}),$$

with $Z_2 \sim \mathcal{N}(0, \mathrm{Var}\{Y - m_0(X)\})$. Because $\sqrt{n}\left\{(P_n - P)\,(Y - m_0)\right\} = O_p(1)$, it holds $(B) = (P_n - P)\,(Y - m_0) = o_p(1)$.

**Nuisance remainder.** Finally, we now prove $(C) = (P_N - P_n)\big(\widehat{m}^{(-)} - m_0\big) = o_p(1)$. Let $\mathcal{T} := \sigma\big(\{\widehat{m}^{(k)}\}_{k=1}^{K}, \widehat{m}^{\star}, \kappa\big)$ denote the $\sigma$–field generated by the trained objects used to score observations (the $K$ fold–specific fits $\{\widehat{m}^{(k)}\}_{k=1}^{K}$, the aggregator $\widehat{m}^{\star}$, and the fold map $\kappa$). Define $\Delta(x) := \widehat{m}^{(-)}(x) - m_0(x)$. Conditional on $\mathcal{T}$, the function $\Delta$ is deterministic. Note that the remainder term in $(\star)$ can be further decomposed as

$$(C) = (P_N - P_n)\Delta = \underbrace{(P_N - P)\Delta}_{\substack{\text{Unlabeled average} \\ R_N}} - \underbrace{(P_n - P)\Delta}_{\substack{\text{Labeled average} \\ R_n}} \tag{41}$$

This remainder term $(C)$ in (41) captures the mismatch between the estimated regression function and the truth; for *consistency*, it suffices that $\|\widehat{m}^{(-)} - m_0\|_{L_2(P_X)} = O_p(1)$. Later for *asymptotic normality* we require $\|\widehat{m}^{(-)} - m_0\|_{L_2(P_X)} = o_p(1)$ so that $(C) = (P_N - P_n)\Delta = O_p(N^{-1/2}\|\Delta\|_{L_2(P_X)}) = o_p(N^{-1/2})$ and the first two fluctuation terms in $(\star)$ dominate.

The unlabeled average $R_N$ in the display above is benign: the unlabeled covariates are independent of the fitted functions, so it behaves like a standard empirical average of a fixed function (recall definition of $m^{(-)}(X)$). The potentially troublesome piece is the *labeled* average $R_n$. Without cross-fitting, the same labeled observations used to train $\widehat{m}$ would also be used to evaluate it, creating dependence and bias in this term. Cross-fitting restores *honesty*—each evaluation point is scored by a model trained on other folds—so that the empirical fluctuations admit the variance bounds used above without invoking restrictive entropy/Donsker conditions (Kennedy, 2024).

**(i) Unlabeled average $R_N$.** Since the unlabeled sample $\{X_i\}_{i=1}^{N}$ is independent of $\mathcal{T}$ (the fits are trained on labeled data only), $\{\Delta(X_i)\}_{i=1}^{N}$ are i.i.d. given $\mathcal{T}$ with mean $P\Delta := \mathbb{E}[\Delta(X) \mid \mathcal{T}]$. Writing $R_N$ in centered summation form,

$$R_N = \frac{1}{N} \sum_{i=1}^{N} \Big(\Delta(X_i) - P\Delta\Big),$$

we have, by independence,

$$\mathbb{E}[R_N \mid \mathcal{T}] = 0, \quad \mathrm{Var}(R_N \mid \mathcal{T}) = \frac{1}{N}\, \mathrm{Var}\big(\Delta(X) \mid \mathcal{T}\big) \leq \frac{1}{N}\mathbb{E}[\Delta(X)^2 \mid \mathcal{T}] = \frac{1}{N}\|\Delta\|_{L_2(P_X)}^2.$$

Now, we use the *Chebyshev's inequality*: For any random variable $Z$ with $\mathbb{E}[Z] = 0$,

$$\mathbb{P}(|Z| \geq t) \leq \frac{\mathrm{Var}(Z)}{t^2} \qquad \text{for all } t > 0.$$

Apply this to $Z = \sqrt{N}\, R_N$ (so that $\mathbb{E}[Z \mid \mathcal{T}] = 0$ and $\mathrm{Var}(Z \mid \mathcal{T}) = N\, \mathrm{Var}(R_N \mid \mathcal{T}) \leq \|\Delta\|^2_{L_2(P_X)}$). Then, for any $s > 0$,

$$\mathbb{P}\Big(|\sqrt{N}\, R_N| \geq s \,\Big|\, \mathcal{T}\Big) \leq \frac{\|\Delta\|^2_{L_2(P_X)}}{s^2}. \tag{42}$$

Taking expectations in both sides of (42) and using $\|\Delta\|_{L_2(P_X)} = O_p(1)$ (i.e., assumption (38)) yields $\sqrt{N}\, R_N = O_p(1)$, thus $R_N = O_p(N^{-1/2})$.

Note that we write $X_N = O_p(1)$ if, for every $\varepsilon > 0$, there exists a finite constant $M > 0$ such that $\mathbb{P}(|X_N| > M) < \varepsilon$ for all sufficiently large $N$. The upper bound $\|\Delta\|^2_{L_2(P_X)}/s^2$ in (42) can be made arbitrarily small by taking $s$ sufficiently large—smaller than any given $\varepsilon > 0$—and any such $s$ can be regarded as $M$; hence, $\sqrt{N}\, R_N = O_p(1)$. Henceforth, we omit this reasoning, as it is straightforward from the context once a bounding inequality of the form (42) is established.

**(ii) Labeled average $R_n$.** We provide a detailed illustration, as this term is the core part where cross-fitting is most essential for proving consistency (and also for proving asymptotic normality in Lemma E.3.).

Let the labeled indices be partitioned into $K$ folds $S = S_1 \cup \cdots \cup S_K$, and for each $k$ let $\widehat{m}^{(k)}$ be the predictor trained on the labeled data $\{(X_j, Y_j) : j \in S \setminus S_k\}$. Let $\kappa(j) = k$ denote the fold map if $j \in S_k$ and the cross–fitted predictor $\widehat{m}^{(-)}(x) := \widehat{m}^{(\kappa(j))}(x)$ when scoring index $j$.

*Decoupling by cross–fitting.* The term $R_n$ can be decomposed as

$$\begin{aligned}
R_n = (P_n - P)\Delta &= \frac{1}{n}\sum_{j \in S}\left\{\widehat{m}^{(-)}(X_j) - m_0(X_j)\right\} - P\Delta \\
&= \frac{1}{n}\sum_{j \in S}\widehat{m}^{(-)}(X_j) - \frac{1}{n}\sum_{j \in S} m_0(X_j) - P\Delta \\
&= \underbrace{\frac{1}{n}\sum_{j \in S}\widehat{m}^{(\kappa(j))}(X_j) - \frac{1}{n}\sum_{j \in S} m_0(X_j)}_{\text{Cross-fitted model fit}} - P\Delta,
\end{aligned} \tag{43}$$

where the first term in (43) is the average prediction from the cross-fitted models and the second term is the corresponding population regression truth evaluated on the labeled sample. Here, term $P\Delta$ are understood conditional on the model that scores each $j$ (i.e., the cross-fitted training set that excludes $j$); we keep the shorthand $P\Delta$ for readability.

This decomposition makes clear why cross-fitting is essential: it ensures that, for each $j \in S_k$, the model used to evaluate $X_j$ – namely $\widehat{m}^{(\kappa(j))}$ – is trained without the pair $(X_j, Y_j)$. Hence

$$\widehat{m}^{(\kappa(j))}(X_j) \perp\!\!\!\perp Y_j \mid X_j,$$

i.e., there is no label leakage. This conditional independence yields an unbiased expansion and clean conditional variance control, which establish consistency and enable the CLT for asymptotic normality later. Thus, the cross-fitted term behaves like an average of i.i.d. random variables. In particular, conditional on $\mathcal{T}$, the variables

$$\Delta(X_j) = \widehat{m}^{(-)}(X_j) - m_0(X_j), \qquad j \in S,$$

are i.i.d. with the same distribution as $\Delta(X)$, and they depend only on $X_j$ (since the fitted model never uses $Y_j$ in training). Refer to (Newey & Robins, 2018; Kennedy, 2024; Chernozhukov et al., 2018) for more details on similar ideas used in the development of cross-fitting and double machine learning methods.

*Conditional mean and variance.* Writing $W_j := \Delta(X_j) - P\Delta$ (so that $\mathbb{E}[W_j \mid \mathcal{T}] = 0$), we have

$$R_n = (P_n - P)\Delta = \frac{1}{n}\sum_{j \in S}(\Delta(X_j) - P\Delta) = \frac{1}{n}\sum_{j \in S} W_j, \quad \mathbb{E}[R_n \mid \mathcal{T}] = \frac{1}{n}\sum_{j \in S}\mathbb{E}[W_j \mid \mathcal{T}] = 0,$$

and, by conditional independence and identical distribution of the $W_j$'s,

$$\operatorname{Var}(R_n \mid \mathcal{T}) = \frac{1}{n^2} \sum_{j \in S} \operatorname{Var}(W_j \mid \mathcal{T}) = \frac{1}{n} \operatorname{Var}\big(\Delta(X) \mid \mathcal{T}\big) \le \frac{1}{n} \mathbb{E}[\Delta(X)^2 \mid \mathcal{T}] = \frac{1}{n} \|\Delta\|_{L_2(P_X)}^2.$$

Recall that Chebyshev's inequality in its conditional form states that, for any random variable $Z$ with $\mathbb{E}[Z \mid \mathcal{T}] = 0$, and for all $s > 0$, $\mathbb{P}(|Z| \ge s \mid \mathcal{T}) \le \operatorname{Var}(Z \mid \mathcal{T})/s^2$.

Apply this to $Z = \sqrt{n}\, R_n$ to obtain, for any $s > 0$,

$$\mathbb{P}\big(|\sqrt{n}\, R_n| \ge s \,\big|\, \mathcal{T}\big) \le \frac{\operatorname{Var}(\sqrt{n}\, R_n \mid \mathcal{T})}{s^2} = \frac{n \operatorname{Var}(R_n \mid \mathcal{T})}{s^2} \le \frac{\|\Delta\|_{L_2(P_X)}^2}{s^2}. \tag{44}$$

Taking expectations in both sides of (44) and using the stochastic boundedness $\|\Delta\|_{L_2(P_X)} = O_p(1)$ yields $\sqrt{n}\, R_n = O_p(1)$, i.e. $R_n = O_p(n^{-1/2})$.

One should note that, without cross–fitting, $\Delta$ would be measurable with respect to the same labeled data used in $P_n$, so the $W_j$'s would no longer be conditionally independent and centered given the training objects. Cross–fitting ensures that, conditional on $\mathcal{T}$, $W_j$ are i.i.d. mean-zero, allowing the variance bound and the Chebyshev control above to hold without further entropy/Donsker assumptions.

**(iii) Conclusion on the nuisance remainder.**   Combining the two parts, we have

$$(C) = (P_N - P_n)\Delta = R_N - R_n = O_p(N^{-1/2}) + O_p(n^{-1/2}) \xrightarrow{p} 0 \quad \text{as } N, n \to \infty.$$

**Conclusion.**   Each underbraced term in $(\star)$ converges to 0 in probability, so $\widehat{\theta}_{\mathrm{PPI}}^{\mathrm{cf}} \to_p \theta_0$. $\qquad\square$

**Lemma E.2** (Cross-fitted empirical-process bound; cf. Lemma 1 in (Kennedy, 2024)). *Let $\{W_i\}_{i=1}^n$ be i.i.d. from a distribution $P$, and let $\mathcal{T}$ be a $\sigma$–field independent of $\sigma(W_1, \ldots, W_n)$ (e.g., the $\sigma$–field generated by the training objects used to construct a cross–fitted predictor from data disjoint from $\{W_i\}_{i=1}^n$). For any $\mathcal{T}$–measurable function $h$ with $\|h\|_{L_2(P)}^2 = \mathbb{E}[h(W)^2] < \infty$, writing $P_n h := n^{-1} \sum_{i=1}^n h(W_i)$, we have*

$$\big(P_n - P\big)h \;=\; O_p\Big(\|h\|_{L_2(P)}/\sqrt{n}\Big).$$

*Equivalently, for every $t > 0$,*

$$\mathbb{P}\left(\frac{|(P_n - P)h|}{\|h\|_{L_2(P)}/\sqrt{n}} \ge t \,\bigg|\, \mathcal{T}\right) \le \frac{1}{t^2}, \qquad \text{hence} \qquad \frac{|(P_n - P)h|}{\|h\|_{L_2(P)}/\sqrt{n}} = O_p(1).$$

*Proof.* Independence implies $(W_1, \ldots, W_n) \mid \mathcal{T} \sim P^{\otimes n}$ a.s., so given $\mathcal{T}$ the $W_i$'s are i.i.d. with common law $P$. Since $h$ is $\mathcal{T}$–measurable, it is fixed when conditioning on $\mathcal{T}$. Thus,

$$\mathbb{E}[h(W_i) \mid \mathcal{T}] = \int h\, dP =: Ph, \qquad \mathbb{E}[h(W_i)^2 \mid \mathcal{T}] = \int h^2\, dP = \|h\|_{L_2(P)}^2,$$

and $\operatorname{Var}(h(W_i) \mid \mathcal{T}) \le \|h\|_{L_2(P)}^2$. With $P_n h := n^{-1} \sum_{i=1}^n h(W_i)$,

$$\mathbb{E}\big[(P_n - P)h \mid \mathcal{T}\big] = 0,$$

and conditional i.i.d.-ness yields

$$\operatorname{Var}\big((P_n - P)h \mid \mathcal{T}\big) = \operatorname{Var}\big(P_n h \mid \mathcal{T}\big) = \frac{1}{n} \operatorname{Var}\big(h(W) \mid \mathcal{T}\big) \le \frac{\|h\|_{L_2(P)}^2}{n}.$$

Let $Z := (P_n - P)h$. From the calculation above, $\mathbb{E}[Z \mid \mathcal{T}] = 0$ and $\mathrm{Var}(Z \mid \mathcal{T}) \leq \|h\|_{L_2(P)}^2/n$. Chebyshev's inequality in conditional form (obtained by applying conditional Markov inequality to $Z^2$):

$$\mathbb{P}(|Z| \geq a \mid \mathcal{T}) = \mathbb{P}(Z^2 \geq a^2 \mid \mathcal{T}) \leq \frac{\mathbb{E}[Z^2 \mid \mathcal{T}]}{a^2} = \frac{\mathrm{Var}(Z \mid \mathcal{T}) + (\mathbb{E}[Z \mid \mathcal{T}])^2}{a^2} = \frac{\mathrm{Var}(Z \mid \mathcal{T})}{a^2}.$$

Finally, choose $a := t \|h\|_{L_2(P)}/\sqrt{n}$ (with $t > 0$). Then

$$\mathbb{P}\left( |(P_n - P)h| \geq t \frac{\|h\|_{L_2(P)}}{\sqrt{n}} \,\Big|\, \mathcal{T} \right) \leq \frac{\mathrm{Var}(Z \mid \mathcal{T})}{t^2 \|h\|_{L_2(P)}^2/n} \leq \frac{1}{t^2}. \tag{$\dagger$}$$

If $\|h\|_{L_2(P)} = 0$, then $h = 0$ $P$–a.s., so $Z \equiv 0$ and the event on the left of ($\dagger$) has probability 0; the bound still holds. Taking expectations over $\mathcal{T}$ yields

$$\mathbb{P}\left( |(P_n - P)h| \geq t \frac{\|h\|_{L_2(P)}}{\sqrt{n}} \right) \leq \frac{1}{t^2},$$

which is equivalent to $(P_n - P)h = O_p(\|h\|_{L_2(P)}/\sqrt{n})$. $\qquad\square$

**Lemma E.3** (Asymptotic normality of the CF-PPI estimator for the mean). *Assume the setup of Lemma E.1. If, in addition, the cross–fitted predictor satisfies*

$$\|\widehat{m}^{(-)} - m_0\|_{L_2(P_X)} = o_p(1), \tag{45}$$

*then*

$$\sqrt{N} \left( \widehat{\theta}_{\mathrm{PPI}}^{\mathrm{cf}} - \theta_0 \right) \xrightarrow{d} \mathcal{N}(0, \sigma_f^2),$$

*with asymptotic variance*

$$\sigma_f^2 = \mathrm{Var}(m_0(X)) + \frac{1}{f} \mathrm{Var}(Y - m_0(X)).$$

*Proof.* We start from the decomposition ($\star$) in *Proof* of Lemma E.1 and multiply by $\sqrt{N}$:

$$\sqrt{N}(\widehat{\theta}_{\mathrm{PPI}}^{\mathrm{cf}} - \theta_0) = \underbrace{\sqrt{N}(P_N - P)m_0}_{\text{(I)}} + \underbrace{\sqrt{N}(P_n - P)(Y - m_0)}_{\text{(II)}} + \underbrace{\sqrt{N}(P_N - P_n)(\widehat{m}^{(-)} - m_0)}_{\text{(III)}}.$$

**Leading terms (I) and (II).** By the i.i.d. CLT, (39) gives

$$\sqrt{N}(P_N - P)m_0 \xrightarrow{d} \mathcal{N}(0, \mathrm{Var}\{m_0(X)\}).$$

Similarly, (40) yields

$$\sqrt{n}(P_n - P)(Y - m_0) \xrightarrow{d} \mathcal{N}(0, \mathrm{Var}\{Y - m_0(X)\}),$$

so

$$\sqrt{N}(P_n - P)(Y - m_0) = \sqrt{\frac{N}{n}} \sqrt{n}(P_n - P)(Y - m_0) \xrightarrow{d} \mathcal{N}\left(0, \frac{1}{f} \mathrm{Var}\{Y - m_0(X)\}\right),$$

because $N/n \to 1/f$. Since the unlabeled and labeled samples are independent, the limits above are independent.

**Remainder term (III).** Let $\Delta := \widehat{m}^{(-)} - m_0$. From the decomposition,

$$\text{(III)} = \sqrt{N}(P_N - P_n)\Delta = \sqrt{N}\left\{(P_N - P)\Delta - (P_n - P)\Delta\right\} = \sqrt{N} R_N - \sqrt{N} R_n,$$

where $R_N := (P_N - P)\Delta$ and $R_n := (P_n - P)\Delta$.

Apply Lemma E.2 with $h = \Delta$ to the two evaluation samples: (i) the unlabeled sample $\{X_i\}_{i=1}^N$ (take $W = X$, sample size $n = N$), and (ii) the labeled evaluation sample $\{X_j\}_{j \in S}$ (cross–fitted, so $X_j \perp\!\!\!\perp \mathcal{T}$, sample size $n$). The lemma gives

$$R_N = O_p\big(\|\Delta\|_{L_2(P_X)}/\sqrt{N}\big), \qquad R_n = O_p\big(\|\Delta\|_{L_2(P_X)}/\sqrt{n}\big).$$

Hence

$$\text{(III)} = \sqrt{N}\, R_N - \sqrt{N}\, R_n = O_p\big(\|\Delta\|_{L_2(P_X)}\big) \; + \; O_p\Big(\sqrt{\tfrac{N}{n}}\, \|\Delta\|_{L_2(P_X)}\Big) = O_p\big(\|\Delta\|_{L_2(P_X)}\big),$$

since $n/N \to f \in (0,1)$ implies $\sqrt{N/n} = O(1)$. Therefore, under the condition (45), we have

$$\text{(III)} = \sqrt{N}\, (P_N - P_n)\big(\widehat{m}^{(-)} - m_0\big) = o_p(1).$$

This controls the nuisance remainder at the $\sqrt{N}$ scale and completes the treatment of term (III).

**Conclusion.** By Slutsky's theorem and independence of the two leading limits,

$$\sqrt{N}\big(\widehat{\theta}_{\text{PPI}}^{\text{cf}} - \theta_0\big) \;\xrightarrow{d}\; \mathcal{N}\Big(0,\; \text{Var}\{m_0(X)\} + \tfrac{1}{f}\, \text{Var}\{Y - m_0(X)\}\Big) = \mathcal{N}(0, \sigma_f^2),$$

which is the claimed result. $\qquad\square$

**Theorem E.4** (Consistency and asymptotic normality of the CF-PPI estimator for the mean). *Let $(X, Y)$ have joint law $P$ with $X \sim P_X$ and $Y = m_0(X) + \varepsilon$, where $\mathbb{E}[\varepsilon \mid X] = 0$ and $\mathbb{E}[Y^2] < \infty$. The target is the population mean $\theta_0 := \mathbb{E}[Y]$. Let $\{X_i\}_{i=1}^N \overset{\text{i.i.d.}}{\sim} P_X$ be an unlabeled sample and $\{(X_j, Y_j)\}_{j \in S}$, $|S| = n$, an independent labeled sample from $P$, with $N, n \to \infty$ and $n/N \to f \in (0,1)$. Let $\widehat{m}^{(-)}$ be the cross–fitted predictor (each labeled index is scored by a model trained without its own fold), and consider*

$$\widehat{\theta}_{\text{PPI}}^{\text{cf}} = \frac{1}{N} \sum_{i=1}^N \widehat{m}^{(-)}(X_i) + \frac{1}{n} \sum_{j \in S} \{Y_j - \widehat{m}^{(-)}(X_j)\}. \tag{46}$$

*Then:*

**(i) Consistency.** *If the out–of–fold error is stochastically bounded in $L_2(P_X)$,*

$$\|\widehat{m}^{(-)} - m_0\|_{L_2(P_X)} = O_p(1),$$

*then $\widehat{\theta}_{\text{PPI}}^{\text{cf}} \xrightarrow{p} \theta_0$.*

**(ii) Asymptotic normality.** *If, in addition, the cross–fitted predictor is $L_2(P_X)$–consistent,*

$$\|\widehat{m}^{(-)} - m_0\|_{L_2(P_X)} = o_p(1),$$

*then*

$$\sqrt{N}\,\big(\widehat{\theta}_{\text{PPI}}^{\text{cf}} - \theta_0\big) \xrightarrow{d} \mathcal{N}(0, \sigma_f^2), \qquad \sigma_f^2 \;=\; \text{Var}\big(m_0(X)\big) \;+\; \frac{1}{f}\, \text{Var}\big(Y - m_0(X)\big).$$

*Proof.* The theorem follows from Lemma E.1 (consistency) together with the Lemma E.3 (asymptotic normality). $\qquad\square$

# F. Proof of Theorem 5.2 (Asymptotic theory of the MEC estimator)

**Lemma F.1** (Curvature–weighted divergence control). *Let $G : \mathcal{V} \to \mathbb{R}$ be strictly convex and twice continuously differentiable with derivative $g := G'$. For any vectors $\omega = (\omega_j)_{j \in S}$ and $d = (d_j)_{j \in S}$,*

$$\sum_{j \in S} \tilde{g}_j'\, (\omega_j - d_j)^2 \;=\; D_G(\omega \| d) + D_G(d \| \omega),$$

*where the Bregman divergence is*

$$D_G(a\|b) = \sum_{j \in S} \left\{ G(a_j) - G(b_j) - g(b_j)(a_j - b_j) \right\},$$

*and*

$$\tilde{g}'_j := \tilde{g}'(d_j, \omega_j) := \int_0^1 g'(d_j + t(\omega_j - d_j))\, dt.$$

*Proof.* Expand the two Bregman divergences coordinatewise:

$$D_G(\omega\|d) = \sum_{j \in S} \left\{ G(\omega_j) - G(d_j) - g(d_j)(\omega_j - d_j) \right\},$$

$$D_G(d\|\omega) = \sum_{j \in S} \left\{ G(d_j) - G(\omega_j) - g(\omega_j)(d_j - \omega_j) \right\}.$$

Adding them and simplifying gives the symmetric Bregman identity

$$D_G(\omega\|d) + D_G(d\|\omega) = \sum_{j \in S} (\omega_j - d_j)\left(g(\omega_j) - g(d_j)\right). \tag{47}$$

For each $j \in S$, apply the fundamental theorem of calculus to $g = G'$ along the segment from $d_j$ to $\omega_j$:

$$g(\omega_j) - g(d_j) = \int_{d_j}^{\omega_j} g'(s)\, ds = \int_0^1 g'(d_j + t(\omega_j - d_j))(\omega_j - d_j)\, dt$$

$$= (\omega_j - d_j) \int_0^1 g'(d_j + t(\omega_j - d_j))\, dt = (\omega_j - d_j)\, \tilde{g}'_j.$$

where $\tilde{g}'_j := \int_0^1 g'(d_j + t(\omega_j - d_j))\, dt$. Here, we used the change of variable $s = d_j + t(\omega_j - d_j)$, so $ds = (\omega_j - d_j)\, dt$, for the second equality.

Substituting this expression for $g(\omega_j) - g(d_j)$ into (47) yields

$$D_G(\omega\|d) + D_G(d\|\omega) = \sum_{j \in S} (\omega_j - d_j)(\omega_j - d_j)\, \tilde{g}'_j = \sum_{j \in S} \tilde{g}'_j\, (\omega_j - d_j)^2,$$

which is exactly the claimed identity. $\qquad\square$

**Lemma F.2** (Consistency of the MEC estimator)**.** *Assume the conditions of Lemma E.1. Fix baseline weights $d = (d_j)_{j \in S}$ (i.e., $d_j \equiv N/n$) and a strictly convex, twice continuously differentiable generator $G$ with derivative $g = G'$ and Bregman divergence $D_G(\cdot\|\cdot)$. Let $h(x) = (1, \widehat{m}^{(-)}(x))^\top$, $z_j = h(X_j)$, and define the calibration subspace*

$$W := \mathrm{span}\{h\} = \mathrm{span}\{1, \widehat{m}^{(-)}(\cdot)\} = \{a + b\,\widehat{m}^{(-)}(\cdot) : a, b \in \mathbb{R}\} \subset L_2(P_X).$$

*Let $\Pi_W$ denote the $L_2(P_X)$ projection onto $W$, and set $m_{0,\perp} := m_0 - \Pi_W m_0$. Let $\widehat{\omega} = (\widehat{\omega}_j)_{j \in S}$ be the calibrated weights obtained by the G–Bregman projection of $d$ onto the calibration constraints $Z^\top \omega = \mu$ with $Z = [z_j^\top]_{j \in S}$ and $\mu = \sum_{i=1}^N h(X_i)$. Consider the MEC estimator*

$$\widehat{\theta}_{\mathrm{MEC}} = \frac{1}{N} \sum_{j \in S} \widehat{\omega}_j\, Y_j.$$

***Assumptions.***

*A1 Moments:. $\mathbb{E}\|h(X)\|^2 < \infty$ (equivalently, $\mathbb{E}[\widehat{m}^{(-)}(X)^2] < \infty$).*

*A2 Weights stability: The calibrated weights satisfy the symmetric Bregman bound*

$$D_G(\widehat{\omega}\|d) + D_G(d\|\widehat{\omega}) = O_p(1).$$

*Then:*
*If* A1–A2 *hold and the projection error is stochastically bounded in* $L_2(P_X)$,

$$\|m_{0,\perp}\|_{L_2(P_X)} = \|m_0 - \Pi_W m_0\|_{L_2(P_X)} = O_p(1),$$

*then*

$$\widehat{\theta}_{\text{MEC}} \xrightarrow{p} \theta_0.$$

*Proof.* By the empirical process notation, we can write

$$\widehat{\theta}_{\text{MEC}} - \theta_0 = \frac{1}{N}\sum_{j\in S}\widehat{\omega}_j Y_j - P m_0$$

$$= \frac{1}{N}\sum_{j\in S}\widehat{\omega}_j Y_j + \frac{1}{N}\left(\sum_{i=1}^N \widehat{m}^{(-)}(X_i) - \sum_{j\in S}\widehat{\omega}_j\widehat{m}^{(-)}(X_j)\right) - P m_0$$

$$= \left\{P_N\widehat{m}^{(-)}\right\} + \left\{P_n^{\widehat{\omega}}(Y - \widehat{m}^{(-)})\right\} - P m_0, \tag{48}$$

where the bracketed term is zero by the calibration balance constraint $P_N\widehat{m}^{(-)} = P_n^{\widehat{\omega}}\widehat{m}^{(-)}$ (i.e., $\sum_{i=1}^N \widehat{m}^{(-)}(X_i) = \sum_{j\in S}\widehat{\omega}_j\widehat{m}^{(-)}(X_j)$).

In what follows, to avoid notational clutter, we write $P_n^\omega$ in place of $P_n^{\widehat{\omega}}$. Since we work throughout with the *calibrated* weights $\widehat{\omega}$ to construct the MEC estimator $\widehat{\theta}_{\text{MEC}} = (1/N)\sum_{j\in S}\widehat{\omega}_j Y_j$, the intended meaning will be clear from the context.

Now, we decompose (48) as follow:

$$\widehat{\theta}_{\text{MEC}} - \theta_0 = \left\{P_N\widehat{m}^{(-)}\right\} + \left\{P_n^\omega(Y - \widehat{m}^{(-)})\right\} - P m_0$$

$$= \left(P_N\widehat{m}^{(-)} - P_N m_0\right) + \left(P_n^d(Y - \widehat{m}^{(-)}) - P_n^d(Y - m_0)\right)$$

$$\quad + \left(P_N m_0 + P_n^d(Y - m_0) - P m_0\right) + \left(P_n^\omega - P_n^d\right)(Y - \widehat{m}^{(-)})$$

$$\text{(add and subtract } P_N m_0 \text{ and } P_n^d(Y - m_0); \text{ split } P_n^\omega(Y - \widehat{m}^{(-)}))$$

$$= \left(P_N\widehat{m}^{(-)} - P_N m_0 - P_n^d(\widehat{m}^{(-)} - m_0)\right) + \left(P_N m_0 + P_n^d(Y - m_0) - P m_0\right)$$

$$\quad + \left(P_n^\omega - P_n^d\right)(Y - \widehat{m}^{(-)})$$

$$= (P_N - P_n^d)(\widehat{m}^{(-)} - m_0) + (P_N - P)m_0 + (P_n^d - P)(Y - m_0)$$

$$\quad + \left(P_n^\omega - P_n^d\right)(Y - \widehat{m}^{(-)})$$

$$= \underbrace{(P_N - P)m_0}_{\substack{\text{unlabeled fluctuation}\\(A)}} + \underbrace{(P_n - P)(Y - m_0)}_{\substack{\text{labeled residual fluctuation}\\(B)}} + \underbrace{(P_N - P_n)(\widehat{m}^{(-)} - m_0)}_{\substack{\text{nuisance remainder}\\(C)}} \tag{49}$$

$$\quad + \underbrace{(P_n^\omega - P_n)(Y - m_0)}_{\substack{\text{WC residual}\\(D)}} - \underbrace{(P_n^\omega - P_n)(\widehat{m}^{(-)} - m_0)}_{\substack{\text{WC nuisance correction}\\(E)}},$$

where the last equality, we used the empirical-process notation, where $P_n^d = P_n$ by definition since $d_j \equiv N/n$.

Note that the terms $(A) + (B) + (C)$ in (49) are exactly the decomposition of $\widehat{\theta}_{\text{PPI}}^{\text{cf}} - \theta_0$ (see the proof of Theorem 5.1).

We briefly interpret the roles of the additional terms $(D)$ and $(E)$ in the MEC decomposition (49). They arise from weight calibration. First, $(P_n^\omega - P_n)$ can be viewed as a *reweighting (rebalancing) empirical operator*: it replaces the design

measure $P_n$ by the calibrated measure $P_n^\omega$. In particular, when $\omega = d$, this operator vanishes; consequently, $(D)$ and $(E)$ are zero, and the decomposition of the MEC estimator coincides with that of CF–PPI.

The term

$$(D) = (P_n^\omega - P_n)(Y - m_0) = \frac{1}{n} \sum_{j \in S} (\widehat{\omega}_j - d_j)(Y_j - m_0(X_j)),$$

measures the effect of reweighting the labeled *residuals*—that is, how each labeled unit's contribution is altered so that the weighted labeled sample better matches the unlabeled population along the calibration moments.

The term

$$(E) = (P_n^\omega - P_n)(\widehat{m}^{(-)} - m_0) = \frac{1}{n} \sum_{j \in S} (\widehat{\omega}_j - d_j)(\widehat{m}^{(-)}(X_j) - m_0(X_j)),$$

is the corresponding *prediction adjustment*: it corrects the part of the estimator that depends on the fitted regression so that, after reweighting, predictions remain aligned with the balanced totals. In short, $(D)$ rebalances residuals and $(E)$ rebalances predictions.

We note that the algebraic decomposition $(A) + (B) + (C) + (D) + (E)$ (49) holds for any choice of weights $\omega = (\omega_j)_{j \in S}$ and any working span $W = \mathrm{span}(h)$.

The benefit of *calibration* with the predictor basis $h = (1, \widehat{m}^{(-)})$, is the balancing condition

$$P_N v = P_n^\omega v, \qquad \text{for all } v \in W, \tag{50}$$

where $W$ is the associated calibration span of predictor basis

$$W = \mathrm{span}\{1, \widehat{m}^{(-)}(\cdot)\} = \{ a + b\,\widehat{m}^{(-)}(\cdot) : a, b \in \mathbb{R} \} \subset L_2(P_X).$$

The term $(C) - (E)$ can be simplified via the balance conditions (50):

$$
\begin{aligned}
(C) - (E) &= (P_N - P_n)(\widehat{m}^{(-)} - m_0) - (P_n^\omega - P_n)(\widehat{m}^{(-)} - m_0) \\
&= (P_N - P_n^\omega)(\widehat{m}^{(-)} - m_0) \\
&= (P_N - P_n^\omega)\{\widehat{m}^{(-)} - \Pi_W m_0 - m_{0,\perp}\} \qquad (m_0 := \Pi_W m_0 + m_{0,\perp}) \\
&= \underbrace{(P_N - P_n^\omega)(\widehat{m}^{(-)} - \Pi_W m_0)}_{=0} - (P_N - P_n^\omega) m_{0,\perp} \\
&= -(P_N - P_n^\omega) m_{0,\perp} \\
&= (P_n^\omega - P_N) m_{0,\perp},
\end{aligned}
\tag{51}
$$

where the second equality holds since intermediate $P_n$-terms cancel, and the equality in (51) holds since $\widehat{m}^{(-)} - m_0 = (\widehat{m}^{(-)} - \Pi_W m_0) - m_{0,\perp}$ with $m_{0,\perp} := m_0 - \Pi_W m_0$ and noting that $v = \widehat{m}^{(-)} - \Pi_W m_0 \in W$.

By this simplification, the out-of-fold predictor $\widehat{m}^{(-)}$ (which is noisy and subtle to handle) appears only through the calibration span via the remainder $m_{0,\perp} := m_0 - \Pi_W m_0$, which is orthogonal to $W$. Thus the algebraic decomposition (49) becomes

$$\widehat{\theta}_{\mathrm{MEC}} - \theta_0 = \underbrace{(P_N - P)m_0}_{(A)} + \underbrace{(P_n - P)(Y - m_0)}_{(B)} + \underbrace{(P_n^\omega - P_N) m_{0,\perp}}_{(C)-(E)} + \underbrace{(P_n^\omega - P_n)(Y - m_0)}_{(D)}. \tag{52}$$

Now we use the decomposition (52) to prove consistency. Later, this decomposition will be also used for proving asymptotic normality in Lemma F.4.

**Terms $(A)$ and $(B)$.** It is straightforward that terms (A) and (B) are $o_p(1)$: by the law of large numbers, $(P_N - P)m_0 \to 0$ and $(P_n - P)(Y - m_0) \to 0$ under Assumption A1.

**Term $(C) - (E)$.**   Recall from (51) that

$$(C) - (E) = (P_n^\omega - P_N)m_{0,\perp} = \underbrace{(P_n^\omega - P_n)m_{0,\perp}}_{\text{(i)}} + \underbrace{(P_n - P)m_{0,\perp}}_{\text{(ii)}} - \underbrace{(P_N - P)m_{0,\perp}}_{\text{(iii)}}.$$

Since $m_{0,\perp} := m_0 - \Pi_W m_0$ is the $L_2(P_X)$–orthogonal residual to $W = \mathrm{span}\{1, \widehat{m}^{(-)}\}$, we have $P m_{0,\perp} = 0$ (because $1 \in W$; i.e., $\langle m_{0,\perp}, h \rangle_{L_2(P_X)} = \int m_{0,\perp}(x) h(x) \, dP_X(x) = 0$ for all $h \in W$; thus, taking $h \equiv 1$ gives $\int m_{0,\perp}(x) \, dP_X(x) = 0$). Let $\mathcal{T}$ be the sigma–field generated by the training procedure producing $\widehat{m}^{(-)}$. Then $W$, $\Pi_W m_0$, and $m_{0,\perp}$ are $\mathcal{T}$–measurable; conditionally on $\mathcal{T}$, $m_{0,\perp} \in L_2(P_X)$ is fixed with mean zero. Hence, by the (conditional) LLN/CLT,

$$\text{(ii)} = (P_n - P)m_{0,\perp} = O_p(n^{-1/2}) \qquad \text{and} \qquad \text{(iii)} = (P_N - P)m_{0,\perp} = O_p(N^{-1/2}),$$

and these rates also hold unconditionally.

By Cauchy–Schwarz with curvature weights,

$$|\text{(i)}| = \left| \frac{1}{n} \sum_{j \in S} (\widehat{\omega}_j - d_j) \, m_{0,\perp}(X_j) \right| \leq \frac{1}{n} \left( \sum_{j \in S} \tilde{g}'_j (\widehat{\omega}_j - d_j)^2 \right)^{1/2} \left( \sum_{j \in S} \frac{m_{0,\perp}(X_j)^2}{\tilde{g}'_j} \right)^{1/2}, \tag{53}$$

where we used the weighted Cauchy–Schwarz inequality with weights

$$\tilde{g}'_j := \tilde{g}'(d_j, \widehat{\omega}_j) := \int_0^1 g'\big(d_j + t(\widehat{\omega}_j - d_j)\big) \, dt > 0:$$

$$\left| \sum_j u_j v_j \right| \leq \left( \sum_j w_j u_j^2 \right)^{1/2} \left( \sum_j \frac{v_j^2}{w_j} \right)^{1/2}, \quad u_j = \omega_j - d_j, \; v_j = m_{0,\perp}(X_j), \; w_j = \tilde{g}'_j.$$

Now note the orders of each factor of the upper bound in (53):

By Lemma F.1 and A2,

$$\sum_{j \in S} \tilde{g}'_j (\widehat{\omega}_j - d_j)^2 = \{D_G(\widehat{\omega} \| d) + D_G(d \| \widehat{\omega})\} = O_p(1).$$

By A1, $m_{0,\perp} \in L_2(P_X)$, so $\mathbb{E}[m_{0,\perp}(X)^2] < \infty$ and, conditionally on $\mathcal{T}$,

$$\frac{1}{n} \sum_{j \in S} m_{0,\perp}(X_j)^2 \xrightarrow{p} \mathbb{E}[m_{0,\perp}(X)^2] \quad \Rightarrow \quad \frac{1}{n} \sum_{j \in S} m_{0,\perp}(X_j)^2 = O_p(1).$$

Since $g'$ is continuous and the calibration solution lies in a feasible (hence tight/compact) region, $\max_{j \in S} \{1/\tilde{g}'_j\} = O_p(1)$. Therefore,

$$\frac{1}{n} \sum_{j \in S} \frac{m_{0,\perp}(X_j)^2}{\tilde{g}'_j} \leq \left( \max_{j \in S} \frac{1}{\tilde{g}'_j} \right) \frac{1}{n} \sum_{j \in S} m_{0,\perp}(X_j)^2 = O_p(1) \cdot O_p(1) = O_p(1).$$

Consequently,

$$\left( \sum_{j \in S} \frac{m_{0,\perp}(X_j)^2}{\tilde{g}'_j} \right)^{1/2} = O_p(\sqrt{n}).$$

Putting the pieces together,

$$|\text{(i)}| \leq \frac{1}{n} O_p(1) \, O_p(\sqrt{n}) = O_p(n^{-1/2}).$$

Hence, with (ii) $= O_p(n^{-1/2})$ and (iii) $= O_p(N^{-1/2})$, we obtain

$$(C) - (E) = o_p(1).$$

**Term** $(D)$. Recall that

$$(D) = (P_n^\omega - P_n)(Y - m_0) = \frac{1}{n}\sum_{j \in S}(\widehat{\omega}_j - d_j)\{Y_j - m_0(X_j)\} = \frac{1}{n}\sum_{j \in S}(\widehat{\omega}_j - d_j)\,\varepsilon_j,$$

where $\varepsilon_j := Y_j - m_0(X_j)$ satisfies $\mathbb{E}[\varepsilon_j \mid X_j] = 0$ and $\mathbb{E}[\varepsilon_j^2] < \infty$ (from A1/Lemma E.1). Conditioning on training/calibration data (denoted as $\mathcal{G}$ for simplicity), so that $\widehat{\omega}_j$ is fixed and does not depend on its own label $Y_j$, we have

$$\mathbb{E}[(D) \mid \mathcal{G}, \{X_j\}_{j \in S}] = \frac{1}{n}\sum_{j \in S}(\widehat{\omega}_j - d_j)\,\mathbb{E}[\varepsilon_j \mid \mathcal{G}, X_j] = \frac{1}{n}\sum_{j \in S}(\widehat{\omega}_j - d_j)\,\mathbb{E}[\varepsilon_j \mid X_j] = 0.$$

Thus, the marginal expectation of $(D)$ is zero as well (i.e., $\mathbb{E}[(D)] = 0$) by the law of iterated expectations.

By weighted Cauchy–Schwarz with positive weights $w_j = \tilde{g}_j' > 0$,

$$|(D)| = \left|\frac{1}{n}\sum_{j \in S}(\widehat{\omega}_j - d_j)\,\varepsilon_j\right| \le \frac{1}{n}\left(\sum_{j \in S}\tilde{g}_j'(\widehat{\omega}_j - d_j)^2\right)^{1/2}\left(\sum_{j \in S}\frac{\varepsilon_j^2}{\tilde{g}_j'}\right)^{1/2}. \tag{54}$$

By Lemma F.1 and A2 (weight stability),

$$\sum_{j \in S}\tilde{g}_j'(\widehat{\omega}_j - d_j)^2 = \{D_G(\widehat{\omega}\|d) + D_G(d\|\widehat{\omega})\} = O_p(1),$$

so the first square root in (54) is $O_p(1)$.

Since $\mathbb{E}[\varepsilon^2] < \infty$, the LLN gives

$$\frac{1}{n}\sum_{j \in S}\varepsilon_j^2 = O_p(1).$$

With $g'$ continuous and the calibrated solution lying in a feasible (hence tight/compact) region,

$$\max_{j \in S}\left\{\frac{1}{\tilde{g}_j'}\right\} = O_p(1) \quad \Rightarrow \quad \frac{1}{n}\sum_{j \in S}\frac{\varepsilon_j^2}{\tilde{g}_j'} \le \left(\max_{j \in S}\frac{1}{\tilde{g}_j'}\right)\frac{1}{n}\sum_{j \in S}\varepsilon_j^2 = O_p(1),$$

so the second square root in (54) is $O_p(\sqrt{n})$.

Therefore,

$$|(D)| \le \frac{1}{n}O_p(1)\,O_p(\sqrt{n}) = O_p(n^{-1/2}) = O_p(N^{-1/2}),$$

since $n/N \to f \in (0,1)$. Hence the WC residual term $(D)$ is $o_p(1)$.

**Conclusion.** Collecting the bounds established above, the terms $(A)$, $(B)$, $(C) - (E)$, and $(D)$ in the decomposition (52) are each $o_p(1)$. Therefore,

$$\widehat{\theta}_{\mathrm{MEC}} - \theta_0 = (A) + (B) + \{(C) - (E)\} + (D) = o_p(1),$$

so $\widehat{\theta}_{\mathrm{MEC}} \xrightarrow{p} \theta_0$. $\qquad\square$

**Lemma F.3** (Regularity linking weight stability and small dual). *Let $G$ be twice continuously differentiable. Assume that*

$$A_n = \frac{1}{n}\sum_{j \in S}q_j z_j z_j^\top \xrightarrow{p} \Sigma_q \succ 0, \qquad q_j = \frac{1}{g'(d_j)}, \quad z_j = h(X_j) = (1, m^{(-)}(X_j)).$$

*Let $\widehat{\lambda}$ be the dual optimizer of the calibration program and define $\widehat{\omega}_j = g^{-1}\big(g(d_j) + z_j^\top\widehat{\lambda}\big)$. Then*

$$\|\widehat{\lambda}\| = O_p(n^{-1/2}) \quad \Longrightarrow \quad D_G(\widehat{\omega}\|d) + D_G(d\|\widehat{\omega}) = O_p(1).$$

*Proof.* By the symmetric–Bregman identity (Lemma F.1),

$$D_G(\omega\|d) + D_G(d\|\omega) = \sum_{j\in S} \tilde{g}'_j \, (\omega_j - d_j)^2, \quad \tilde{g}'_j = \int_0^1 g'(d_j + t(\omega_j - d_j))\, dt > 0.$$

From the KKT relation $g(\widehat{\omega}_j) = g(d_j) + z_j^\top \widehat{\lambda}$ and a second-order Taylor expansion of $g^{-1}$ at $g(d_j)$,

$$\widehat{\omega}_j - d_j \;=\; q_j\, z_j^\top \widehat{\lambda} \;+\; r_j(\widehat{\lambda}), \qquad q_j := \frac{1}{g'(d_j)}, \qquad |r_j(\widehat{\lambda})| \le C\, \|\widehat{\lambda}\|^2, \tag{55}$$

where $C < \infty$ by local boundedness of $(g^{-1})''$. The boundedness of $g'$ implies $q_j$ and $\tilde{g}'_j$ are $O_p(1)$.

Using $(a+b)^2 \le 2a^2 + 2b^2$ and the boundedness of $q_j$ and $\tilde{g}'_j$, we obtain

$$\begin{aligned}
D_G(\widehat{\omega}\|d) + D_G(d\|\widehat{\omega}) = \sum_{j\in S} \tilde{g}'_j \big(q_j\, z_j^\top \widehat{\lambda} + r_j(\widehat{\lambda})\big)^2 &\le 2\sum_{j\in S} \tilde{g}'_j\, q_j^2\, (z_j^\top \widehat{\lambda})^2 \;+\; 2\sum_{j\in S} \tilde{g}'_j\, r_j(\widehat{\lambda})^2 \\
&\le c_1 \sum_{j\in S} q_j\, (z_j^\top \widehat{\lambda})^2 \;+\; c_2\, n\, \|\widehat{\lambda}\|^4 = c_1\, \widehat{\lambda}^\top \Big(\sum_{j\in S} q_j z_j z_j^\top\Big)\widehat{\lambda} \;+\; c_2\, n\, \|\widehat{\lambda}\|^4 \\
&= c_1\, n\, \widehat{\lambda}^\top \Big(\tfrac{1}{n}\sum_{j\in S} q_j z_j z_j^\top\Big)\widehat{\lambda} \;+\; c_2\, n\, \|\widehat{\lambda}\|^4, \tag{$\star$}
\end{aligned}$$

for some finite constants $c_1, c_2 > 0$. Since $A_n := n^{-1}\sum_{j\in S} q_j z_j z_j^\top \xrightarrow{p} \Sigma_q \succ 0$, the first term in $(\star)$ is $n\, O_p(1)\, \|\widehat{\lambda}\|^2$. If $\|\widehat{\lambda}\| = O_p(n^{-1/2})$, then $n\|\widehat{\lambda}\|^2 = O_p(1)$ and $n\|\widehat{\lambda}\|^4 = O_p(n^{-1}) = o_p(1)$, hence

$$D_G(\widehat{\omega}\|d) + D_G(d\|\widehat{\omega}) = O_p(1).$$

$\square$

**Lemma F.4** (Asymptotic normality of the MEC estimator). *Assume the conditions of Lemma E.1 and Lemma F.2. Consider the MEC estimator*

$$\widehat{\theta}_{\mathrm{MEC}} = \frac{1}{N}\sum_{j\in S} \widehat{\omega}_j\, Y_j.$$

*Assumptions.*

*A1 Moments:.* $\mathbb{E}\|h(X)\|^2 < \infty$ *(equivalently, $\mathbb{E}[\widehat{m}^{(-)}(X)^2] < \infty$).*

*A2 Weights stability: The calibrated weights satisfy the symmetric Bregman bound*

$$D_G(\widehat{\omega}\|d) \;+\; D_G(d\|\widehat{\omega}) \;=\; O_p(1).$$

*A3 Weighted Gram limit: With*

$$A_n := \frac{1}{n}\sum_{j\in S} q_j\, z_j z_j^\top, \qquad q_j := \frac{1}{g'(d_j)},$$

*one has $A_n \xrightarrow{P} \Sigma_q \succ 0$.*

*A4 Small dual. Let $\widehat{\lambda}$ be the dual optimizer of the calibration program with $\|\widehat{\lambda}\| = O_p(n^{-1/2})$.*

*Then:*
*If A1–A4 hold and the projection error is $L_2(P_X)$-consistent,*

$$\|m_{0,\perp}\|_{L_2(P_X)} = \|m_0 - \Pi_W m_0\|_{L_2(P_X)} = o_P(1),$$

*then*

$$\sqrt{N}\,\big(\widehat{\theta}_{\mathrm{MEC}} - \theta_0\big) \xrightarrow{d} \mathcal{N}\big(0,\, \sigma_f^2\big), \qquad \sigma_f^2 = \mathrm{Var}\big(m_0(X)\big) + \frac{1}{f}\,\mathrm{Var}\big(Y - m_0(X)\big).$$

(Remark on Assumption A2. Assumption A2 follows directly from A3 and A4 (Lemma F.3). Hence, A2 need not be stated separately; we retain it for readability.)

*Proof.* We start from the decomposition used for proof of consistency in Lemma F.2:

$$\widehat{\theta}_{\text{MEC}} - \theta_0 = \underbrace{(P_N - P)m_0}_{(A)} + \underbrace{(P_n - P)(Y - m_0)}_{(B)} + \underbrace{(P_n^\omega - P_N)\, m_{0,\perp}}_{(C)-(E)} + \underbrace{(P_n^\omega - P_n)(Y - m_0)}_{(D)}. \tag{56}$$

Multiplying $\sqrt{N}$ on both sides of (56) yields

$$\sqrt{N}\,\big(\widehat{\theta}_{\text{MEC}} - \theta_0\big) = \underbrace{\sqrt{N}\,(P_N - P)m_0}_{(I) = \sqrt{N}(A)} + \underbrace{\sqrt{N}\,(P_n - P)(Y - m_0)}_{(II) = \sqrt{N}(B)}$$
$$+ \underbrace{\sqrt{N}\,(P_n^\omega - P_N)\, m_{0,\perp}}_{(III) = \sqrt{N}\{(C)-(E)\}} + \underbrace{\sqrt{N}\,(P_n^\omega - P_n)(Y - m_0)}_{(IV) = \sqrt{N}(D)}. \tag{57}$$

**Terms** $(I)$ **and** $(II)$**.** By the CLT for the unlabeled and labeled samples with $n/N \to f \in (0,1)$,

$$\sqrt{N}\,(P_N - P)m_0 \xrightarrow{d} \mathcal{N}\big(0, \text{Var}\{m_0(X)\}\big), \qquad \sqrt{N}\,(P_n - P)(Y - m_0) \xrightarrow{d} \mathcal{N}\big(0, \tfrac{1}{f}\, \text{Var}\{\varepsilon\}\big),$$

where $\varepsilon := Y - m_0(X)$ with $\mathbb{E}[\varepsilon \mid X] = 0$. Moreover, the two terms are jointly asymptotically normal and $\text{Cov}\big(m_0(X), \varepsilon\big) = 0$, so their joint limit has zero covariance. Hence,

$$\sqrt{N}\,(P_N - P)m_0 + \sqrt{N}\,(P_n - P)(Y - m_0) \xrightarrow{d} \mathcal{N}\Big(0,\ \text{Var}\{m_0(X)\} + \tfrac{1}{f}\, \text{Var}\{Y - m_0(X)\}\Big).$$

**Term** $(III)$**.** Recall $(III) = \sqrt{N}\{(C) - (E)\} = \sqrt{N}\,(P_n^\omega - P_N)m_{0,\perp}$ with $m_{0,\perp} := m_0 - \Pi_W m_0$ and $W = \text{span}\{1, \widehat{m}^{(-)}\}$. We use the same splitting as in the consistency proof:

$$(C) - (E) = (P_n^\omega - P_N)m_{0,\perp} = \underbrace{(P_n^\omega - P_n)m_{0,\perp}}_{(i)} + \underbrace{(P_n - P)m_{0,\perp}}_{(ii)} - \underbrace{(P_N - P)m_{0,\perp}}_{(iii)}.$$

Let $\mathcal{T}$ be the $\sigma$-field generated by the training procedure that produces $\widehat{m}^{(-)}$. Then $W = \text{span}\{1, \widehat{m}^{(-)}\}$, $\Pi_W m_0$, and $m_{0,\perp} := m_0 - \Pi_W m_0$ are $\mathcal{T}$-measurable. Because $1 \in W$ and $m_{0,\perp} \perp W$ in $L_2(P_X)$, we have $\mathbb{E}[m_{0,\perp}(X)] = P m_{0,\perp} = 0$.

We assume the *projection error* is small:

$$\|m_{0,\perp}\|_{L_2(P_X)} = \|m_0 - \Pi_W m_0\|_{L_2(P_X)} = o_p(1),$$

which is weaker than requiring $\|\widehat{m}^{(-)} - m_0\|_{L_2(P_X)} = o_p(1)$. Indeed, since $\Pi_W m_0$ is the $L_2(P_X)$-projection of $m_0$ onto $W$,

$$\|m_0 - \Pi_W m_0\|_{L_2(P_X)} = \inf_{a,b\in\mathbb{R}} \|m_0 - (a + b\,\widehat{m}^{(-)})\|_{L_2(P_X)} \leq \|m_0 - \widehat{m}^{(-)}\|_{L_2(P_X)}.$$

Thus $L_2$-consistency of $\widehat{m}^{(-)}$ is sufficient (but not necessary) for the projection error to vanish.

*Control of* (ii) *and* (iii). Conditionally on $\mathcal{T}$, the function $m_{0,\perp}$ is fixed, mean–zero, and square–integrable. By the cross-fitted empirical-process bound (Lemma E.2; applicable under $n/N \to f \in (0,1)$),

$$(ii) = (P_n - P)m_{0,\perp} = O_p\Big(\tfrac{\|m_{0,\perp}\|_{L_2(P_X)}}{\sqrt{n}}\Big),$$
$$(iii) = (P_N - P)m_{0,\perp} = O_p\Big(\tfrac{\|m_{0,\perp}\|_{L_2(P_X)}}{\sqrt{N}}\Big),$$

conditionally on $\mathcal{T}$, and hence also unconditionally by iterated expectation. Multiplying by $\sqrt{N}$ gives

$$\sqrt{N}\,(ii) = O_p\Big(\sqrt{\tfrac{N}{n}}\, \|m_{0,\perp}\|_{L_2(P_X)}\Big) = o_p(1), \qquad \sqrt{N}\,(iii) = O_p\big(\|m_{0,\perp}\|_{L_2(P_X)}\big) = o_p(1),$$

since $\|m_{0,\perp}\|_{L_2(P_X)} = o_p(1)$ and $N/n = O(1)$.

*Control of* (i). Apply the weighted Cauchy–Schwarz inequality with weights $w_j = \tilde{g}'_j > 0$ (defined in the proof of Lemma F.2):

$$|(\mathrm{i})| = \left| \frac{1}{n} \sum_{j \in S} (\widehat{\omega}_j - d_j) \, m_{0,\perp}(X_j) \right| \le \frac{1}{n} \underbrace{\left( \sum_{j \in S} \tilde{g}'_j (\widehat{\omega}_j - d_j)^2 \right)^{1/2}}_{\star} \underbrace{\left( \sum_{j \in S} \frac{m_{0,\perp}(X_j)^2}{\tilde{g}'_j} \right)^{1/2}}_{*}. \tag{58}$$

By Lemma F.1 and A2 (weight stability),

$$\sum_{j \in S} \tilde{g}'_j (\widehat{\omega}_j - d_j)^2 = \{D_G(\widehat{\omega}\|d) + D_G(d\|\widehat{\omega})\} = O_p(1),$$

so the first square root ($\star$) in (58) is $O_p(1)$.

For the second square root ($*$), use

$$\frac{1}{n} \sum_{j \in S} \frac{m_{0,\perp}(X_j)^2}{\tilde{g}'_j} \le \left( \max_{j \in S} \tfrac{1}{\tilde{g}'_j} \right) \frac{1}{n} \sum_{j \in S} m_{0,\perp}(X_j)^2.$$

Because $g'$ is continuous and the calibrated solution lies in a feasible (hence tight/compact) region, $\max_{j \in S}\{1/\tilde{g}'_j\} = O_p(1)$. Conditionally on $\mathcal{T}$, a (conditional) LLN yields

$$\frac{1}{n} \sum_{j \in S} m_{0,\perp}(X_j)^2 = \|m_{0,\perp}\|^2_{L_2(P_X)} + o_p(1),$$

so

$$\frac{1}{n} \sum_{j \in S} \frac{m_{0,\perp}(X_j)^2}{\tilde{g}'_j} = O_p\big(\|m_{0,\perp}\|^2_{L_2(P_X)}\big).$$

Hence

$$\left( \sum_{j \in S} \frac{m_{0,\perp}(X_j)^2}{\tilde{g}'_j} \right)^{1/2} = O_p\big(\sqrt{n}\, \|m_{0,\perp}\|_{L_2(P_X)}\big).$$

Plugging into (58) gives

$$|(\mathrm{i})| \le \frac{1}{n} O_p(1) \cdot O_p\big(\sqrt{n}\, \|m_{0,\perp}\|_{L_2(P_X)}\big) = O_p\big(n^{-1/2}\, \|m_{0,\perp}\|_{L_2(P_X)}\big),$$

and therefore

$$\sqrt{N}\,(\mathrm{i}) = O_p\Big(\sqrt{\tfrac{N}{n}}\, \|m_{0,\perp}\|_{L_2(P_X)}\Big) = o_p(1),$$

since $N/n = O(1)$ and $\|m_{0,\perp}\|_{L_2(P_X)} = o_p(1)$.

*Putting the pieces together.* We have shown $\sqrt{N}\,(\mathrm{i}) = o_p(1)$, $\sqrt{N}\,(\mathrm{ii}) = o_p(1)$, and $\sqrt{N}\,(\mathrm{iii}) = o_p(1)$; hence

$$(III) = \sqrt{N}\{(C) - (E)\} = \sqrt{N}\,(P_n^\omega - P_N)m_{0,\perp} = \sqrt{N}\{(\mathrm{i}) + (\mathrm{ii}) - (\mathrm{iii})\} = o_p(1).$$

**Term** $(IV)$. Recall

$$(IV) = \sqrt{N}(D) = \sqrt{N}\,(P_n^\omega - P_n)(Y - m_0) = \frac{\sqrt{N}}{N} \sum_{j \in S} (\widehat{\omega}_j - d_j)\, \varepsilon_j$$

$$= \frac{1}{\sqrt{N}} \sum_{j \in S} (\widehat{\omega}_j - d_j)\, \varepsilon_j, \qquad \varepsilon_j := Y_j - m_0(X_j).$$

*KKT linearization of the weights.* From the calibration KKT conditions, $g(\widehat{\omega}_j) = g(d_j) + z_j^\top \widehat{\lambda}$ with $z_j := h(X_j) = (1, \widehat{m}^{(-)}(X_j))^\top$ and dual vector $\widehat{\lambda} \in \mathbb{R}^2$. Since $g^{-1}$ is $C^2$ and strictly increasing, a Taylor expansion of $g^{-1}$ at $g(d_j)$ yields, for some $\xi_j$ between $g(d_j)$ and $g(d_j) + z_j^\top \widehat{\lambda}$,

$$\widehat{\omega}_j - d_j = q_j\, z_j^\top \widehat{\lambda} \,+\, r_{jN}, \qquad q_j := \frac{1}{g'(d_j)}, \qquad r_{jN} = \frac{1}{2}(z_j^\top \widehat{\lambda})^2\, (g^{-1})''(\xi_j).$$

With $(g^{-1})''$ locally bounded and $\mathbb{E}\|z\|^2 < \infty$ (A1), we have the uniform bound

$$|r_{jN}| \;\leq\; C\,(z_j^\top \widehat{\lambda})^2 \;=\; O_p(\|\widehat{\lambda}\|^2) \qquad \text{(uniformly in } j\text{).}$$

Under A4 ($\|\widehat{\lambda}\| = O_p(n^{-1/2})$), it follows that $r_{jN} = o_p(\|\widehat{\lambda}\|)$ since $|r_{jN}|/\|\widehat{\lambda}\| \leq C\,(z_j^\top \widehat{\lambda})^2/\|\widehat{\lambda}\| \leq C\|z_j\|^2\|\widehat{\lambda}\|^2/\|\widehat{\lambda}\| = C\|z_j\|^2\|\widehat{\lambda}\| \xrightarrow{p} 0$.

*Decomposition.* Thus, term $(IV)$ can be decomposed as

$$(IV) = \frac{1}{\sqrt{N}} \sum_{j \in S} (\widehat{\omega}_j - d_j)\, \varepsilon_j = \frac{1}{\sqrt{N}} \sum_{j \in S} (q_j\, z_j^\top \widehat{\lambda} \,+\, r_{jN})\, \varepsilon_j$$

$$= \underbrace{\frac{1}{\sqrt{N}} \sum_{j \in S} q_j (z_j^\top \widehat{\lambda})\, \varepsilon_j}_{T_{1N}} \,+\, \underbrace{\frac{1}{\sqrt{N}} \sum_{j \in S} r_{jN}\, \varepsilon_j}_{T_{2N}}.$$

*Control of $T_{1N}$ (linear term).* Rewrite

$$T_{1N} = \sqrt{\frac{n}{N}}\, \widehat{\lambda}^\top \left\{ \frac{1}{\sqrt{n}} \sum_{j \in S} q_j z_j\, \varepsilon_j \right\}.$$

By A1, we have $\mathbb{E}\|z\|^2 = \mathbb{E}\|h(X)\|^2 < \infty$; from the standing conditions we also have $\mathbb{E}[\varepsilon \mid X] = 0$ and $\mathbb{E}[\varepsilon^2] < \infty$. Under these moment conditions and the weighted Gram limit (A3),

$$A_n := \frac{1}{n} \sum_{j \in S} q_j z_j z_j^\top \xrightarrow{p} \Sigma_q \succ 0,$$

a multivariate CLT yields

$$\frac{1}{\sqrt{n}} \sum_{j \in S} q_j z_j\, \varepsilon_j \xrightarrow{d} \mathcal{N}\big(0,\, \mathrm{Var}(\varepsilon)\, \Sigma_q\big) \quad \Rightarrow \quad \left\| \frac{1}{\sqrt{n}} \sum_{j \in S} q_j z_j\, \varepsilon_j \right\| = O_p(1).$$

With the small–dual assumption $\|\widehat{\lambda}\| = O_p(n^{-1/2})$ (A4) and $n/N \to f \in (0,1)$,

$$T_{1N} = \sqrt{\tfrac{n}{N}}\, \|\widehat{\lambda}\|\, O_p(1) = O_p(n^{-1/2}) = o_p(1).$$

*Control of $T_{2N}$ (remainder).* From the Taylor remainder and local boundedness of $(g^{-1})''$, $|r_{jN}| \leq C\,(z_j^\top \widehat{\lambda})^2$ for some constant $C < \infty$. By Cauchy–Schwarz,

$$|T_{2N}| = \left| \frac{1}{\sqrt{N}} \sum_{j \in S} r_{jN}\, \varepsilon_j \right| \leq \frac{C}{\sqrt{N}} \left( \sum_{j \in S} (z_j^\top \widehat{\lambda})^4 \right)^{1/2} \left( \sum_{j \in S} \varepsilon_j^2 \right)^{1/2}.$$

We now bound the two factors. Write $u := \widehat{\lambda}/\|\widehat{\lambda}\|$ (when $\widehat{\lambda} \neq 0$); then

$$(z_j^\top \widehat{\lambda})^4 = \big(z_j^\top (\|\widehat{\lambda}\| u)\big)^4 = \big(\|\widehat{\lambda}\|\, z_j^\top u\big)^4 = \|\widehat{\lambda}\|^4 (z_j^\top u)^4.$$

We also have

$$\mathbb{E}[(z^\top u)^4] \leq \mathbb{E}\left[\|z\|^4 \|u\|^4\right] = \mathbb{E}\|z\|^4 < \infty,$$

since by Cauchy–Schwarz, $|z^\top u| \leq \|z\|\,\|u\|$ and $\|u\| = 1$ by $\mathbb{E}\|z\|^4 < \infty$ (A1 strengthened to fourth moments). Thus, by the LLN,

$$\frac{1}{n}\sum_{j \in S}(z_j^\top \widehat{\lambda})^4 = \|\widehat{\lambda}\|^4 \cdot \frac{1}{n}\sum_{j \in S}(z_j^\top u)^4 = \|\widehat{\lambda}\|^4 \left\{\mathbb{E}[(z^\top u)^4] + o_p(1)\right\} = O_p(\|\widehat{\lambda}\|^4).$$

Therefore $\sum_{j \in S}(z_j^\top \widehat{\lambda})^4 = n\,O_p(\|\widehat{\lambda}\|^4)$. Similarly, $\sum_{j \in S}\varepsilon_j^2 = n\,\mathbb{E}[\varepsilon^2] + o_p(n) = O_p(n)$ since $\mathbb{E}[\varepsilon^2] < \infty$.

Putting the bounds together, we have

$$|T_{2N}| \leq \frac{C}{\sqrt{N}}\left(n\,O_p(\|\widehat{\lambda}\|^4)\right)^{1/2}\left(n\,O_p(1)\right)^{1/2} = \frac{C}{\sqrt{N}}\left(\sqrt{n}\,O_p(\|\widehat{\lambda}\|^2)\right)\left(\sqrt{n}\,O_p(1)\right).$$

Thus,

$$|T_{2N}| = \frac{C}{\sqrt{N}}\,n\,O_p(\|\widehat{\lambda}\|^2) = \sqrt{n}\,\sqrt{\frac{n}{N}}\,O_p(\|\widehat{\lambda}\|^2).$$

Since $n/N \to f \in (0,1)$, we have $\sqrt{n/N} = \sqrt{f} + o(1)$, which can be absorbed into the $O_p(\cdot)$ term; hence

$$|T_{2N}| = \sqrt{n}\,O_p(\|\widehat{\lambda}\|^2).$$

Under A4, $\|\widehat{\lambda}\| = O_p(n^{-1/2})$, so $\|\widehat{\lambda}\|^2 = O_p(n^{-1})$, and

$$|T_{2N}| = \sqrt{n}\,O_p(n^{-1}) = O_p(n^{-1/2}) = o_p(1).$$

*Conclusion for* $(IV)$. Both pieces vanish:

$$(IV) = \sqrt{N}(D) = T_{1N} + T_{2N} = o_p(1).$$

**Conclusion.** Collecting the bounds established above, we have from (57) that

$$\sqrt{N}\left(\widehat{\theta}_{\mathrm{MEC}} - \theta_0\right) = \underbrace{\sqrt{N}(P_N - P)m_0 + \sqrt{N}(P_n - P)(Y - m_0)}_{\xrightarrow{d}\ \mathcal{N}\left(0,\ \mathrm{Var}\{m_0(X)\} + \frac{1}{f}\,\mathrm{Var}\left(Y - m_0(X)\right)\right)} + \underbrace{\sqrt{N}\{(C) - (E)\} + \sqrt{N}(D)}_{o_p(1)}.$$

Thus, we proved

$$\widehat{\theta}_{\mathrm{MEC}} \xrightarrow{d} \mathcal{N}\left(\theta_0,\ \frac{1}{N}\left\{\mathrm{Var}\{m_0(X)\} + \frac{1}{f}\,\mathrm{Var}\left(Y - m_0(X)\right)\right\}\right).$$

$\square$

**Theorem F.5.** *Assume the conditions of Lemma E.1 and Lemma F.2, and consider the MEC estimator*

$$\widehat{\theta}_{\mathrm{MEC}} = \frac{1}{N}\sum_{j \in S}\widehat{\omega}_j\,Y_j.$$

*Define the calibration subspace*

$$W := \mathrm{span}\{h\} = \mathrm{span}\{1, \widehat{m}^{(-)}(\cdot)\} = \{\,a + b\,\widehat{m}^{(-)}(\cdot) : a, b \in \mathbb{R}\,\} \subset L_2(P_X),$$

*and let $\Pi_W$ denote the $L_2(P_X)$ projection onto $W$, $\Pi_W m_0 = \arg\min_{v \in W} \mathbb{E}\left[(m_0(X) - v(X))^2\right]$. Define the projection error $m_{0,\perp} := m_0 - \Pi_W m_0$.*

*Assumptions*

*A1 Moments:* $\mathbb{E}\|h(X)\|^2 < \infty$ *(equivalently,* $\mathbb{E}[\widehat{m}^{(-)}(X)^2] < \infty$*).*

*A2 Weights stability: Calibrated weights satisfy the symmetric Bregman bound*

$$D_G(\widehat{\omega}\|d) + D_G(d\|\widehat{\omega}) = O_p(1).$$

*A3 Weighted Gram limit: With*

$$A_n := \frac{1}{n}\sum_{j\in S} q_j\, z_j z_j^\top, \qquad q_j := \frac{1}{g'(d_j)},$$

*one has* $A_n \xrightarrow{P} \Sigma_q \succ 0.$

*A4 Small dual. Let* $\widehat{\lambda}$ *be the dual optimizer of the calibration program. Then* $\|\widehat{\lambda}\| = O_p(n^{-1/2})$.

*Then:*

**(i) Consistency.** *If A1–A2 hold and the projection error is stochastically bounded in* $L_2(P_X)$,

$$\|m_{0,\perp}\|_{L_2(P_X)} = \|m_0 - \Pi_W m_0\|_{L_2(P_X)} = O_p(1),$$

*then* $\widehat{\theta}_{\mathrm{MEC}} \xrightarrow{P} \theta_0$.

**(ii) Asymptotic normality.** *If A1–A4 hold and the projection error is* $L_2(P_X)$*-consistent,*

$$\|m_{0,\perp}\|_{L_2(P_X)} = \|m_0 - \Pi_W m_0\|_{L_2(P_X)} = o_P(1),$$

*then*

$$\sqrt{N}\left(\widehat{\theta}_{\mathrm{MEC}} - \theta_0\right) \xrightarrow{d} \mathcal{N}\left(0,\, \sigma_f^2\right), \qquad \sigma_f^2 = \mathrm{Var}\left(m_0(X)\right) + \frac{1}{f}\,\mathrm{Var}\left(Y - m_0(X)\right).$$

*Proof.* The theorem follows from Lemma F.2 (consistency) together with Lemma F.4 (asymptotic normality). □

## G. Settings of machine-learning predictors and supplemental plots and table for the simulation experiment in the main document

### G.1. Settings of ML predictors used in simulations

For MEC and CF–PPI (6), all learners are trained only on the labeled folds and evaluated out of fold via $K=5$-fold cross-prediction; the unlabeled plug-in term uses the average of the $K$ fold-specific predictors. For vanilla PPI (4), each learner is fit once on all labeled data with no sample splitting, and the same fitted model is used to predict both labeled and unlabeled covariates (thereby reusing labels in the bias-correction term). Hyperparameters are held fixed across methods and label fractions $f$ to ensure a fair comparison; any undercoverage of confidence intervals should therefore be attributed to the estimation procedure rather than to differences in the ML predictor.

**Kernel ridge regression (KRR).** We use a Gaussian kernel with an unpenalized intercept. The kernel is $k_\ell(x, x') = \exp\big(-\|x - x'\|_2^2/(2\ell^2)\big)$ with $\ell = \sqrt{2d}$ (where $d$ is the covariate dimension), and the ridge penalty scales as $\lambda = c_\lambda n^{-\alpha}$ with $c_\lambda = 0.01$ and $\alpha = 0.5$. For simplicity, assume the labeled index set is $S = \{1, \ldots, n\}$. Given labeled data $\{(X_i, Y_i)\}_{i=1}^n$, let $K \in \mathbb{R}^{n \times n}$ be the Gram matrix $(K)_{ij} = k_\ell(X_i, X_j)$, set $A = K + n\lambda I_n$, and write $\mathbf{1} \in \mathbb{R}^n$ for the all-ones vector. Following our implementation, the unpenalized intercept is imposed via the projection $w = \frac{A^{-1}\mathbf{1}}{\mathbf{1}^\top A^{-1}\mathbf{1}}$ and $H = KA^{-1}\big(I_n - \mathbf{1}w^\top\big) + \mathbf{1}w^\top$. The in-sample fitted values are $\widehat{m}(X_{1:n}) = HY$ with degrees of freedom $\mathrm{df} = \mathrm{tr}(H)$. For a new covariate $x$, let $k(x) = (k_\ell(x, X_1), \ldots, k_\ell(x, X_n))^\top$; the predictor is

$$\widehat{m}(x) = k(x)^\top A^{-1}\big(I_n - \mathbf{1}w^\top\big)Y + w^\top Y.$$

We use $K=5$-fold cross-prediction: each learner is trained on the $K-1$ labeled folds and evaluated out of fold to produce $\widehat{m}^{(-)}(X_i)$ for the held-out $i$; for unlabeled covariates $x \in U$, the plug-in term averages the $K$ fold-specific predictions, $\bar{\mu}_U(x) = K^{-1}\sum_{k=1}^K \widehat{m}^{(-k)}(x)$. For MEC, the predictor basis used in calibration is $h(x) = (1, \widehat{m}^{(-)}(x)) \in \mathbb{R}^p$ ($p = 2$), keeping the calibration step low-dimensional and numerically stable across $d$ and $n$. Hyperparameters $(\ell, \lambda)$ are fixed across label fractions $f$ for comparability.

**Random forest (RF).** We fit regression forests (using the `ranger` package in R) for a continuous outcome with $T = 500$ trees; at each split, a feature subset of size $\lfloor\sqrt{d}\rfloor$ is considered (where $d$ is the covariate dimension); minimum leaf size 5; unlimited depth; and sample fraction 1.0 (bootstrap sampling). Trees are grown to (approximate) purity subject to the leaf-size constraint; no post–pruning is applied. Out-of-bag variance estimation is disabled (variance is handled by the inference layer).

Given labeled data $\{(X_i, Y_i)\}_{i=1}^n$, the forest predictor averages the tree predictors, $\widehat{m}(x) = (1/T)\sum_{t=1}^T \widehat{m}_t(x)$, where each $\widehat{m}_t$ is a regression tree trained on a bootstrap sample of the labeled data while restricting candidate split variables at each node to $\lfloor\sqrt{d}\rfloor$. We use $K=5$-fold cross–prediction exactly as in the KRR implementation.

**Feedforward neural network (FNN).** We fit a one–hidden–layer feedforward network for a continuous outcome using the `nnet` package in R. The network uses $H = 3$ hidden units with sigmoid activation and a linear output layer. We apply $\ell_2$ weight decay with penalty `decay = 10` and cap optimization at `maxit = 100` iterations. Inputs are standardized using the training means and standard deviations, and the same transformation is applied at prediction. Hyperparameters are held fixed across all label fractions $f$ for comparability. We use $K=5$-fold cross–prediction exactly as in the KRR implementation.

**k-nearest neighbors (kNN).** We use $k$-NN regression via the `kknn` package in R. We fix $k = 15$ neighbors, the Minkowski distance with exponent $p = 2$ (Euclidean), and a `rectangular` kernel (uniform weights over the $k$ nearest neighbors). Features are standardized using the training means and standard deviations prior to distance computation, and the same transformation is applied at prediction time. Given labeled data $\{(X_i, Y_i)\}_{i=1}^n$, the predictor is the locally weighted average $\widehat{m}(x) = (\sum_{i=1}^n w_i(x)\, Y_i)/\sum_{i=1}^n w_i(x)$, with $w_i(x) = K(\|x - X_i\|_2/r_k(x))$, where $r_k(x)$ is the distance from $x$ to its $k$-th nearest neighbor and $K(u) = \mathbf{1}\{u \le 1\}$ for the rectangular kernel. Hyperparameters are held fixed across all label fractions $f$ for comparability. We use $K=5$-fold cross–prediction exactly as in the KRR implementation.

### G.2. Supplemental plots for the simulation experiment in the main document

Figure 5 presents the full results from the main simulation experiment ($N = 1000$, $f \in \{0.10, 0.15, \ldots, 0.50\}$, $n = fN \in \{100, 150, \ldots, 500\}$, $d = 10$, $\sigma_y = 5$), including MEC with four generators—quadratic, Kullback–Leibler (KL), empirical

likelihood (EL), and squared Hellinger. MEC variants are shown in dashed or dotted colored lines for clarity. Across all generators, MEC exhibits nearly identical performance: coverage remains close to the nominal $95\%$ level, interval widths lie between the classical and oracle references, and bias is substantially reduced compared to vanilla PPI across label fractions $f$. These results confirm that the proposed MEC framework is robust to the choice of generator and consistently improves finite-sample efficiency while maintaining valid inference.

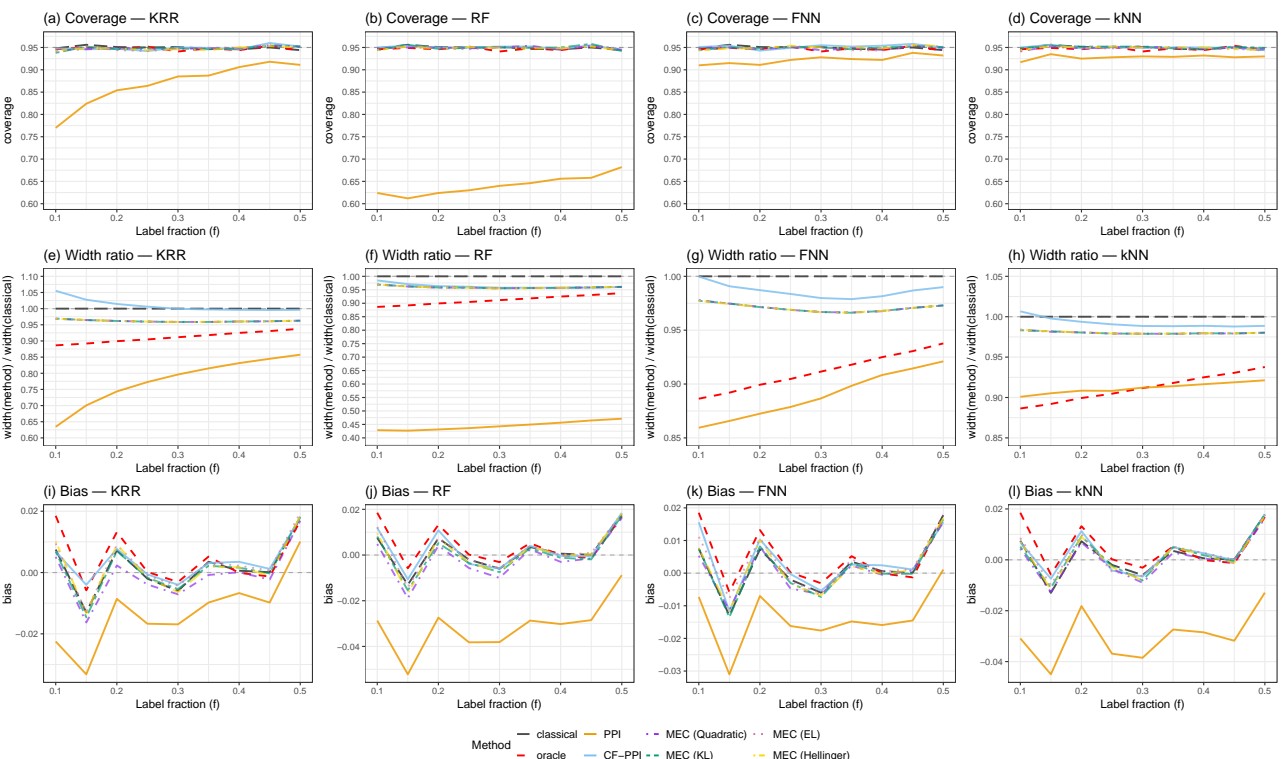

*Figure 5.* Supplemental plots corresponding to Figure 1 in the main document, showing MEC with four generators—quadratic, Kullback–Leibler (KL), empirical likelihood (EL), and squared Hellinger. MEC performance is robust to the generator choice. Unlabeled sample size $N = 1000$; labeled size $n = fN$ with $f \in [0.1, 0.5]$; covariate dimension $d = 10$ with i.i.d. $X \sim \mathcal{N}(0,1)$; outcome noise standard deviation $\sigma_y = 5$.

### G.3. Supplemental table for the simulation experiment in the main document

Table 2 reports additional simulation results for the setting considered in Section 6 of the main document, with $N = 1000$, $n = 100$, and label fraction $f = n/N = 0.1$. The predictor $\widehat{m}(\cdot)$ is fitted using KRR. In addition to the methods reported in the main text, we include PPI++ (Angelopoulos et al., 2024) with cross-fitting and calibration estimators based on the 11-dimensional moment-feature basis

$$h(X_j) = (1, X_{j1}, X_{j2}, \dots, X_{j10}) \in \mathbb{R}^{11}.$$

We also report the effective sample size (ESS) and the coefficient of variation (CV) of the calibrated weights as empirical diagnostics for weight stability, where $\mathrm{ESS} = (\sum_{j \in S} \widehat{\omega}_j)^2 / \sum_{j \in S} \widehat{\omega}_j^2 \in [1, n]$ and $\mathrm{CV} = \mathrm{sd}\{\widehat{\omega}_j : j \in S\}/\bar{\omega}$, with $\bar{\omega} = n^{-1} \sum_{j \in S} \widehat{\omega}_j$. Values of ESS close to $n = |S| = 100$ and CV close to zero indicate stable, nearly uniform calibrated weights, whereas smaller ESS and larger CV indicate more dispersed or concentrated weights.

The results have two main implications. First, PPI++ has operating characteristics very close to those of MEC with the quadratic generator. For example, the RMSE, coverage, and mean confidence-interval length of PPI++ are nearly identical to those of MEC (Quadratic). This empirical finding is consistent with Theorem 4.1, which shows that MEC admits a dual GREG representation and is asymptotically equivalent to PPI++ when the Bregman generator is quadratic. More specifically,

*Table 2.* Additional simulation results including PPI++ as a baseline and weight calibration using an 11-dimensional moment-feature basis. ESS and CV are reported only for the calibration methods.

| Method | Bias | RMSE | Coverage | Mean CI Len | ESS | CV |
|---|---|---|---|---|---|---|
| Classical | 0.008 | 0.561 | 0.948 | 2.237 | – | – |
| PPI (oracle) | 0.019 | 0.501 | 0.946 | 1.983 | – | – |
| PPI (no cross-fitting) | -0.023 | 0.583 | 0.770 | 1.420 | – | – |
| CF-PPI | 0.007 | 0.597 | 0.948 | 2.361 | – | – |
| Calibration via Moment Basis (Quadratic) | 0.013 | 0.608 | 0.958 | 2.507 | 89.004 | 0.348 |
| Calibration via Moment Basis (KL) | 0.013 | 0.618 | 0.953 | 2.524 | 88.396 | 0.358 |
| Calibration via Moment Basis (EL) | 0.013 | 0.626 | 0.958 | 2.552 | 86.410 | 0.392 |
| Calibration via Moment Basis (Hellinger) | 0.011 | 0.620 | 0.958 | 2.532 | 87.627 | 0.371 |
| MEC (Quadratic) | 0.005 | 0.554 | 0.944 | 2.168 | 98.934 | 0.084 |
| MEC (KL) | 0.007 | 0.556 | 0.938 | 2.168 | 98.876 | 0.086 |
| MEC (EL) | 0.010 | 0.557 | 0.940 | 2.166 | 98.839 | 0.087 |
| MEC (Hellinger) | 0.010 | 0.557 | 0.943 | 2.168 | 98.836 | 0.087 |
| PPI++ (with cross-fitting) | 0.004 | 0.556 | 0.944 | 2.171 | – | – |

*Note:* Mean CI Len = mean length of the 95% confidence interval; RMSE = root mean squared error; ESS = effective sample size; CV = coefficient of variation.

in the quadratic case, the MEC estimator reduces to a regression-adjusted form whose slope coincides with the optimal PPI++ tuning coefficient up to the finite-sample scaling factor $1/(1 + n/N) = N/(N + n)$, which is close to one in the semi-supervised regime $N \gg n$. Thus, the simulation results support the interpretation that MEC generalizes PPI++ in a dual sense.

Second, the results illustrate why a crude moment-feature basis is not recommended for the semi-supervised mean estimation problem considered in this paper. Although moment-feature calibration attains near-nominal coverage, it produces larger RMSEs and longer confidence intervals than both CF–PPI and MEC. This deterioration is accompanied by less stable calibrated weights: the ESSs for the moment-feature basis are around 86–89 and the CVs are around 0.35–0.39, whereas the MEC weights based on the two-dimensional predictor basis have ESSs close to $n = 100$ and CVs close to zero. These diagnostics show that calibrating on non-data-adaptive moment features can induce unnecessary weight variability without improving efficiency. In contrast, MEC uses the out-of-fold ML predictor as a low-dimensional, outcome-relevant calibration basis, leading to substantially more stable weights and better finite-sample performance.

# H. Additional simulation experiments

## H.1. Simulation setup

In this section, we conduct additional simulation experiments using the same synthetic-data procedure described in Section 6 of the main document. Recall that we generate two i.i.d. samples: an unlabeled covariate set of size $N$, $\{X_i\}_{i=1}^{N}$, and a labeled sample $\{(X_j, Y_j)\}_{j \in S}$ with $|S| = n$, where

$$Y_j = m_0(X_j) + \varepsilon_j, \qquad \varepsilon_j \sim \mathcal{N}(0, \sigma_y^2).$$

The true regression function is $m_0(x) = \sum_{k=1}^{d} g_k(x_k)$ with $g_1(x) = e^{-x}$, $g_2(x) = x^2$, $g_3(x) = x$, $g_4(x) = \mathbb{I}\{x > 0\}$, $g_5(x) = \cos x$, and $g_k(x) \equiv 0$ for $k = 6, \ldots, d$.

We control (i) the unlabeled sample size $N$; (ii) the labeled set $S$ of size $n = fN$; (iii) the labeling fraction $f$; (iv) the covariate dimension $d$; (v) the correlation $\rho$, where the covariates $X = (x_1, \ldots, x_d)^\top \in \mathbb{R}^d$ are generated as mean-zero Gaussian with AR(1) covariance $\Sigma_{ij} = \rho^{|i-j|}$ (so $\rho = 0$ yields $\Sigma = I_d$, i.e., independent coordinates); (vi) the outcome-noise standard deviation $\sigma_y$; and (vii) the number of folds $K$ used for sample splitting.

Recall that, in Section 6, we set $N = 1000$, $d = 10$, $\rho = 0$, $\sigma_y = 5$, and varied $n = fN$ with $f \in [0.1, 0.5]$, and we showed that MEC outperforms CF–PPI and vanilla PPI; see Figures 1 and 5.

We consider the following setup for additional simulations:

- **Additional simulation experiment 1.** Vary the covariate dimension $d$ from 5 to 15, fixing $N = 1000$, $n = 200$ ($f = 0.2$), $\rho = 0$, and $\sigma_y = 5$.

- **Additional simulation experiment 2.** Vary the outcome-noise standard deviation $\sigma_y$ from 1 to 10, fixing $N = 1000$, $n = 200$ ($f = 0.2$), $\rho = 0$, and $d = 10$.

- **Additional simulation experiment 3.** Vary the covariate correlation $\rho$ from 0 to 0.8, fixing $N = 1000$, $n = 200$ ($f = 0.2$), $\sigma_y = 5$, and $d = 10$.

- **Additional simulation experiment 4.** Vary the number of folds $K$ from 5 to 10, fixing $N = 1000$, $n = 200$ ($f = 0.2$), $\rho = 0$, $\sigma_y = 5$, and $d = 10$.

The purpose of Additional Simulation Experiments 1–3 is to investigate how MEC, vanilla PPI, and CF–PPI behave as the data-generating setup becomes more challenging (e.g., larger $d$, higher noise $\sigma_y$, or stronger correlation $\rho$) and to assess each method's robustness to these factors. Additional Simulation Experiment 4 examines sensitivity to the number of folds $K$ used for cross-fitting in MEC and CF–PPI; ideally, performance should be insensitive to $K$ (i.e., stable across reasonable choices of $K$).

Predictors are tuned identically across methods, as detailed in Subsection G.1. We report nominal 95% coverage and the width ratio $\mathrm{WR}_{\mathrm{method}}(f)$ relative to the classical label-only estimator. For MEC, we consider only the quadratic generator, since MEC's results are robust to the choice of generator.

### H.2. Additional simulation experiment 1: varying covariate dimension $d$

We examine how the covariate dimension $d$ affects the performance of MEC, CF–PPI, and vanilla PPI. The results are summarized in Figure 6.

Across all learners and values of $d$, MEC maintains near-nominal coverage and consistently attains tighter confidence intervals than CF–PPI. As $d$ increases, CF–PPI remains valid (maintaining near-nominal coverage, like MEC) but exhibits performance degradation for KRR, FNN, and kNN. In particular, for KRR and FNN, CF–PPI performs worse than the classical estimator when $d = 15$; this indicates that cross-prediction can ensure valid inference but does not automatically deliver efficiency gains. Vanilla PPI under-covers throughout, and the under-coverage becomes more pronounced as $d$ grows because label reuse interacts unfavorably with higher-dimensional predictors.

The efficiency advantage of MEC is stable as $d$ increases. The gap $\mathrm{WR}_{\mathrm{MEC}}(f) - \mathrm{WR}_{\mathrm{oracle}}(f)$ remains relatively small even at $d = 15$, indicating that MEC continues to borrow effectively from predicted outcomes of unlabeled covariates and mitigates the efficiency shortfall that CF–PPI exhibits.

Numerically, MEC's robustness arises because its calibration operates in a fixed, two-dimensional basis $h = (1, m^{(-)})$, independent of $d$. The dual Newton solver (Section C.1) therefore remains well-conditioned regardless of $N$, $n$, and $d$; the calibrated weights are stable; and the optimization objective plateaus at a bounded level. By projecting onto $\mathrm{span}\{1, m^{(-)}\}$, MEC removes the component of the signal captured by the predictor and confines any variance inflation to the orthogonal remainder; thus, even when raw prediction error grows with dimension, efficiency is preserved so long as the predictor continues to capture a meaningful low-dimensional signal.

Overall, increasing $d$ makes the problem harder for all methods, but MEC remains both valid and the most efficient among the competitors, whereas CF–PPI is moderately sensitive to dimension and vanilla PPI is invalid due to systematic under-coverage.

### H.3. Additional simulation experiment 2: varying standard deviation $\sigma_y$

We examine how the outcome-noise standard deviation $\sigma_y$ affects the performance of MEC, CF–PPI, and vanilla PPI. The results are summarized in Figure 7.

Across all learners and noise levels, MEC maintains near-nominal coverage and consistently yields tighter valid confidence intervals than CF–PPI. As $\sigma_y$ increases, intervals inflate for all methods, but MEC's width ratios remain well below those of CF–PPI. CF–PPI generally preserves validity but shows substantial efficiency deterioration at larger $\sigma_y$, especially for KRR, FNN, and kNN, where efficiency can even fall below that of the classical estimator. Vanilla PPI under-covers across the board, with under-coverage worsening as $\sigma_y$ grows due to label reuse interacting with noisier residuals.

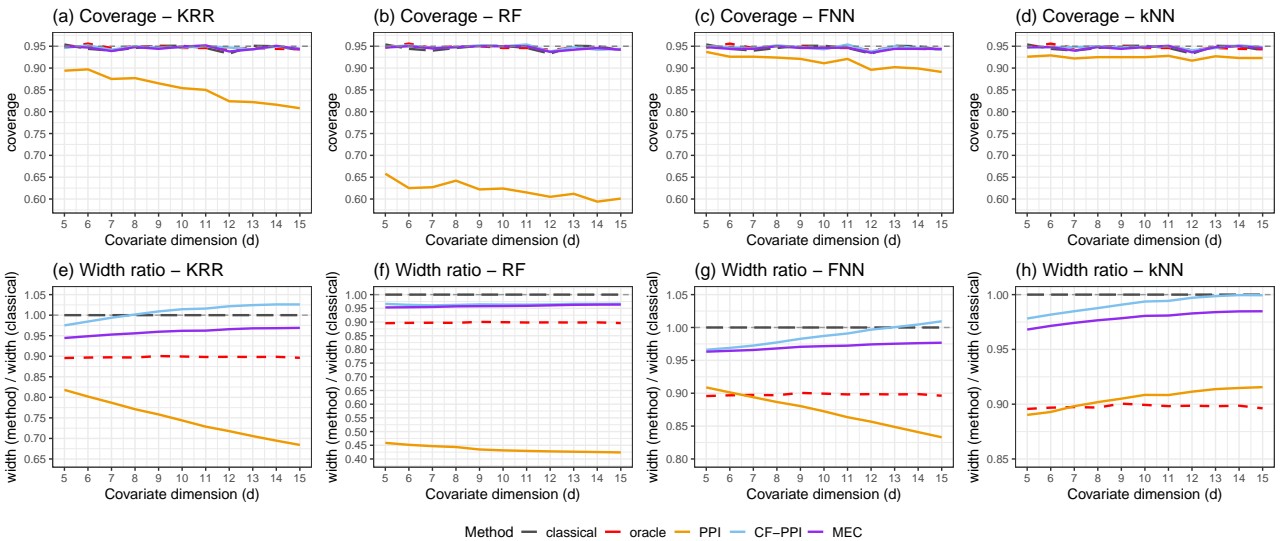

*Figure 6.* Additional Simulation Experiment 1: effect of covariate dimension $d$. We vary $d$ from 5 to 15 while fixing $N = 1000$, $n = 200$ ($f = 0.2$), $\rho = 0$, and $\sigma_y = 5$. For MEC, we display only the quadratic generator for visual clarity, since MEC's results are robust to the choice of generator.

The efficiency advantage of MEC is stable as $\sigma_y$ increases: the gap $\mathrm{WR}_{\mathrm{MEC}}(f) - \mathrm{WR}_{\mathrm{oracle}}(f)$ remains relatively small even at $\sigma_y = 10$, indicating that MEC continues to borrow effectively from unlabeled covariates while controlling misspecification-driven variance inflation.

Overall, increasing $\sigma_y$ makes the problem harder for all procedures, but MEC remains both valid and the most efficient among the competitors. CF–PPI is a reasonable baseline for validity yet can forfeit efficiency at high noise, whereas vanilla PPI remains invalid due to systematic under-coverage.

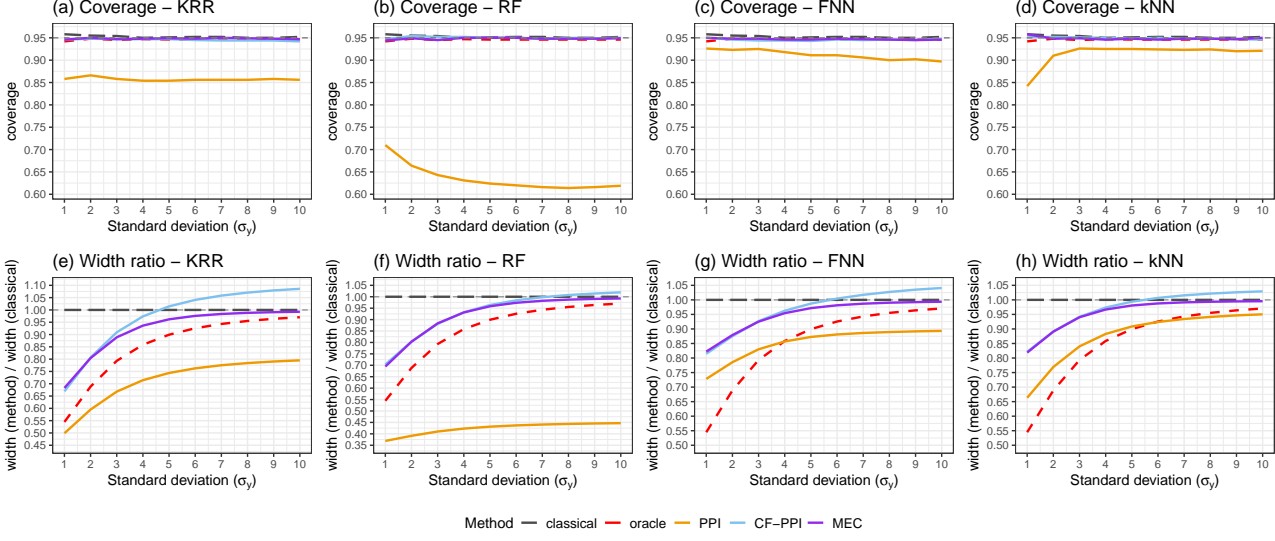

*Figure 7.* Additional Simulation Experiment 2: effect of standard deviation $\sigma_y$. We vary $\sigma_y$ from 1 to 10 while fixing $N = 1000$, $n = 200$ ($f = 0.2$), $\rho = 0$, and $d = 10$.

## H.4. Additional simulation experiment 3: varying covariate correlation $\rho$

We study the effect of covariate correlation by varying the AR(1) parameter $\rho$; results appear in Figure 8. Across learners and correlation levels, MEC maintains near-nominal coverage and consistently delivers tighter valid intervals than CF–PPI. As $\rho$ increases, intervals widen for all methods, yet MEC's width ratios remain well below those of CF–PPI. CF–PPI generally preserves validity, whereas vanilla PPI under-covers throughout. Overall, MEC is the most robust and efficient method across correlation levels, CF–PPI is valid but less efficient under strong correlation, and vanilla PPI is invalid.

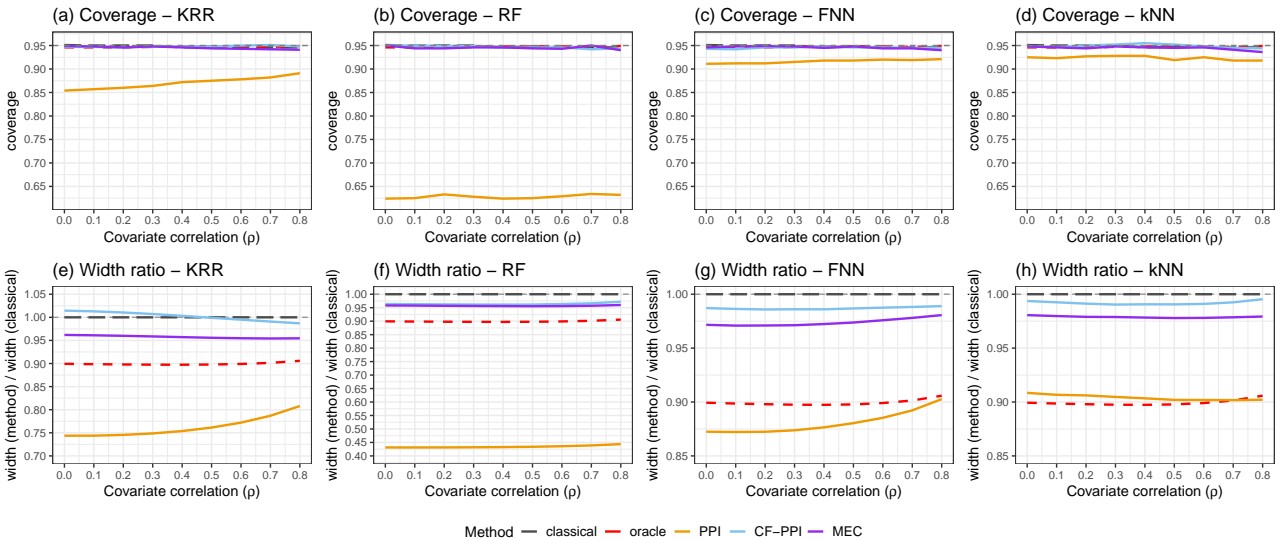

*Figure 8.* Additional Simulation Experiment 3: effect of covariate correlation $\rho$. We vary $\rho$ from 0 to 0.8 while fixing $N = 1000$, $n = 200$ ($f = 0.2$), $\sigma_y = 5$, and $d = 10$.

## H.5. Additional simulation experiment 4: varying fold number $K$

We assess sensitivity to the number of folds $K$ used for cross-prediction in MEC and CF–PPI. Results are summarized in Figure 9. Across learners and both choices of $K$, MEC maintains near-nominal coverage and achieves tighter valid intervals than CF–PPI, with only negligible differences between $K = 5$ and $K = 10$. CF–PPI generally preserves validity across $K$ values. Vanilla PPI under-covers regardless of $K$ since it does not employ sample splitting. Overall, performances of MEC and CF–PPI are largely insensitive to $K$. Given similar statistical behavior and lower computational cost, $K = 5$ is a practical default.

## I. Real data application: Energy Efficiency dataset

We apply the proposed MEC method to the *Energy Efficiency* dataset, available from the UCI Machine Learning Repository. The dataset comprises 768 building designs with eight continuous covariates: $x_1$ = relative compactness, $x_2$ = surface area, $x_3$ = wall area, $x_4$ = roof area, $x_5$ = overall height, $x_6$ = orientation, $x_7$ = glazing area, and $x_8$ = glazing area distribution. The data include two response variables, *Heating Load* and *Cooling Load*; in this application we set $y$ to be *Heating Load*. There are no missing values.

We follow a semi-supervised setup, where a subset of observations is randomly designated as labeled and the remainder as unlabeled (covariates only). Specifically, we take $n = 115$ labeled pairs $\{(X_j, Y_j) : j \in S\}$ and $N = 653$ unlabeled covariates $\{X_i : i = 1, \ldots, N\}$, so the labeling fraction is $f = n/N = 115/653 \approx 0.176$. Our target parameter is the population mean of the response, $\theta = \mathbb{E}[Y]$ (i.e., the population mean of Heating Load), under the working model $Y = m(X) + \varepsilon$ with $\mathbb{E}[\varepsilon \mid X] = 0$, where $m$ is unknown. Because this is a real dataset, the true value of $\theta$ is unknown; as a reference, we compute the full-sample mean using all 768 observations, obtaining $\bar{Y}_{\text{full}} = 22.307$, which we use as the ground truth for this application.

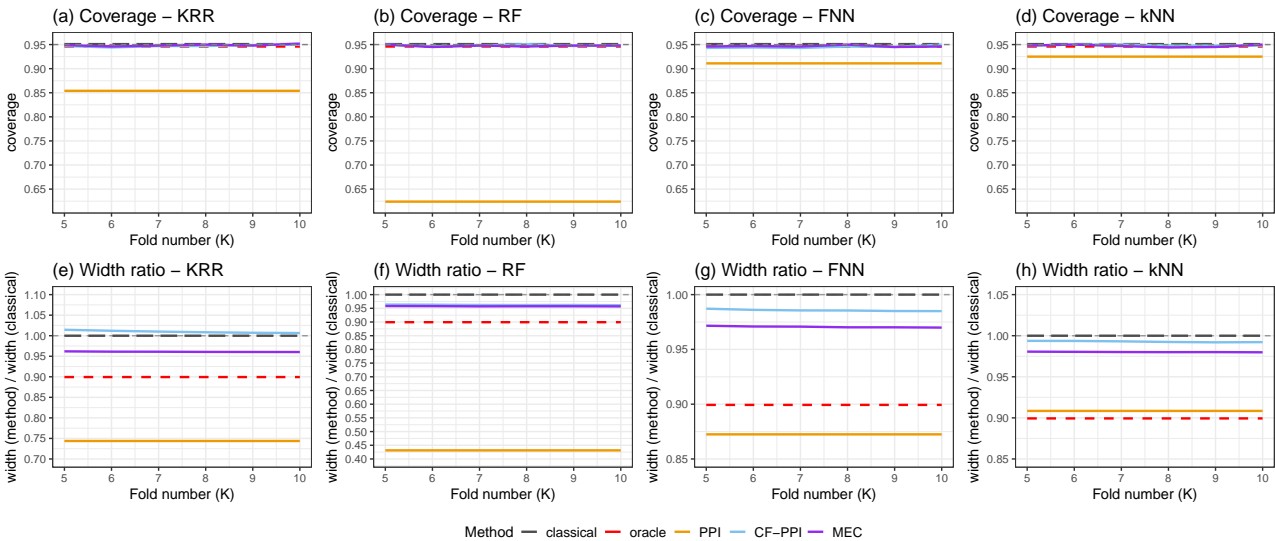

*Figure 9.* Additional Simulation Experiment 4: effect of fold number $K$ for sample-splitting. We vary $K$ from 5 to 10 while fixing $N = 1000$, $n = 200$ ($f = 0.2$), $\rho = 0$, $\sigma_y = 5$, and $d = 10$.

We compare the classical estimator $\bar{Y}_n$ based on the labeled subset ($n = 115$), the classical estimator $\bar{Y}_{\text{full}}$ computed from the entire dataset (768 observations; used as a reference/ground truth), vanilla PPI, CF–PPI, and MEC (quadratic, KL, EL, and Hellinger generators) across four learners—KRR, RF, FNN, and kNN. We report point estimates with 95% confidence intervals. Learner hyperparameters follow Subsection G.1.

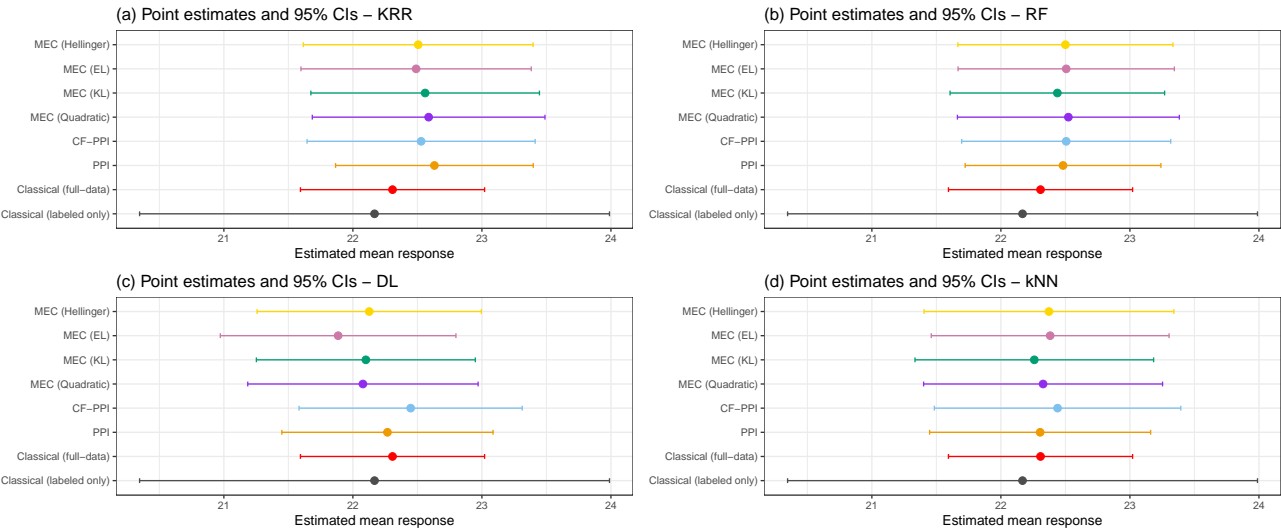

*Figure 10.* Real-data application (aligned case). Point estimates and 95% confidence intervals for the classical, PPI, CF–PPI, and MEC estimators across four learners (KRR, RF, FNN, and kNN) using the Energy Efficiency dataset. The labeled mean $\bar{Y}_n$ is close to the reference mean $\bar{Y}_{\text{full}} = 22.307$. All debiasing methods produce estimates consistent with the reference and yield tighter intervals than the classical labeled-only estimator, demonstrating improved efficiency by leveraging unlabeled covariates.

Because we construct the labeled set by random subsampling of the full data, $\bar{Y}_n$ may differ from $\bar{Y}_{\text{full}}$ due to sampling variability. To assess the robustness of the bias–correction in PPI-based debiased methods (i.e., vanilla PPI, CF–PPI, and MEC), we present two scenarios: one in which $\bar{Y}_n$ is close to $\bar{Y}_{\text{full}}$, and another in which they differ noticeably.

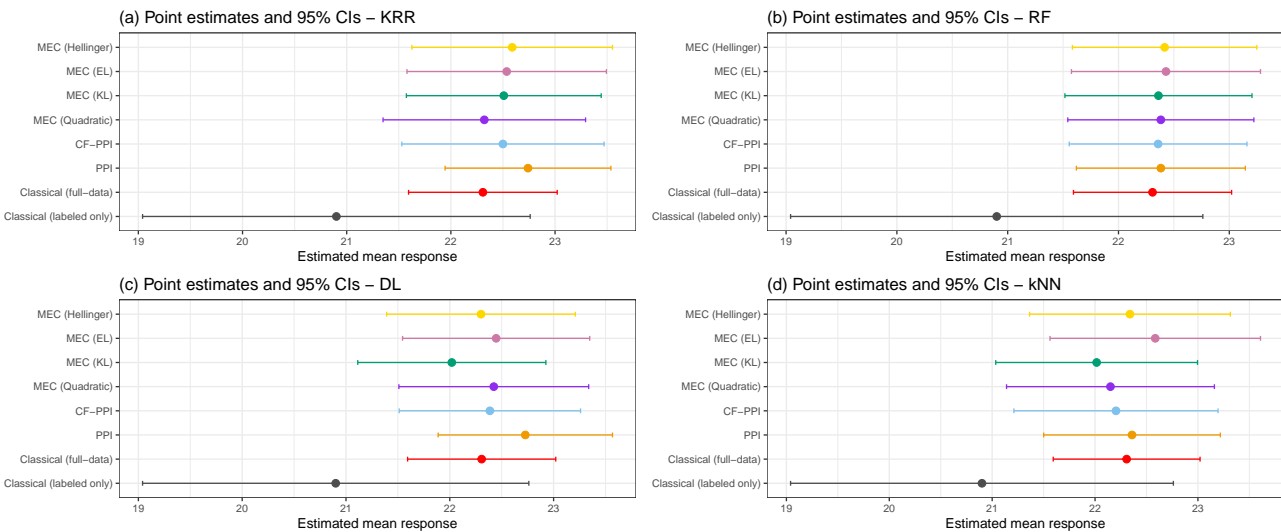

*Figure 11.* Real-data application (off-aligned case). In this scenario, $\bar{Y}_n$ differs noticeably from $\bar{Y}_{\text{full}}$. The bias-correction property of PPI, CF–PPI, and MEC is evident: point estimates are pulled toward the reference mean, and the resulting 95% confidence intervals are tighter than those of the classical labeled-only estimator.

A desirable debiased method should (i) produce a point estimate close to the reference value $\bar{Y}_{\text{full}} = 22.307$, and (ii) yield a 95% confidence interval that is tighter than the labeled-only benchmark $\bar{Y}_n \pm 1.96 \, \widehat{\text{SE}}_n$ yet not tighter than the full-data benchmark $\bar{Y}_{\text{full}} \pm 1.96 \, \widehat{\text{SE}}_{\text{full}}$. Observing these patterns indicates that the debiasing mechanism is operating as intended. Figure 10 and Figure 11 present the results. In the aligned case (Figure 10), all estimators deliver broadly consistent point estimates with only modest differences in precision. The debiased methods (PPI, CF–PPI, and MEC) yield intervals that are tighter than those of the classical labeled-only estimator. Vanilla PPI attains the narrowest intervals among the debiased methods, which may be indicative of overfitting due to the label-reuse. In the off-aligned case (Figure 11), the bias correction of CF–PPI and MEC is more pronounced than for vanilla PPI: their point estimates shift toward the reference mean. Because this is a real-data setting, comparisons between CF–PPI and MEC are difficult; see simulation experiments for comparison.

