# OpenReview forum: "MEC: Machine-Learning-Assisted Generalized Entropy Calibration for Semi-Supervised Mean Estimation"
_ICML.cc/2026/Conference — ICML 2026 regular_

### Official Review · Reviewer_iLWW · 2026-03-02

**Soundness:** 4
**Presentation:** 4
**Significance:** 4
**Originality:** 3
**Overall Recommendation:** 5
**Confidence:** 4

**Summary:**

This paper studies statistical inference for semi-supervised mean estimation. Prediction-powered inference (PPI) has previously been proposed, and several extensions have been developed. In this paper, the authors introduce a weight calibration framework for PPI. The weights are calibrated using a Bregman divergence subject to calibration constraints based on carefully constructed moment conditions.

**Compliance With Llm Reviewing Policy:**

Affirmed.

**Final Justification:**

My concerns have been fully addressed. I have decided to maintain my original score.

**Key Questions For Authors:**

Major Comments

The paper employs the Bregman divergence for weight calibration. Would other divergence measures lead to a similarly effective estimation procedure?

In addition, the target weight is stated as $N/n$. Is this quantity effectively replaced by $1/n$ with a different weight?


Minor Comments

In equation (7), $\delta_i$ is not defined.

Line 252: “The last equality” → “The second and last equalities.”

Line 257: “see Section B.1 of the Appendix for details.” The Lagrange multiplier $\lambda$ has already been mentioned on the right-hand side of page 4.

Equation (12) follows from the calibration constraint; however, this connection may not be immediately clear to the reader. It would be helpful to make this relationship more explicit.

**Limitations:**

yes

**Strengths And Weaknesses:**

Soundness: This paper is built upon two main ideas. The first is the use of cross-fitting, which is already well established in the literature. The second is a weight calibration approach under moment constraints that incorporate both an intercept and an out-of-fold predictor basis. In particular, this latter idea provides meaningful contributions and demonstrates a certain level of originality.

Presentation: The paper is clearly written and easy to follow. It provides many convincing explanations, and the differences from previous research are clearly stated.

Significance: The topic of this paper is fundamental and important. It proposes a simple yet powerful idea for improving semi-supervised mean estimation. To demonstrate the effectiveness of the proposed method, the paper presents necessary conditions in a more organized and systematic manner than previous studies. The proposed framework is also easy for researchers to implement in practice.

Originality: Weight calibration itself is not a new idea. However, the specific calibration constraints adopted in this paper are particularly interesting. While moment constraints typically involve terms such as $1, x, x^2, x^3$, and so on, this paper instead uses a regression model estimated via cross-fitting together with an intercept. This approach yields several desirable properties, including improved statistical efficiency, mitigation of the curse of dimensionality, and computational efficiency.

---

> ### Author Rebuttal · Authors · 2026-03-28
>
> We sincerely thank the reviewer for the very positive and thoughtful assessment of our paper. We are especially grateful that recognized the role of the cross-fitted (out-of-fold) predictor basis as a key source of our paper's originality, as well as the practical and statistical advantages of the proposed calibration formulation. Below, we briefly respond to your questions and comments.
>
> * **Response to Q1: Would other divergence measures (other than Bregman divergence) lead to a similarly effective estimation procedure?**
>   Thank you for this important question. We believe the answer is yes from the perspective of computational efficiency, as long as a dual Newton solver, as in Section B.1 of the Appendix, can be constructed. However, whether using other divergence measures would yield statistical efficiency similar to that of the Bregman divergence remains an open question.
>
>   More specifically, from a purely *computational perspective*, for any well-defined divergence, as long as the chosen divergence leads to (i) a well-posed constrained optimization problem of the form in Eq. (22), and (ii) an explicit dual representation through the convex conjugate, one may still obtain calibrated weights under the same calibration constraint, and our MEC framework can potentially be extended with similar computational efficiency. This is mainly because the computational efficiency ultimately comes from the dual expression via the Lagrange multiplier, which reduces the optimization dimension from $n$ to $2$ in our setting (i.e., the dimension-reduction benefit described in Lines 220-228 would still apply).
>
>   However, it remains an open question how the choice of divergence affects the *statistical properties* of the resulting estimator, relative to the estimator based on the Bregman divergence. We anticipate that, if a different divergence were used, the conditions needed to invoke the asymptotic properties & semiparametric efficiency theory (such as A1-A4 in Theorem 5.2) would also change, since those conditions rely on divergence-specific algebra (e.g., Lemma E.1 for the Bregman divergence).
>
>   We will add a remark in the revision to clarify this point.
>
> * **Response to Q2: Can the baseline weight $d_j = N/n$ ($j=1,\ldots,n$) be effectively replaced by $\tilde d_j = 1/n$ ($j=1,\ldots,n$) with a different normalization?**
>   Thank you for this helpful question. Yes, this is essentially a matter of normalization. In our paper, we use the baseline weight  $d_j=N/n$ so that, summing over the labeled set $S$, we have $\sum_{j\in S} d_j = N.$ This is aligned with the calibrated weights, which satisfy $\sum_{j\in S}\hat \omega_j = N$ under the calibration constraint in Eqs. (9) and (11). Since our target is the population mean, the MEC estimator is naturally expressed in the calibration-estimator form $(1/N)\sum_{j\in S}\hat \omega_j Y_j$; see Eq. (12). Alternatively, one may work with normalized baseline weights $\tilde d_j = 1/n$ and normalized calibrated weights $\tilde\omega_j=\hat \omega_j /N,$ in which case $\sum_{j\in S}\tilde\omega_j=1$ and the estimator becomes $\sum_{j\in S}\tilde\omega_j Y_j.$ We will add a short clarification in the revision to make this point explicit.
>
> * **Response to Minor Comment 1**
>   Thank you for pointing this out. In the revision, we will define the notation $\delta_i$ explicitly when it first appears: $\delta_i=1$ if unit $i$ is labeled, and $\delta_i=0$ otherwise.
>
> * **Response to Minor Comment 2**
>   Thank you for pointing this out. In the revision, we will change "The last equality" to "The second and last equalities," as you suggested.
>
> * **Response to Minor Comment 3**
>   Thank you for pointing this out. In the revision, we will revise the sentence for clarity, since the Lagrange multiplier $\lambda$ has already been introduced earlier on the page, as you noted.
>
> * **Response to Minor Comment 4**
>   Thank you for pointing this out. In the revision, we will revise the flow and make the derivation leading to Eq. (12) more explicit, as you suggested.

---

> > ### Author Rebuttal · Reviewer_iLWW · 2026-04-02
> >
> > Thank you for your replies. I will keep my score.

---

> > > ### Author Response · Authors · 2026-04-02
> > >
> > > Thank you for your review and for your positive assessment of our work. We will revise the manuscript accordingly.

---

### Official Review · Reviewer_tfi5 · 2026-03-09

**Soundness:** 3
**Presentation:** 3
**Significance:** 2
**Originality:** 2
**Overall Recommendation:** 3
**Confidence:** 5

**Summary:**

This paper investigates how to construct statistically valid and efficient inference procedures for estimating the population means in semi-supervised settings, where labeled data are scarce but unlabeled covariates are abundant. This paper develops two new variants of prediction-powered inference (PPI), namely cf-PPI and MEC-PPI, which combine cross-fitting with calibration-based weighting to improve statistical efficiency and robustness to predictor misspecification. The main methodological contribution is the introduction of a calibration-based rectification step using Bregman projections that reweights the labeled observations to better align with the target population, thereby improving efficiency properties relative to existing PPI variants, such as vanilla PPI. The paper establishes consistency and asymptotic normality under relatively mild conditions and shows that the proposed estimator can achieve the semiparametric efficiency bound.

**Compliance With Llm Reviewing Policy:**

Affirmed.

**Final Justification:**

My perspective is as follows. First, the initial part of the paper establishes asymptotic normality for estimating $E[Y]$ in a semi-supervised setting, a classical and well-studied problem. I have attached several references on this topic. Second, the latter part of the paper focuses on calibration, which, to the best of my knowledge, is also well established in the causal inference literature. Off the top of my head, I am aware of references for linear calibration, which I mentioned previously, although I am not certain whether the extension using Bregman divergence has already been studied. Even if it has not, the main novelty of the present paper appears to be extending the calibration step to a general Bregman divergence. In my view, this does not constitute a substantial contribution, since the corresponding calculations seem fairly similar and straightforward. For these reasons, I feel the lack of novelty and have decided to retain my current score.

**Key Questions For Authors:**

1. I would suggest that the authors include a more detailed discussion comparing and contrasting their approach and contributions with the existing literature on semi-supervised inference as well as causal inference (in particular, the missing data framework), based on the points raised in the weakness section above.

2. I would also encourage the authors to discuss/add a remark regarding the estimation of a more general functional of $P_{X, Y}$, the joint distribution of $(X, Y)$, in particular, when it is possible to gain, following their approaches, in terms of asymptotic variance. I believe it should be related to the influence function and its dependence on the marginal distribution of $X$.

I would be happy to revise my score if the concerns are addressed adequately.

**Limitations:**

Yes.

**Strengths And Weaknesses:**

**Strengths**

1. The paper studies an important problem in modern statistics and machine learning, namely, valid statistical inference in semi-supervised settings where labeled data are scarce but unlabeled covariates are abundant. This is a practically relevant regime and closely connected to many real-world applications.

2. The paper is well-written and well-organized. The claims are supported by simulation studies.


**Weaknesses**

1. I would suggest the authors consider the extensive literature on semiparametrically efficient estimation in the semi-supervised setting (see, for example, [1,2] among many others). The issue of semiparametrically efficient estimation under semi-supervised learning has been studied quite thoroughly. A key insight from this body of work is that, in many problems, the availability of additional unlabeled samples helps improve the *variance* of an estimator, but not the *rate* of convergence, unless additional structural assumptions are imposed (for instance, when the covariates lie on a low-dimensional manifold; see [3] for one such example).  Asymptotically, in the semi-supervised regime, we may think of the distribution of $X$ as being essentially known (imagine $N$ is much larger than $n$, which is also studied under the name of ideal semi-supervised estimator in [1], and also in [4]). Consequently, the statistical problem reduces to constructing an efficient estimator for the parameter of interest under the constraint of knowing the marginal distribution of $X$. In many cases, this leads to a reduction in the limiting variance while the convergence rate remains unchanged.

In this context, I would also suggest that the authors present the main results (Theorem 5.1 and Theorem 5.2) in terms of $\sqrt{n}(\hat\theta - \theta_0)$ rather than $\sqrt{N}(\hat\theta - \theta_0)$. Writing the results in terms of $\sqrt{N}$ may give the misleading impression that the method achieves a faster rate (i.e., $N$ instead of $n$), which is not the case here since the authors assume $n/N \to f \in (0,1)$. A clearer presentation, for instance for $\hat\theta_{\rm PPI}^{\rm cf}$, would be $$\sqrt{n}(\hat \theta_{\rm PPI}^{\rm cf} - \theta_0) \implies \mathcal{N}\left(0, f\mathrm{Var}(m_0(X)) + \mathrm{Var}(Y - m_0(X))\right).$$
This formulation makes the interpretation transparent. In particular, when $f=0$, i.e., when $n/N \to 0$ (meaning that the number of unlabeled samples is much larger than the number of labeled samples), the limiting variance becomes $\mathrm{Var}(Y - m_0(X))$, which is precisely the semiparametrically efficient variance in the semi-supervised setting. In other words, the gain here is in the *variance*, not in the *rate*, and this point should be emphasized more clearly.

Furthermore, I would also encourage the authors to connect their results to the literature on causal inference and missing data. The estimator considered here is essentially a simplified version of the augmented inverse probability weighted (AIPW) estimator. Indeed, the present setup can be viewed as a missing-data problem in which labels are missing at random with probability $f \in (0,1)$. This perspective is closely related to causal inference (where treatment outcomes are missing for the control group and vice versa) and has been studied extensively in the literature (see, e.g., [5, 6] for early work).

Overall, given the extensive literature on semiparametric theory in semi-supervised learning, missing data problems, and causal inference, it would be helpful for the authors to position their contribution more clearly in relation to these existing works and to highlight the precise novelty of their approach.




2. Regarding the second approach, namely fine-tuning or post-processing an estimator $\hat m$ of $m_0$, related ideas have also been studied in the causal inference literature (see, e.g., [7, 8] and references therein). The conclusions there are largely similar: if one post-processes the estimator $\hat m$, then (i) there is no loss in terms of the convergence rate or the asymptotic variance (that is why it is called ``no-harm calibration" in [8]), and (ii) consistency and asymptotic normality can be established under weaker conditions, similar to as highlighted by the authors in Theorem 5.2. Although both papers deal with squared error loss rather than general Bregman divergence, the difference is essentially incremental. In light of these connections, it would be helpful for the authors to provide a clearer discussion that compares and contrasts their approach with the existing literature, and to articulate more explicitly what is gained in the present framework relative to what is already known.



*References*

1. Zhang, Anru, Lawrence D. Brown, and T. Tony Cai. "Semi-supervised inference: General theory and estimation of means." (2019): 2538-2566.
2. Azriel, David, et al. "Semi-supervised linear regression." Journal of the American Statistical Association 117.540 (2022): 2238-2251.
3. Azizyan, Martin, Aarti Singh, and Larry Wasserman. "Density-sensitive semisupervised inference." (2013): 751-771.
4. Chakrabortty, Abhishek, and Tianxi Cai. "Efficient and adaptive linear regression in semi-supervised settings." (2018): 1541-1572.
5. Robins, James M., and Ya'acov Ritov. "Toward a curse of dimensionality appropriate (CODA) asymptotic theory for semi‐parametric models." Statistics in medicine 16.3 (1997): 285-319.
6. Robins, James M., Andrea Rotnitzky, and Lue Ping Zhao. "Estimation of regression coefficients when some regressors are not always observed." Journal of the American statistical Association 89.427 (1994): 846-866.
7. Guo, Kevin, and Guillaume Basse. "The generalized oaxaca-blinder estimator." Journal of the American Statistical Association 118.541 (2023): 524-536.
8. Cohen, Peter L., and Colin B. Fogarty. "No-harm calibration for generalized Oaxaca–Blinder estimators." Biometrika 111.1 (2024): 331-338.

---

> ### Author Rebuttal · Authors · 2026-03-26
>
> We appreciate your thoughtful review and comments. Below, we provide a point-by-point response to your questions.
>
> * **1. Re-expression of the asymptotic distribution for CF-PPI/MEC estimators.**
>   We agree that expressing the asymptotic distribution in terms of $n$, rather than $N$, makes the interpretation clearer. **We will revise Theorems 5.1/5.2, together with the related discussion.**
>
> * **2. Response to Q1: contribution of our work within the existing literature and its advantages.**
>   **2-a.** **We appreciate the reviewer’s references, and we will expand the related discussion.** We acknowledge that, ***at a high level***, there is a rich body of literature related to our work and, more broadly, that the core idea behind PPI itself has been “floating around” in related literatures, as acknowledged by the author of PPI (see 25:40 in seminar: https://www.youtube.com/watch?v=TjEWvEKtJM8). Some natural overlap with the survey-sampling/AIPW/semiparametric literatures is inevitable at a high level.
>
>   **2-b.** We believe this is why it is important to assess **the specific topic (in our case: ML-assisted GEC (MEC) for semi-supervised mean estimation)** on which the current work makes a method/theoretical contribution. In our view, the proposed MEC framework has clear merit in that it is, to our knowledge, the first to adapt the generalized entropy calibration (GEC) framework of Kwon et al. (2025) to the PPI framework and to clarify its theoretical advantages by contrasting Theorem 5.1(w/o calibration) with Theorem 5.2 (with calibration). We believe that this insight is novel in the semi-supervised learning literature.
>
>   **2-c.** We agree that our MEC method is closely related to the generalized Oaxaca-Blinder (GOB) method of Guo et al. (2023) and Cohen & Fogarty (2024), with the crucial difference that GOB uses linear calibration explicitly, whereas MEC imposes the calibration structure implicitly via GEC. We will discuss these works in the revision.
>
>   **2-d.** Two advantages of the MEC framework, compared with other popular debiasing frameworks (e.g., DML/TMLE), are as follows:
>
>      * **(I)** The robustness against nuisance estimation error is controlled by the calibration subspace $W=span (h)$ with user-specified basis $h$ in a data-adaptive way. The MEC used $h(\cdot)=(1,\widehat m^{(-)}(\cdot))$, which yields the robustness to affine transformations of the predictor. One may choose a richer one, such as $h(\cdot)=(1,\widehat m^{(-)}(\cdot),\widehat m^{(-)}(\cdot)^2),$ to intensify robustness property (see our response to Reviewer FgG1 for a detailed illustration).
>
>      * **(II)** The current MEC framework (with a single nuance parameter $m(\cdot)$) can be flexibly extended to settings with propensity score estimation (two nuance parameters, $m(\cdot)$ and $\pi(\cdot)$), as described in our response to Reviewer 9Cw2.
>
>   **We will provide much more detailed discussion on this point in the revision.**
>
> * **3. Response to Q2: a remark on more general functionals of $P_{X,Y}$ and when unlabeled covariates are beneficial in improving efficiency.**
>   We agree that semi-supervised mean estimation is a special case in which unlabeled covariates can improve semiparametric efficiency through asymptotic variance reduction and such gains are not automatic for an arbitrary functional of $P_{X,Y}$. Rather, they depend on how the marginal distribution of $X$ enters the efficient influence function (EIF).
>
>   Under a general moment-equation formulation in the semi-supervised setting, $E[U(\theta;X,Y)]=0$, the EIF takes the form $\phi^{eff}(O)= I(\theta_0)^{-1} [\bar U(\theta_0;X)+(\delta/f)(U(\theta_0;X,Y)-\bar U(\theta_0;X))]$, where $I(\theta_0)=E[\partial U(\theta;X,Y)/\partial \theta^\top \mid_{\theta=\theta_0}]$ and $\bar U(\theta;X)=E[U(\theta;X,Y)\mid X]$. Thus, the EIF variance of $\hat\theta$ is $Var(\phi^{eff}(O)) = I(\theta_0)^{-1} \Sigma I(\theta_0)^{-1 T}$, where $\Sigma =  Var[\bar U(\theta_0;X)] + E[Var(U(\theta_0;X,Y)\mid X)]/f$. Seeing the first part ($Var[\bar U(\theta_0;X)]$), the potential efficiency gain from unlabeled covariates is driven by $X$-predictable component $\bar U(\theta_0;X)$. When this varies with $X$, unlabeled covariates can be used to reduce the asymptotic variance. When it is degenerate, such a gain need not occur; in this case, not only MEC but also any method using the unlabeled covariate distribution may fail to yield efficiency gains. In our work, we have $U(\theta;X,Y)=Y-\theta$ and $\bar U(\theta_0;X)=m_0(X)-\theta_0$; thus, unlabeled covariates can improve efficiency, and MEC can be effective. **We will elaborate on this point in detail in revision.**
>
> * **Authors' request for rescoring.**
>   We will revise the paper by expanding the related-work discussion. We respectfully ask you to reconsider your evaluation.
>
> * **References**
>
>   1. Kwon, Y., Kim, J. K., & Qiu, Y. (2025). Debiased calibration estimation using generalized entropy in survey sampling. JASA.

---

> > ### Author Rebuttal · Reviewer_tfi5 · 2026-04-03
> >
> > I sincerely thank the authors for their detailed responses. I would also encourage them to include a more thorough discussion clarifying the novelty of their work relative to the existing semiparametric inference and causal inference literature. Based on the current revision, I would like to maintain my present score.

---

> > > ### Author Response · Authors · 2026-04-03
> > >
> > > Thank you for your comments. We are happy that your second question has been resolved. We also appreciate your suggestion to clarify the novelty of our work relative to the semiparametric inference and causal inference literature. In our first rebuttal, we had essentially exhausted the word limit allowed by the system, and this rebuttal is likewise subject to a strict word limit. We are grateful for the opportunity to state the distinction more explicitly.
> > >
> > > * **Existing Works & Primary Reference.**
> > >
> > >   The novelty of our work does not lie in cross-fitting, the use of weighting calibration, GREG/(A)IPW-style correction, or balancing methods, all of which are well established in semiparametric statistics, causal inference, and survey sampling. These ideas have been extensively developed in the literature on missing data and causal inference, semiparametric efficiency, double/debiased machine learning, and calibration-based survey estimation (e.g., Robins et al., 1994; van der Laan, 2010; Chernozhukov et al., 2018; Deville and Sarndal, 1992; Fuller, 2002; Wu and Sitter, 2001; Hainmueller, 2012; Breidt and Opsomer, 2017; Haziza and Beaumont, 2017). Please understand that it would be quite exhaustive to list all of the relevant works.
> > >
> > >   The recent paper on generalized entropy calibration (GEC) by Kwon et al.(2025) is the primary reference for understanding our contribution. We therefore recommend the references cited in that paper as well. The key difference is that Kwon et al (2025) is concerned with estimating a finite-population mean, whereas our goal is to use GEC, via Bregman projection, for super-population inference under the PPI framework.
> > >
> > > * **What is new in our work?**
> > >
> > >   Our first major novelty is the use of a cross-fitted, data-adaptive predictor basis within the PPI and GEC framework. In principle, any flexible machine learning (ML) method can be used to construct this data-adaptive basis, thereby fully leveraging recent developments in modern ML methods. Methodologically, this represents two further advances over (i) the original PPI framework, in which the ML system is mainly used to construct the bias-correction term (rectifier), and (ii) the GEC work of Kwon et al. (2025), which is primarily designed for moment-feature-type bases. To the best of our knowledge, this specific combination has not been developed in the existing semiparametric, causal inference, or survey sampling literature.
> > >
> > >   Our second major novelty is theoretical. We prove that MEC is robust to affine transformations of the prediction basis, in the sense made precise in our theory (Theorem 5.1/5.2). To the best of our knowledge, no existing work has established comparable theoretical results on the benefits of GEC in the super-population setting, as in Theorems 5.1 and 5.2. Our proof technique for the asymptotic behavior of the MEC estimator is also novel: it relies primarily on empirical process theory, avoids Donsker conditions, which may be restrictive in modern ML settings, and connects naturally to semiparametric efficiency theory (Tsiatis, 2006).
> > >
> > > * **References**
> > >
> > >   1. Robins,J.M., Rotnitzky,A., and Zhao, L.P. Estimation of regression coefficients when some regressors are not always observed. JASA, 1994.
> > >
> > >   2. van der Laan,M. J. Targeted maximum likelihood based causal inference: Part I. The International Journal of Biostatistics, 2010.
> > >
> > >   3. Chernozhukov,V., Chetverikov,D., Demirer,M., Duflo,E., Hansen,C., Newey, W., and Robins, J. M. Double/debiased machine learning for treatment and structural parameters. The Econometrics Journal, 2018.
> > >
> > >   4. Deville,J.-C. and Sarndal,C.-E. Calibration estimators in survey sampling. JASA, 1992.
> > >
> > >   5. Fuller,W.A. Regression estimation for survey samples. Survey Methodology, 2002.
> > >
> > >   6. Wu, C. and Sitter, R. R. A model-calibration approach to using complete auxiliary information from survey data. JASA, 2001.
> > >
> > >   7. Breidt,F.J. and Opsomer,J.D. Model-assisted survey estimation with modern prediction techniques. Statistical Science, 2017.
> > >
> > >   8. Hainmueller,J. Entropy balancing for causal effects: A multivariate reweighting method to produce balanced samples in observational studies. Political Analysis, 2012.
> > >
> > >   9. Haziza,D. and Beaumont,J.-F. Construction of weights in surveys: A review. Statistical Science, 2017.
> > >
> > >   10. Kwon,Y., Kim,J.K., & Qiu,Y. Debiased calibration estimation using generalized entropy in survey sampling. JASA, 2025.
> > >
> > >   11. Tsiatis, A. A.(2006). Semiparametric theory and missing data. New York, Springer.
> > >
> > > ---
> > >
> > >   **We will revise the manuscript thoroughly to clarify the novelty of our work relative to the existing semiparametric inference and causal inference literature. We would also sincerely appreciate your understanding that we reached the word limit permitted by the system. Please find the following link for our answer to your additional question regarding the novelty of our work.**
> > >
> > > **LINK**: https://limewire.com/d/mH1pL#cARsRlHz6P
> > > ---

---

### Official Review · Reviewer_FgG1 · 2026-03-10

**Soundness:** 4
**Presentation:** 3
**Significance:** 3
**Originality:** 4
**Overall Recommendation:** 5
**Confidence:** 3

**Summary:**

The paper proposes an improved PPI method based on generalized entropy calibration. By constructing calibration weights through Bregman projections, the method aims to avoid label reuse while improving robustness to predictor misspecification. A key theoretical claim is that MEC depends on the projection error of the true regression function onto the calibration subspace, rather than the raw prediction error of the learned predictor itself, which may lead to weaker conditions than CF-PPI.

**Compliance With Llm Reviewing Policy:**

Affirmed.

**Final Justification:**

I believe this is a good paper. It leverages GEC for PPI and arrives at an elegant result, while also clarifying its connections to existing PPI methods. The authors have provided detailed responses to the weaknesses and questions I raised, and their clarifications have adequately addressed my concerns. I have therefore increased my score to 5.

**Key Questions For Authors:**

Equation (7) introduces the indicator $\delta_i$ without an explicit definition. Although its meaning can be inferred from the context, it would improve readability if the authors define $\delta_i$ clearly when it first appears.

The paper's use of a two-dimensional calibration subspace is appealing from the standpoint of computational simplicity and numerical stability. On the other hand, enlarging the calibration space by including additional basis functions or multiple predictors might further reduce the projection error and potentially improve efficiency, though possibly at the expense of heavier computation and more variable calibration weights. I would be interested in the authors' view on this tradeoff. Is the two-dimensional construction intended to be theoretically preferred, or mainly adopted as a practically convenient choice?

It would be helpful to show the empirical distribution of the calibrated weights. For example, the authors could report or visualize the weight distribution to assess whether extreme weights occur in practice and whether such values affect the stability or effectiveness of the method.

In Section 4.4, the paper claims that the MEC slope estimator $\hat{\beta}_1$ is asymptotically equivalent to the optimal tuning result in Example 6.1 of PPI++. However, the optimal tuning coefficient in PPI++ includes an additional $(1+n/N)^{-1}$ factor, whereas the MEC expression presented here is a plain covariance-to-variance ratio. Under the asymptotic regime adopted in this paper, namely $n/N \to f \in (0,1)$, these two expressions do not generally coincide asymptotically. I therefore ask the authors to clarify in what precise sense this claimed equivalence should be understood. Moreover, I believe it is necessary to include PPI++ itself as an explicit baseline in the empirical comparison.

**Limitations:**

While Assumptions A2 and A4 are understandable as technical conditions for the theoretical results, they appear relatively strong and not easy to verify in practice. I do not view their use as inherently problematic, but the review form explicitly asks authors to discuss limitations.

**Strengths And Weaknesses:**

Strength
The paper proposes an extension of CF-PPI based on generalized entropy calibration (GEC), leading to an elegant weighted estimator (Equation 12). The authors establish asymptotic properties for both CF-PPI and the proposed MEC estimator, and show that MEC achieves consistency and asymptotic normality under weaker conditions. The paper also provides simulation studies under different generators, machine-learning predictors, and data-generating settings to support the empirical performance of the method. In addition, the manuscript offers useful insights into the relationships between MEC and existing approaches such as CF-PPI and PPI++.

Weakness
The empirical evaluation mainly focuses on prediction performance and confidence interval coverage, while less attention is given to the behavior of the calibrated weights themselves. Although the paper discusses connections with CF-PPI and PPI++, the empirical evaluation does not include PPI++ as a baseline. In addition, it would be helpful to clarify under what settings MEC is expected to provide the most benefit compared with these existing methods.

---

> ### Author Rebuttal · Authors · 2026-03-27
>
> We greatly appreciate your feedback and insight. Below, we provide a point-by-point response to your questions/weakness/limitations.
>
> * **Revision plan: omitted notation $\delta_i$ in Eq. (7).**
>   Thank you for pointing this out. We agree that $\delta_i$ should be defined explicitly when it first appears in Eq. (7): $\delta_i=1$ if unit $i$ is labeled and $\delta_i=0$ otherwise. In the revision, we will introduce this notation.
>
> * **Revision plan: discussion of the PPI++.**
>   Thank you for this important comment. We agree that our current wording about the relationship between MEC and PPI++ needs more detail, due to the factor $1/(1+f)=1/(1+n/N)=N/(N+n)$ in Example 6.1 of Angelopoulos et al.(2024). This factor is close to 1 when $N$ is much larger than $n$, and we omitted detail because this regime is common in typical semi-supervised settings. In the revision, we will clarify this connection. We will also add PPI++ as an explicit baseline in the simulation study and expand the discussion accordingly.
>
> * **Revision plan: discussion on the stability of calibrated weights.**
>   Thank you for your comments on this issue. The calibrated weights, $\hat{\omega}_j$ ($j=1,\dots,n$) indeed play an crucial role, and stability assessment is important in practice. We will provide more detail on this point in the discussion, particularly from the perspective of practical assessment. A common way to assess stability is to visualize the empirical distribution of $\hat{\omega}_j$. In addition, a standard summary measure is the effective sample size (ESS), $ESS = (\sum_j \hat{\omega}_j)^2 / \sum_j \hat{\omega}_j^2 \in [1,n],$ which is close to $1$ when the weights are highly uneven or extreme, and close to $n$ when the weights are nearly uniform. One may also calculate the coefficient of variation (CV), $CV = sd(w)/mean(w),$ which increases as the weights become more dispersed.
>
>   Empirically, the weights of the MEC estimator, based on the out-of-fold predictor basis $h(x)=(1,\hat m^{(-)}(x))$, tend to be stable (ESS close to $n$ and CV relatively small) due to the low-dimensional calibration basis. One may more ambitiously use a richer basis, such as a predictor basis augmented with additional moment features, for e.g., $h(x)=(1,\hat m^{(-)}(x),x_{j1},x_{j1}^2,\ldots,x_{j1}x_{j2})$, but this comes at the cost of heavier computation and potentially less stable weights. (see Line 205~219 in manuscript for relevant references.)
>
>   **Interestingly, different choices of the calibration basis correspond to different estimators/robustness properties:**
>
>    - **(I)** The constant basis $h(x)=1$ reduces to the CF-PPI estimator (Zrnic & Candès(2024)).
>
>    - **(II)** The out-of-fold predictor basis $h(x)=(1,\hat m^{(-)}(x))$ leads to MEC estimator (our work), where the specific choice of quadratic generator is closely related to the PPI++ estimator (Angelopoulos et al.(2024)), and yields robustness to affine transformations of the predictor.
>
>    - **(III)** One may further consider a richer basis such as $h(x)=(1,\hat m^{(-)}(x),\hat m^{(-)}(x)^2,\hat m^{(-)}(x)^3)$, which may provide greater robustness to misspecification of the functional form of the predictor beyond affine transformations.
>
>   - **(IV)** One may consider multiple predictors, for e.g., $h(x)=(1,\hat m_1^{(-)}(x),\hat m_2^{(-)}(x),\hat m_3^{(-)}(x))$, where $\hat m_1(\cdot)$, $\hat m_2(\cdot)$, and $\hat m_3(\cdot)$ are different ML predictors, which provides greater robustness against misspecification of any single predictor by allowing the calibration step to leverage complementary predictive features across learners.
>
>   Importantly, the robustness is asymptotic in nature (Theorem 5.2), whereas weight stability is a finite-sample behavior. Thus, in practice, one must consider the trade-off between robustness and weight stability on a case-by-case basis for the specific problem at hand. As a default, $h(x)=(1,\hat m^{(-)}(x))$ often provides a practical compromise.
>
>   **In the revision, we will add a discussion of weight stability. We will share our programming code for calculating ESS and CV for readers.**
>
> * **Revision plan: discussion of the practical limitations of A2/A4.**
>   Thank you for raising this point. We agree that A2/A4 are not easy to verify directly in practice. In the revision, we will add this limitation and explain that empirical weight-stability diagnostics (ESS/CV), together with visual checks as in Figure 3 can serve as practical complements to these assumptions.
>
> * **Authors' request for rescoring.**
>   Thank you again for your helpful comments. We will revise the paper by clarifying several points raised in your review. We respectfully ask you to reconsider your overall score, or at least the component scores (Soundness/Presentation/Significance).
>
> * **References**
>
>   1. Angelopoulos, Duchi, & Zrnic.(2024). PPI++: Efficient prediction-powered inference. arXiv
>
>   2. Zrnic & Candès (2024).Cross-prediction-powered inference. PNAS

---

> > ### Author Rebuttal · Reviewer_FgG1 · 2026-04-01
> >
> > 1. Thank you for the clarification. My concern is specifically about lines 55--59, where the paper states that \\(n(N)/N \\to f \\in (0,1)\\). This implies a strictly positive asymptotic labeling fraction. However, the earlier discussion frames the regime as \\(N \\gg n\\), which would typically suggest \\(n/N \\to 0\\). These two statements appear inconsistent, and I would encourage the authors to revise the wording for greater logical precision and clarity.
> >
> > 2. At a minimum, I think the revision should provide empirical support in three directions:
> > (1) an explicit comparison with PPI++ as a baseline;
> > (2) concrete weight-stability results, including visualizations of the calibrated weights and summaries such as ESS and CV; and
> > (3) a sensitivity analysis across different calibration bases.
> >
> > Without such results, the current discussion remains largely qualitative, and I would be reluctant to increase my score based on rebuttal text alone. If the authors provide these empirical results, either in the rebuttal or via an anonymous url, I would be willing to increase my score by +1.

---

> > > ### Author Response · Authors · 2026-04-01
> > >
> > > We appreciate your further feedback and follow-up questions, which will help improve the quality of our paper.
> > >
> > > **1. Response to additional question Q1.**
> > >
> > >   Thank you for your clarifying question. We think that the notation $N \gg n$ in this context should be interpreted as referring to some *fixed finite values* of $N$ and $n$, where $N$ is much larger than $n$, for example, $N = 100,000$ and $n = 500$. On the other hand, $n(N)/N \to f \in (0,1)$ is an asymptotic statement as $N \to \infty$ and $n = n(N) \to \infty$. Therefore, these two statements do not conflict with each other; rather, one may view $N \gg n$ as a snapshot along the limiting process $n(N)/N \to f \in (0,1)$ when the limit $f$ is small. In such a snapshot, $N/(N+n)$ is close to 1. We will clarify this point in the revision.
> > >
> > > **2. Response to additional question Q2.**
> > >
> > > We are happy to provide additional simulation results in response to your request. The following table reports simulation results after adding the PPI++ estimator and the feature-moment basis with $h(x_j) = (1,x_{j1},x_{j2},\cdots,x_{j10})\in \mathbb{R}^{11}$ , along with the effective sample size (ESS) and coefficient of variation (CV) as stability measures for the calibrated weights. Here, PPI++ used cross-fitting for a fair comparison with MEC, since MEC uses cross-fitting by default. The simulation setting follows exactly that of Section 6 in the main document with $f =N/(N+n) = 1000/(1000 + 100)= 0.1$. The predictor $\hat{m}(\cdot)$ is fitted using kernel ridge regression (KRR), with its hyperparameters tuned as described in the Appendix. The results are in the following **Table 1**:
> > >
> > > **Table 1. Additional simulation results including PPI++ (Angelopoulos et al., 2023) as a baseline and weight calibration using an 11-dimensional moment-feature basis. ESS and CV are reported only for the calibration methods.**
> > >
> > > | Method | Bias | RMSE | Coverage | Mean CI Len | ESS | CV |
> > > |:--|--:|--:|--:|--:|--:|--:|
> > > | Classical | 0.008 | 0.561 | 0.948 | 2.237 | -- | -- |
> > > | PPI (oracle) | 0.019 | 0.501 | 0.946 | 1.983 | -- | -- |
> > > | PPI (no cross-fitting) | -0.023 | 0.583 | 0.770 | 1.420 | -- | -- |
> > > | CF-PPI | 0.007 | 0.597 | 0.948 | 2.361 | -- | -- |
> > > | Calibration via Moment Basis (Quadratic) | 0.013 | 0.608 | 0.958 | 2.507 | 89.004 | 0.348 |
> > > | Calibration via Moment Basis (KL) | 0.013 | 0.618 | 0.953 | 2.524 | 88.396 | 0.358 |
> > > | Calibration via Moment Basis (EL) | 0.013 | 0.626 | 0.958 | 2.552 | 86.410 | 0.392 |
> > > | Calibration via Moment Basis (Hellinger) | 0.011 | 0.620 | 0.958 | 2.532 | 87.627 | 0.371 |
> > > | MEC (Quadratic) | 0.005 | 0.554 | 0.944 | 2.168 | 98.934 | 0.084 |
> > > | MEC (KL) | 0.007 | 0.556 | 0.938 | 2.168 | 98.876 | 0.086 |
> > > | MEC (EL) | 0.010 | 0.557 | 0.940 | 2.166 | 98.839 | 0.087 |
> > > | MEC (Hellinger) | 0.010 | 0.557 | 0.943 | 2.168 | 98.836 | 0.087 |
> > > | PPI++ (with cross-fitting) | 0.004 | 0.556 | 0.944 | 2.171 | -- | -- |
> > > NOTE: Mean CI Len: mean length of the 95% confidence interval, RMSE: root mean squared error.
> > >
> > > In the above results, we can see that the operating characteristics of PPI++ are very close to those of MEC (Quadratic). In fact, the MEC results are also quite similar across different generators. This is why, in the paper, we mainly report the MEC results with the quadratic generator. As seen in the table, for MEC methods, the ESS is very close to $n = 100$ and the CV is close to 0, indicating stable weights for MEC based on the 2-dimensional predictor basis. In contrast, the moment-calibration basis (which is not data-adaptive) leads to ESSs around 90 and moderately-small CVs, indicating less stable weights. Indeed, as shown in the table, calibration using moment features performs worse than CF-PPI in terms of RMSE (mainly due to the instability of calibrated weights). For this reason, in the paper, we do not recommend using a moment-feature basis for the semi-supervised mean estimation problem, although it is conventional in survey sampling (Lines 205-209), and instead recommend using a predictor basis.
> > >
> > > We hope this resolves some of your concerns, and we would be happy to discuss any further questions you may have.
> > >
> > > *NOTE: As for richer basis functions, this is part of our future research. In particular, the basis with the item (IV) in our rebuttal theoretically seems related to the Super Learner method (Van der Laan et al., 2007). We respectfully note that this direction is beyond the scope of the present paper.*
> > >
> > > * **References**
> > >
> > > 1. Angelopoulos, A. N., Duchi, J. C., & Zrnic, T. (2023). Ppi++: Efficient prediction-powered inference. arXiv preprint arXiv:2311.01453.
> > >
> > > 2. Van der Laan, M. J., Polley, E. C., & Hubbard, A. E. (2007). Super learner.
> > >
> > > ---
> > >
> > > **Thank you for your review and for your positive assessment of our work reflected in your raised score. We will revise the manuscript accordingly.**
> > >
> > > ---

---

### Official Review · Reviewer_9Cw2 · 2026-03-11

**Soundness:** 3
**Presentation:** 3
**Significance:** 2
**Originality:** 2
**Overall Recommendation:** 2
**Confidence:** 5

**Summary:**

The paper studies estimation of a mean outcome with missing labels by proposing a PPI-style estimator based on calibrated weighting. Concretely, the method combines cross-fitting with balancing weights constructed through a Bregman-divergence formulation, placing the proposal in the broader literature on calibrated weighting and semiparametric estimation under missing at random. The paper therefore offers a perspective that connects the PPI framework to established ideas from missing-data theory, causal inference, survey calibration, and debiased machine learning. In addition to the estimator itself, the paper provides theoretical guarantees for consistency and asymptotic normality under standard assumptions, thereby giving a formal inferential treatment of this weighting-based approach to label-missingness problems.

**Compliance With Llm Reviewing Policy:**

Affirmed.

**Final Justification:**

## Decision summary

Overall, the authors did a poor job of situating the paper within the prior literature on semiparametric statistics, missing data, and cross-fitted AIPW/debiased machine learning/targeted learning. Although the first rebuttal helped clarify some of the intended contributions, I raised additional prior work that appears highly relevant. The authors indicated that they would add citations, but they did not further clarify how their results relate to that literature.

After reviewing the relevant literature myself, my view is that the proposed estimators and much of the supporting theory are either special cases of existing results in semiparametric missing-data theory or require only modest extensions of prior work. That said, I do think the paper has some value in bringing balancing-weighting ideas from these areas into PPI. But as written, it does not clearly distinguish what is new from what is already known. For that reason, I recommend rejection.

## What is new and what is not

The paper studies estimation of a mean outcome under missing at random (MAR), a classical problem with a large semiparametric literature [1--4]. In a precise sense, this problem is already well understood: the class of regular, asymptotically linear, $\sqrt{n}$-consistent estimators has long been characterized through influence-function theory; see, for example, the foundational work of Robins, Rotnitzky, and coauthors, as well as [9]. With machine learning, the standard estimators are AIPW and TMLE, and the authors' proposal appears to be a special case of the latter. The paper seems to fit directly into this framework, yet it provides little discussion of that connection.

The specific PPI estimator proposed in the paper appears to be a special case of **cross-fitted targeted minimum loss estimation (TMLE)** [1,2], a standard semiparametric framework for combining cross-fitting and machine learning while retaining valid asymptotic inference. In this formulation, the estimator updates the missingness-probability weights using a **Bregman divergence** loss and the covariates $(1,\hat m(X))$, where $\hat m$ is a preliminary estimator of the outcome regression. This choice is not arbitrary: it is motivated by the influence function for the mean outcome under MAR. The intercept enforces consistency for the true constant weight, while inclusion of $\hat m(X)$ ensures that the weights balance the relevant function so that the empirical influence-function moment is zero. This is a standard TMLE construction and yields a weighted estimator equivalent to an **AIPW estimator**. Accordingly, both the estimator and its asymptotic theory appear to fall within the existing theory for cross-fitted TMLE/AIPW estimators developed in [1] and [3].

A closely related TMLE estimator appears in Section 6.4 of [2] for estimating a counterfactual mean outcome. There, logistic loss is used to target, or balance, the missingness probabilities with respect to $\hat m(X)$. Relative to that construction, the main differences here seem to be the choice of adjustment loss and the inclusion of an intercept to represent the constant weight. Accordingly, the proposed estimator is better viewed as a particular implementation of **cross-validated (or cross-fitted) TMLE** [1], rather than as a fundamentally new estimation framework.

Even setting TMLE aside, the equivalence between balancing-weight estimators and AIPW estimators in missing data is not new; a modern treatment is given in [5b], see also [5a].

The main remaining point of potential novelty is the use of a **Bregman divergence** as the loss function for the targeted nuisance. However, this also has prior precedent: the use of Bregman divergences in TMLE/AIPW-type procedures, more broadly within debiased machine learning, is studied in [7a] and [7b], with closely related ideas also appearing in the balancing literature [5,6].

As a result, the paper’s theorems appear to follow from prior work, with at most minor modifications in some cases.

- **Theorem 4.1** states a known equivalence between balancing weights and AIPW estimators; see, for example, [5a,5b, 6]. Since the underlying balancing equations do not depend on the use of a Bregman divergence, this result appears to follow directly, or with only minor modification, from existing results. If there are differences, the authors should discuss them
- **Theorem 5.1** establishes asymptotic normality of a standard cross-fitted AIPW estimator. Without cross-fitting, this is classical [8,9]. Cross-fitting to accommodate generic machine learning is by now standard in debiased machine learning [1,3,4], so the analysis likewise appears to follow directly from existing theory.
- **Theorem 5.2** appears to be the paper’s most novel result. Even so, it seems best viewed as an incremental extension of Theorem 1 in [1], since the estimator here is a specific instance of TMLE. The main distinction is that the authors relax some nuisance-consistency and rate requirements by exploiting calibrated loss functions for the weights. But this idea is also not new: calibration-based losses for weight and probability estimation have already been used to weaken such requirements in [10,11], among other related works.

These connections and references are not discussed in the paper, despite being highly relevant to the estimator and its theory. They should be cited explicitly, and the paper should clearly distinguish what is genuinely new from what is a special case, reformulation, or modest extension of prior work. Addressing this would require substantial revision and a materially different manuscript. I therefore recommend rejection, while encouraging the authors to revise the paper so its contributions can be more fairly assessed.


**TMLE / DML**

[1] Zheng and van der Laan (2011), *Cross-validated targeted minimum-loss-based estimation*.

[2] van der Laan and Rubin (2006), *Targeted maximum likelihood learning*.

[3] van der Laan and Rose (2011), *Targeted Learning*.

[4] Chernozhukov et al. (2018), *Double/debiased machine learning for treatment and structural parameters*.

**Balancing weights / AIPW**

[5a] Bruns-Smith et al. (2025), *Augmented balancing weights as linear regression*.

[5b] Hejazi and van der Laan. "Revisiting the propensity score's central role: Towards bridging balance and efficiency in the era of causal machine learning."


[6] Chattopadhyay and Zubizarreta (2023), *On the implied weights of linear regression for causal inference*.




**Bregman-based weighting**

[7a] Hines and Miles (2025), *Learning density ratios in causal inference using Bregman-Riesz regression*.

[7b] Kato (2025), *A Unified Theory for Causal Inference: Direct Debiased Machine Learning via Bregman-Riesz Regression*.

**Classical missing-data / DR**

[8] Bang and Robins (2005), *Doubly robust estimation in missing data and causal inference models*.

[9] van der Laan and Robins (2003), *Unified Methods for Censored Longitudinal Data and Causality*.

**Calibrated losses**

[10] Ghosh, S., and Z. Tan (2022). "Doubly robust semiparametric inference using regularized calibrated estimation with high-dimensional data."\

[11] Tan, Z  (2020)  "Model-assisted inference for treatment effects using regularized calibrated estimation with high-dimensional data."

**Key Questions For Authors:**

**Questions for the authors.**

- It would be helpful for the paper to state more explicitly which aspects of the proposed method are genuinely new relative to the existing literature.

- More specifically, is the use of balancing or calibration weights for estimating IPW-type targets, such as mean outcomes under missing at random or counterfactual means in causal inference, intended as a novel contribution of this paper? If not, it would help to clarify how the present estimator differs from existing balancing-weight and calibration-based estimators.

- Similarly, is the representation or interpretation of the calibrated-weight estimator as an AIPW-type estimator new, or is it mainly a reformulation of known connections between balancing estimators and regression-based or doubly robust estimators? A clearer discussion of this point would help readers understand the paper's relationship to prior work.

- It would be helpful to clarify which parts of the asymptotic theory are genuinely new. In particular, how does the analysis differ from standard results for cross-fitted DML or AIPW estimators, or from existing theory for balancing-weight estimators of mean outcomes? The paper suggests that one distinction from parts of the survey-sampling calibration literature is the use of a superpopulation framework rather than a finite-population or subpopulation framework. However, the superpopulation framework is already standard in semiparametric statistics and causal inference, and calibration-based estimators in survey sampling are closely connected to AIPW and related machine-learning-assisted estimators. As written, it is therefore unclear whether this aspect represents a new theoretical contribution or mainly a reformulation of existing ideas in a different framework.


- More broadly, if the main contribution is to introduce calibration weighting into the PPI framework, I would encourage the authors to emphasize this contribution directly and explain more clearly how it extends, differs from, or improves upon existing semiparametric and missing-data methods.

**Limitations:**

Yes

**Strengths And Weaknesses:**

This paper proposes a variant of prediction-powered inference (PPI) for estimating a mean outcome with missing labels using calibrated weighting. The method constructs balancing weights via Bregman divergences and yields a cross-fitted estimator closely connected to classical balancing-weight and augmented inverse-probability estimators for mean estimation under missing-at-random labels. The paper is clearly written and the method appears technically sound. In my view, the main contribution is to bring these ideas into the PPI setting, together with a data-adaptive weighting construction and a superpopulation-style asymptotic analysis. I found the rebuttal helpful and have raised my score. My main remaining concern is that the paper should position its contributions more carefully relative to the literatures on missing-data estimation, calibrated weighting, targeted learning, and debiased machine learning. As currently written, the related-work discussion does not fully clarify which aspects are genuinely new and which are extensions of existing ideas to the PPI setting.

Overall, I think the paper could be a useful contribution, especially if revised to better clarify its scope and relationship to prior work. In particular, the novelty seems to lie less in the use of superpopulation-based semiparametric tools themselves, which are standard in adjacent literatures, and more in their adaptation to PPI, together with the particular weighting formulation and dimension-reduction ideas.

**Weaknesses**

- **Positioning relative to prior work could be clearer.** Supervised learning with missing outcomes is fundamentally a missing-data problem, and there is a large literature spanning roughly three decades on estimating mean outcomes under missing at random while allowing flexible machine-learning methods. In particular, influence-function-based approaches such as doubly robust estimation, debiased machine learning, and targeted learning have been developed for this setting, as well as for substantially more challenging variants. The proposed estimator appears to be largely a special case of AIPW-type and balancing-weight estimators for missing-data problems, a class long studied in semiparametric statistics, causal inference, and missing-data analysis; see, for example, Robins, Rotnitzky, and Zhao (1994), van der Laan and Robins (2003), and van der Laan and Rose (2011). More generally, the paper does not yet clearly situate either the estimator or its supporting theory within this broader literature. This does not diminish the value of the paper, but it does suggest that the paper should more clearly separate its genuinely new contributions from the surrounding, already well-established framework.

- **Some components have important precedents.** The use of calibrated weighting and Bregman-divergence-based balancing weights is connected to a broader literature in survey sampling, causal inference, and debiased machine learning. Likewise, data-adaptive updates based on preliminary estimators have clear parallels in targeted learning/TMLE and related semiparametric work. The paper’s contribution would come through more clearly if these connections were discussed more explicitly.

- The asymptotic theory is standard for cross-fitted AIPW or debiased machine learning estimators. In particular, the authors assume consistency of the machine learning model to establish consistency and asymptotic normality, which is standard in the debiased machine learning literature. However, this is less consistent with the original motivation of PPI. A central feature of the original PPI paper by Anastasios et al. was that valid inference could be obtained without requiring model consistency. Similarly, the cross-fitting extension, Cross-PPI, developed by Zrnić and Candès, used stability arguments to establish inference without consistency assumptions. Although the authors acknowledge that consistency-based arguments are more standard in semiparametric inference, this choice removes much of what is distinctive about the PPI framework.


- Bruns-Smith, David, Oliver Dukes, Avi Feller, and Elizabeth L. Ogburn. “Augmented balancing weights as linear regression.” *Journal of the Royal Statistical Society: Series B (Statistical Methodology)* (2025), qkaf019.

- Chattopadhyay, Ambarish, Christopher H. Hase, and José R. Zubizarreta. “Balancing vs modeling approaches to weighting in practice.” *Statistics in Medicine* 39(24) (2020): 3227–3254.

- Chattopadhyay, Ambarish, and José R. Zubizarreta. “On the implied weights of linear regression for causal inference.” *Biometrika* 110(3) (2023): 615–632.

- Hainmueller, Jens. “Entropy balancing for causal effects: A multivariate reweighting method to produce balanced samples in observational studies.” *Political Analysis* 20(1) (2012): 25–46.

- Hines, Oliver J., and Caleb H. Miles. “Learning density ratios in causal inference using Bregman-Riesz regression.” *arXiv* preprint arXiv:2510.16127 (2025).

- Josey, Kevin P., Elizabeth Juarez-Colunga, Fan Yang, and Debashis Ghosh. “A framework for covariate balance using Bregman distances.” *Scandinavian Journal of Statistics* 48(3) (2021): 790–816.


- Song, Yilin, Dan M. Kluger, Harsh Parikh, and Tian Gu. “Demystifying Prediction Powered Inference.” *arXiv* preprint arXiv:2601.20819 (2026).

- Wang, Yixin, and José R. Zubizarreta. “Minimal dispersion approximately balancing weights: asymptotic properties and practical considerations.” *Biometrika* 107(1) (2020): 93–105.

---

> ### Author Rebuttal · Authors · 2026-03-26
>
> We greatly appreciate your comments. We agree that our work builds on several prior works: generalized entropy calibration (GEC) (Kwon et al.,2025), PPI (Angelopoulos et al.,2023), Bregman projection (Hines & Miles,2025), and model assisted survey sampling (Kim et al.,2025), under semi-supervised mean estimation/semiparametric inference (Zhang et al.,2019).
>
>   * **Cordial request for correction.**
>
>     **We respectfully believe two precedence claims are inaccurate. To preserve double-blind review, we do not elaborate further.**
>
>     $\fbox{C-1}$ **Pervin et al.(2026) does not precede our work based on the objective submission timeline. We respectfully ask the reviewer to consult the references in that paper. This is a major issue, as it bears directly on the originality of our work.**
>
>     $\fbox{C-2}$ **The 7th reference should be Kim et al.(2025), rather than Kwon et al.(2025), published in *Survey Methodology*. This is a minor issue.**
>
>   We maintain that our work makes a distinct method/theoretical contribution beyond the existing works:
>
> * **1. Overall originality of our work**
>
>   **1-a.** We note that the work by Kwon et al. (published in JASA in Sep 2025) is a primary reference for our work, extending the entropy calibration of Deville and Särndal(1992) to GEC framework. However, both are developed under a *finite-population for survey sampling*, whereas our work is formulated under a *superpopulation framework for semi-supervised mean estimation*. **Accordingly, the proof techniques differ substantially: we rely mainly on empirical process/semiparametric efficiency theory. To our knowledge, no reference in your Official Review developed GEC through an empirical process-based analysis as we do.**
>
>    **1-b. At a high-level,** some overlap with the survey sampling/AIPW/semiparametric literatures is inevitable, as noted in our response to Reviewer tfi5.
>
> * **2. Method/Theoretical originality relative to prior work**
>
>   **2-a.** Our work is the first to adapt the GEC framework to PPI in a principled way, where the rectifier is incorporated through a Bregman projection under the calibration constraint $\sum_{j \in S} \omega_j h(X_j) = \sum_{i=1}^N h(X_i),$ with the cross-fitted predictor basis $h(X) = (1, \hat{m}^{(-)}(X)).$
>
>    **2-b.** This specific use of the predictor basis (2-a) distinguishes our work from Josey et al.(2020). Although both their work and ours adopted Bregman projection, Josey et al. used pre-specified balance functions (i.e., non-data-adaptive basis), whereas **we used the out-of-fold predictor basis $h(X)=(1,\hat m^{(-)}(X))$ (i.e., data-adaptive basis)**.
>
>   **2-c.** We note that the work by Pervin et al. (2026) is empirical/review-oriented, whereas we provided a deeper theoretical result showing that the benefit of weight calibration via the predictor basis is $\|\|m_{0,\perp}\|\| \leq \|\|\hat m^{(-)} - m_0 \|\|$ (see Eq. (16) in our paper); MEC requires a weaker condition than PPI/CF-PPI to achieve the semiparametric efficiency lower bound. **This point is novel in the literature**.
>
>    **2-d.** Through the dual representation of the MEC estimator (Theorem 4.1), we show that the proposed MEC is closely related to the PPI++ (Angelopoulos et al.,2024) as a special case when the quadratic generator is chosen. **To the best of our knowledge, this connection is novel**.
>
> * **3. Flexible extension of our MEC framework to Poisson labeling designs (missing-at-random setting).**
>
>   While we mainly work under the original PPI setup with a single nuisance parameter (predictor $m(x)$) and a known labeling fraction $f=n/N,$ our MEC framework can be readily extended to Poisson labeling design as follows:
>
>     - First, introduce the label observation indicator $\delta_i$ and unknown labeling propensity $\pi_i=\pi(X_i)=P(\delta_i=1 \mid X_i,Y_i)=P(\delta_i=1 \mid X_i),$ which is estimated by $\hat\pi_i=\hat\pi(X_i),$ using ML methods, and replace the rectifier term in PPI by the IPW form $(1/N)\sum_{i=1}^N (\delta_i/\hat{\pi}_i)(Y_i-\hat m^{(-)}(X_i)).$
>    - Second, perform the Bregman projection using the baseline weights $\hat d_i = 1/\hat{\pi}_i$ with the predictor basis $h(X)=(1,\hat m^{(-)}(X)).$
>
>   This extension leads to an AIPW-type MEC estimator. **We will provide details in the revision.**
>
> * **Authors' request for rescoring.**
>   We will revise the paper by expanding the discussion in greater detail and by explicitly stating the paper’s contribution. We respectfully ask that you kindly take our correction request into account in your review and reconsider your assessments of originality and the overall score.
>
> * **References**
>
>   1. Zhang,A., Brown,L. D., & Cai,T. T.(2019).Semi-supervised inference: General theory and estimation of means.AOS
>
>   2. Kwon,Y., Kim,J. K., & Qiu,Y.(2025).Debiased calibration estimation using generalized entropy in survey sampling.JASA.
>
>   3. Angelopoulos,A. N., Duchi,J. C., & Zrnic,T.(2024).PPI++: Efficient prediction-powered inference.arXiv

---

> > ### Author Rebuttal · Reviewer_9Cw2 · 2026-04-03
> >
> > Thank you for the clarification. I appreciate the authors’ response and their commitment to better situating the paper within the semiparametric, causal-inference, and missing-data literatures, and I have updated my assessment accordingly. In particular, I appreciate the correction that Pervin et al. (2026) does not predate this work. This does not materially change my overall review, since I had cited it as one of several related references on covariate balancing via Bregman divergences and related special cases, an area that I felt was not adequately discussed.
> > The authors also acknowledge prior work on using Bregman divergences to construct balancing weights in the covariate-balancing literature (e.g., Josey et al.). From that perspective, the paper’s novelty seems to lie less in Bregman-based weighting itself and more in the use of a data-dependent predictor basis, together with its adaptation and analysis in the PPI setting.
> >
> > In my view, the paper’s main contributions are: (1) bringing balancing-weight ideas from causal inference and semiparametric statistics into PPI; and (2) introducing a dimension-reduced weighting approach building on prior work using Bregman divergences and balancing. However, as written, the paper’s positioning and related-work discussion should be substantially revised.
> >
> > **(Re 2)**
> >
> > - I agree that the specific data-dependent basis $(1,\hat\mu)$ may be novel in the context of PPI. More broadly, however, balancing on data-dependent features is not and should be situated relative to prior work. For example, [1] studies balancing on targeted, data-dependent functions that approximate the outcome regression; the authors' more limited choice $(1,\hat\mu)$ appears to be a special case. There are also clear links to the TMLE/targeted learning literature [2,5,6], where nuisance estimates are updated so that weighting or regression estimators take an AIPW/TMLE form. Relatedly, [7], [8], and outcome-adaptive lasso [11] use preliminary outcome-regression information or other data-dependent summaries for dimension reduction or propensity-score adjustment. These approaches are not identical, but they provide important context and should be discussed.
> >
> > **(Closely related to TMLE)**
> >
> > Section 6.4 of [2] describes a post hoc targeting step that uses an initial outcome-regression estimate to update a propensity-score model so that the resulting weighted estimator matches an AIPW/TMLE estimator. The parallel here is that, in both cases, a first-stage outcome-regression estimate informs a subsequent weighting adjustment. The present approach differs in two main ways: it calibrates the weights directly, rather than through a propensity model, which makes it closer in spirit to automatic debiased machine learning [9]; and it uses a balancing basis that includes both an intercept and a preliminary outcome-regression estimate, while the TMLE-style approach uses the intercept as offset and the outcome-regression estimate as ``clever covariate". Overall, the estimator appears to be a targeted-weighting adaptation to the present setting, with the extension beyond [2] seeming fairly incremental.
> >
> > **(Re 1a)**
> >
> > -  I do not view the superpopulation/empirical process perspective as the paper's main methodological novelty. The proposed estimator appears to fall within the broader class of AIPW-type estimators for missing-data problems, for which empirical process methods and semiparametric efficiency theory are already standard [1,2,4,6]. From this perspective, the paper brings these ideas into PPI rather than introducing a fundamentally new analytical framework. I am comfortable with that positioning, provided the relevant pre-PPI literature is clearly cited and the contribution is scoped accordingly.
> >
> > [1] van der Laan MJ, Robins JM. *Unified Methods for Censored Longitudinal Data and Causality*. Springer, 2003.
> >
> > [2] van der Laan MJ, Rose S. *Targeted Learning*. Springer, 2011.
> >
> > [4] Robins JM, Rotnitzky A, Zhao LP. Analysis of semiparametric regression models for repeated outcomes with missing data. *JASA*. 1995;90(429):106–121.
> >
> > [5] van der Laan MJ, Rubin D. Targeted maximum likelihood learning. 2006.
> >
> > [6] Chernozhukov V, Chetverikov D, Demirer M, Duflo E, Hansen C, Newey W, Robins J. Double/debiased machine learning for treatment and structural parameters.
> >
> > [7] Benkeser D, Cai W, van der Laan MJ. A nonparametric super-efficient estimator of the average treatment effect. 2020:484–495.
> >
> > [8] D’Amour A, Franks A. Deconfounding scores: feature representations for causal effect estimation with weak overlap. *arXiv:2104.05762*, 2021.
> >
> > [9] Chernozhukov V, Newey WK, Singh R. Automatic debiased machine learning of causal and structural effects. *Econometrica*. 2022;90(3):967–1027.
> >
> > [10] Wainstein L, Bai H. Targeted function balancing. *arXiv:2203.12179*, 2022.
> >
> > [11] Shortreed SM, Ertefaie A. Outcome-adaptive lasso: variable selection for causal inference. *Biometrics*. 2017;73(4):1111–1122.

---

> > > ### Author Response · Authors · 2026-04-03
> > >
> > > **[First Reply Rebuttal Comment]**
> > >
> > > Thank you for your responses. We will revise the manuscript thoroughly based on your recommendations and suggestions, which will greatly improve the quality of our paper. We also sincerely appreciate the additional references you provided, which we will carefully review and incorporate. Above all, we appreciate your updated and more favorable view of our paper compared with your initial assessment.
> > >
> > > ---
> > >
> > > **[Second Reply Rebuttal Comment]**
> > >
> > > * **Procedural Unfairness: Authors' concern about the reviewer's delete/edit of the initial [Official Review (Weaknesses)] after our [Rebuttal] had already been submitted.**
> > >
> > >   We respectfully note that your initial **[Official Review (Weaknesses)]** appears to have been changed after our **[Rebuttal]** had already been submitted. We prepared our response in **[Rebuttal]** based on the version of the **[Official Review]** visible during the rebuttal period. Because some contents in the initial **[Official Review (Weaknesses)]** to which we had already successfully replied (and your score was raised from 2 to 3) now appears to have been deleted, and because new content has been added to **[Official Review (Weaknesses)]** after our rebuttal was submitted (and your score was then lowered again from 3 to 2), the current record may no longer fully reflect the exchange as it originally occurred. This may make it difficult for other reviewers/AC/SAC/PC to objectively evaluate the discussion.
> > >
> > > *  **Deleted references/paragraph from the initial [Official Review (Weaknesses)] after our [Rebuttal] had already been submitted.**
> > >
> > >        "The main distinctive feature appears to be the use of Bregman divergences to construct balancing weights, but this idea also has substantial precedent in the literature; see, for example, Josey et al. (2021), Kwon et al. (2025), and Pervin et al. (2026). See also Hines et al. (2025) for a recent review."
> > >
> > >      - We note that this discussion point, originally raised as a weakness of our method, was deleted after the rebuttal.
> > >
> > >     As noted in **[Rebuttal]**, your assessment treating Pervin et al.(2026) as prior work relative to ours is incorrect. In addition, your citation to Kwon et al.(2025) was inaccurate:
> > >
> > >     1. Pervin,M.,Wang,H.,&Kim,J.K.(2026). Generalized entropy calibration for inference with partially observed data: A unified framework. arXiv.[v1 submitted on Sun,22Feb2026]
> > >
> > >     2. Kwon,Y.,Qiu,Y.,& Park,J.(2025). Model-assisted calibration estimation using generalized entropy calibration in survey sampling. Survey Methodology
> > >
> > >     Additionally, we point out that the following reference in **[Official Review]** was submitted on [Wed,28Jan 2026], which coincides with ICML submission deadline. Thus, it may not appropriately be treated as prior work:
> > >
> > >     3. Song, Yilin, Dan M. Kluger, Harsh Parikh, and Tian Gu. "Demystifying Prediction-Powered Inference.'' arXiv (2026).
> > >
> > >   *  **Added paragraph from the revised [Official Review (Weaknesses)] after our [Rebuttal] had already been submitted.**
> > >
> > >          "Likewise, data-adaptive updates based on preliminary estimators have clear parallels in targeted learning/TMLE and related semiparametric work. The paper’s contribution would come through more clearly if these connections were discussed more explicitly."
> > >
> > >      - We note that the phrases **"clear parallels in targeted learning/TMLE''** and **"these connections"** did NOT appear in your initial **[Official Review]**. Thus, we did not have any opportunity to address this point in **[Rebuttal]**, nor to discuss the connection between MEC and TMLE there.
> > >
> > >       As for **(Closely related to TMLE)** in **[Rebuttal Acknowledgement]**, while we understand your view that MEC and TMLE are parallel at a high level, we respectfully disagree with the implication that MEC is merely a fairly incremental extension of TMLE. In our understanding, TMLE update is carried out via a targeted fluctuation of an initial estimator using a clever covariate so that the resulting estimator solves the efficient influence function estimating equation, with its intuition arising from semiparametric efficiency theory, as developed by van der Laan and collaborators(2006). By contrast, MEC directly constructs calibrated weights through a Bregman projection under PPI framework with a data-adaptive predictor basis, closely connected to survey sampling literature beginning with Deville and Sarndal(1992). Thus, while there is a conceptual parallel, estimator construction, mathematical role of the update step, and proof strategy are largely different.
> > >
> > >   * **Request to Reviewers/AC/SAC/PC.**
> > >
> > >     We respectfully request reconsideration of the assessment in light of the potential procedural unfairness caused by post-rebuttal edits to **[Official Review]**. We would like to share the initial **[Official Review]** and our rebuttal to reviewer's **[Final Justification]** at the following link:
> > >
> > > ---
> > > **LINK**: https://limewire.com/d/DEi61#Q8rtKtjhbo
> > > ---

---

### Decision · Program_Chairs · 2026-04-30

**Decision:**

Accept (regular)

**Comment:**

This paper proposes Machine-Learning-Assisted Generalized Entropy Calibration (MEC) for semi-supervised mean estimation. By utilizing Bregman projections onto a cross-fitted predictor basis, the authors introduce a calibration-weighted variant of prediction-powered inference (PPI).
The review process revealed a sharp divide between practicality and theoretical rigor. Reviewers FgG1 and iLWW championed the paper, as they appreciated the method's computational elegance and finite-sample weight stability. The other two reviewers raised concerns regarding the paper's theoretical framing. The ideas proposed in this paper heavily overlap with classical semiparametric missing-data theory and causal inference (e.g., AIPW/TMLE); however, they fail to clearly discuss this overlap.  Crucially, the manuscript initially presented its asymptotic theorems scaled by the total population size ($\sqrt{N}$) rather than the labeled sample size ($\sqrt{n}$). This created a misleading illusion of an improved convergence rate, whereas the method actually only achieves standard variance reduction. This was partially resolved during the rebuttal phase.

In light of this overlap the contributions of this paper are 1) adapting the calibration to Bregman projection and 2) translating ideas from semiparametric inference into another field (PPI). While there can be a great value in this translation work, this needs to be done honestly by appropriately discussing the literature.

Note on the review process: The authors raised valid procedural concerns regarding Reviewer 9Cw2 (e.g., chronological inaccuracies in cited prior work). These procedural factors were noted, but they do not invalidate the foundational theoretical critiques raised independently by Reviewer tfi5.